# Symmetries in Overparametrized Neural Networks: A Mean-Field View

**Javier Maass Martínez**
Center for Mathematical Modeling
University of Chile
javier.maass@gmail.com

**Joaquín Fontbona**
Center for Mathematical Modeling
University of Chile
fontbona@dim.uchile.cl

## Abstract

We develop a Mean-Field (MF) view of the learning dynamics of overparametrized Artificial Neural Networks (NN) under distributional symmetries of the data w.r.t. the action of a general compact group $G$. We consider for this a class of generalized shallow NNs given by an ensemble of $N$ multi-layer units, jointly trained using stochastic gradient descent (SGD) and possibly symmetry-leveraging (SL) techniques, such as Data Augmentation (DA), Feature Averaging (FA) or Equivariant Architectures (EA). We introduce the notions of weakly and strongly invariant laws (WI and SI) on the parameter space of each single unit, corresponding, respectively, to $G$-invariant distributions, and to distributions supported on parameters fixed by the group action (which encode EA). This allows us to define symmetric models compatible with taking $N \to \infty$ and give an interpretation of the asymptotic dynamics of DA, FA and EA in terms of Wasserstein Gradient Flows describing their MF limits. When activations respect the group action, we show that, for symmetric data, DA, FA and freely-trained models obey the exact same MF dynamic, which stays in the space of WI parameter laws and attains therein the population risk's minimizer. We also provide a counterexample to the general attainability of such an optimum over SI laws. Despite this, and quite remarkably, we show that the space of SI laws is also preserved by these MF distributional dynamics even when freely trained. This sharply contrasts the finite-$N$ setting, in which EAs are generally not preserved by unconstrained SGD. We illustrate the validity of our findings as $N$ gets larger, in a teacher-student experimental setting, training a student NN to learn from a WI, SI or arbitrary teacher model through various SL schemes. We lastly deduce a data-driven heuristic to discover the largest subspace of parameters supporting SI distributions for a problem, that could be used for designing EA with minimal generalization error.

## 1 Introduction

Learning in complex tasks, employing ever larger datasets, has strongly benefited from the implementation and training of Artificial Neural Networks (NN) with a huge number of parameters; as well as from training schemes or architectures that can leverage underlying symmetries of the data in order to reduce the problem's complexity (see [33, 34] for general reference). This raises questions, on one hand, of understanding the puzzling generalizability in overparametrized NN; and on the other, of when and how symmetry-leveraging (SL) techniques (such as Data Augmentation, Feature Averaging or Equivariant Architectures), can induce useful biases towards learning with symmetries, without hindering approximation and generalization properties. The recent Mean-Field (**MF**) theory of NN (see [16] and further references below) provides a partial, yet promissory, viewpoint to address the first question for shallow NN: in the Mean-Field *Limit* (**MFL**) of an infinitely wide hidden layer, stochastic gradient descent (SGD) training dynamics approximates the Wasserstein Gradient Flow

38th Conference on Neural Information Processing Systems (NeurIPS 2024).

(**WGF**) of certain *convex population risk* on the space of distributions on parameters. Confluently, the incorporation of combined algebraic and probabilistic viewpoints have yielded a more complete view of the benefits of SL techniques under symmetry (see e.g. [13, 27, 59] and further references below); however, it is not clear if and how those findings can scale to overparametrized NN and their **MFL**.

In this work we develop a systematic **MF** analysis of the limiting learning dynamics of a class of generalized shallow NNs, under distributional symmetries of the data w.r.t. the action of a compact group, and including the possible effects of employing some of the most popular SL techniques. The effect of symmetries on the **WGF** dynamics was already studied in [35], in the particular case of two-layer ReLU networks, under data generated by a function symmetric w.r.t. a single orthogonal transformation. We consider our (independent [1]) work to largely broaden the scope and applicability of such initial contributions, as it provides a unified **MF** interpretation for both the use of SL techniques under general distributional invariances, and the interplay of such symmetries at the levels of data, architectures and training dynamics. The paper unfolds as follows:

In Section 2 we introduce a class of generalized shallow models with multi-layer units on which we will focus, we recall **WGF**s and their role in the **MFL** of NN training dynamics, and review the SL techniques to be studied. Section 3 contains the bulk of our contributions, as we study how SL techniques applied on these models can be interpreted in terms of their limiting **WGF**s, how they relate to each other in terms of the optima of their corresponding population risks, and how their limiting **MF** training dynamics behave with or without symmetric data. Finally, Section 4 presents the empirical *validation* of our main theoretical results through some numerical simulations; it also suggests a potential heuristic for discovering data-driven *parameter-sharing* schemes that lead to *optimal* equivariant architectures in ML problems. Proofs and complements to our results can be found in the Supplementary Material (henceforth SuppMat for short), together with a discussion of the scope and limitations of our results, as well as a summary of the notation and abbreviations used.

## 2 Preliminaries

### 2.1 Supervised learning with generalized shallow neural networks

Let $\mathcal{X}$, $\mathcal{Y}$ and $\mathcal{Z}$ be separable Hilbert Spaces, termed as the *feature*, *label* and *parameter* spaces respectively. Typically, these are finite-dimensional, e.g. $\mathcal{X} = \mathbb{R}^d$ and $\mathcal{Y} = \mathbb{R}^c$ (for $c, d \in \mathbb{N}^*$) with $\mathcal{Z}$ the space of affine transformations between hidden layers. We write $\mathcal{P}(\cdot)$ for the space of Borel probability measure on a metric space $(\cdot)$. Let $\pi \in \mathcal{P}(\mathcal{X} \times \mathcal{Y})$ denote the data distribution from which i.i.d. samples $(X, Y) \in \mathcal{X} \times \mathcal{Y}$ will be drawn, and $\ell : \mathcal{Y} \times \mathcal{Y} \to \mathbb{R}$ be a **convex** loss function. Consider also an *activation function* $\sigma_* : \mathcal{X} \times \mathcal{Z} \to \mathcal{Y}$. We introduce a general class of shallow NN:

**Definition 1.** *A shallow neural network model of parameter* $\theta := (\theta_i)_{i=1}^N \in \mathcal{Z}^N$ *is the function* $\Phi_\theta^N : \mathcal{X} \to \mathcal{Y}$ *given by* $\Phi_\theta^N(x) := \frac{1}{N} \sum_{i=1}^N \sigma_*(x; \theta_i)$, $\forall x \in \mathcal{X}$. *Equivalently, if* $\nu_\theta^N := \frac{1}{N} \sum_{i=1}^N \delta_{\theta_i}$ *is the empirical measure associated with* $\theta \in \mathcal{Z}^N$, *we can write* $\forall x \in \mathcal{X}$, $\Phi_\theta^N(x) = \langle \sigma_*(x; \cdot), \nu_\theta^N \rangle$ *or, abusing notation, simply* $\Phi_\theta^N = \langle \sigma_*, \nu_\theta^N \rangle$.

In the setting where $\mathcal{X} = \mathbb{R}^d$, $\mathcal{Y} = \mathbb{R}^c$ and $\mathcal{Z} = \mathbb{R}^{c \times b} \times \mathbb{R}^{d \times b} \times \mathbb{R}^b$ (for $b \in \mathbb{N}^*$), if we consider, for $z = (W, A, B) \in \mathcal{Z}$ and $\sigma : \mathbb{R}^b \to \mathbb{R}^b$, $\sigma_*(x, z) := W\sigma(A^T x + B)$; then $\Phi_\theta^N$ (with $N \in \mathbb{N}$, $\theta \in \mathcal{Z}^N$) corresponds exactly to a single-hidden-layer neural network with $N$ hidden units. Depending on $\sigma_*$, however, these shallow NN models can represent settings that go far beyond this first example. In fact, $\sigma_*$ can be taken to be an entire Multi-Layer NN model, in which case $\Phi_\theta^N$ will represent an **ensemble** of $N$ such units trained simultaneously (see SuppMat-C.1). As we will also shortly see, for suitable subspaces of $\mathcal{Z}$, this modelling extends to renowned equivariant architectures such as CNNs, DeepSets and GNNs. Beyond NNs, this setting can also model the deconvolution of sparse spikes, RBF networks, density estimation via MMD minimization, among many others (see [16, 62, 69]).

This class thus allows for non-trivial internal units, while enabling the width $N \to \infty$ consistently, and regardless of the possible underlying structure of the (fixed size) units represented by $\sigma_*$. Inspired by this possibility, and by our writing of *shallow NN models*, we define a more general notion:

**Definition 2** (**Shallow Model**). *A shallow model is any function of the form* $\Phi_\mu(x) := \langle \sigma_*(x; \cdot), \mu \rangle$ *for some* $\mu \in \mathcal{P}(\mathcal{Z})$ *(whenever the integral makes sense for all* $x \in \mathcal{X}$). *We write* $\Phi_\mu := \langle \sigma_*, \mu \rangle$ *and denote the space of such models as* $\mathcal{F}_{\sigma_*}(\mathcal{P}(\mathcal{Z}))$.

---

[1] We became aware of the work [35] after a first version of this paper was posted.

Classically, we want to find a NN model that performs well with respect to $\pi$ and $\ell$. More precisely, having fixed an *architecture* (given here by $N$ and $\sigma_*$), we consider the *generalization error* or *population risk* given by $R(\theta) = \mathbb{E}_\pi \left[ \ell(\Phi_\theta^N(X), Y) \right]$, and look for a vector of parameters $\theta \in \mathcal{Z}^N$ attaining $\min_{\theta \in \mathcal{Z}^N} R(\theta)$. However, not only is this function highly non-convex and hard to optimize; but in practice we generally don't have access to $\pi$ (and thus $R$) and we have to solve this problem only with a set of i.i.d. data samples $\{(X_k, Y_k)\}_{k \in \mathbb{N}}$ drawn from $\pi$. Thus, the usual approach to minimizing this population risk is to *train* a *NN model* $\Phi_\theta^N$, through an SGD scheme (see e.g. [7]):

- First, initialize $\theta_i^0, \forall i \in \{1, \ldots, N\}$, i.i.d. from a fixed distribution $\mu_0 \in \mathcal{P}(\mathcal{Z})$.
- Iterate, for $k \in \mathbb{N}$, defining $\forall i \in \{1, \ldots, N\}$:

$$\theta_i^{k+1} = \theta_i^k - s_k^N \left( \nabla_z \sigma_*(X_k, \theta_i^k) \cdot \nabla_1 \ell(\Phi_{\theta^k}^N(X_k), Y_k) + \tau \nabla r(\theta_i^k) \right) + \sqrt{2\beta s_k^N} \xi_i^k. \quad (1)$$

Here, $s_k^N = \varepsilon_N \varsigma(k\varepsilon_N)$ is the *step-size* (or *learning rate*), parametrized in terms of $\varsigma : \mathbb{R}_+ \to \mathbb{R}_+$ a regular function and $\varepsilon_N > 0$. Also, we have a penalization function $r : \mathcal{Z} \to \mathbb{R}$, regularizing Gaussian noise $\xi_i^k \overset{i.i.d.}{\sim} \mathcal{N}(0, \mathrm{Id}_\mathcal{Z})$ independent from the initialization and data, and $\tau, \beta \geq 0$. When $\tau, \beta > 0$, the method is called stochastic gradient Langevin dynamics, noted SGLD ([74]), or simply *noisy* SGD. An *infinite i.i.d.* sample from $\pi$ will be needed when letting later $N \to \infty$. When $\pi$ is the empirical measure of a finite dataset, we are performing empirical-risk minimization (which of course is not the same as minimizing generalization error, but follows the same mathematical formulation).

In principle, there are no guarantees that this training procedure will be truly optimizing $R(\theta)$ let alone approaching its minimum. However, by extending the definition of the **generalization error** to models in $\mathcal{F}_{\sigma_*}(\mathcal{P}(\mathcal{Z}))$, one gets the convex functional $R : \mathcal{P}(\mathcal{Z}) \to \mathbb{R}$ given by $R(\mu) := \mathbb{E}_\pi \left[ \ell(\Phi_\mu(X), Y) \right]$. The problem on $\mathcal{Z}^N$ is thus lifted to the **convex** optimization problem on $\mathcal{P}(\mathcal{Z})$:

$$\min_{\mu \in \mathcal{P}(\mathcal{Z})} R(\mu). \quad (2)$$

Accordingly, this motivates looking at the evolution of empirical measures $(\nu_k^N)_{k \in \mathbb{N}} := (\nu_{\theta^k}^N)_{k \in \mathbb{N}} \subseteq \mathcal{P}(\mathcal{Z})$ instead of that of the specific parameters $(\theta^k)_{k \in \mathbb{N}} \subseteq \mathcal{Z}^N$. The **MF** approach to NNs (see [16, 53, 62, 67]) aims at providing theoretical guarantees for problem (2), justifying that a *global optimum* of the population risk can be approximated by training a NN with SGD for large $N$. We next provide some necessary background on **WGF**s and on the **MF** theory of shallow NN models.

## 2.2 Wasserstein gradient flow and mean-field limit of shallow models

We briefly recall some elements of Optimal Transport and Wasserstein Gradient Flows, referring to [1, 63, 71] for further background. Let $\mathcal{Z}$ be a Hilbert space with norm $\| \cdot \|$ and, for $p \in [1, \infty)$, let $\mathcal{P}_p(\mathcal{Z}) := \{ \mu \in \mathcal{P}(\mathcal{Z}) : \int_\mathcal{Z} \|\theta\|^p \mu(d\theta) < +\infty \}$ be the space of probability measures on $\mathcal{Z}$ with finite $p$-th moment. We endow this space with the $p$-th *Wasserstein distance*, defined as: $W_p(\mu, \nu) := \left[ \inf_{\gamma \in \Pi(\mu, \nu)} \mathbb{E}_\gamma [\|X - Y\|^p] \right]^{\frac{1}{p}}, \forall \mu, \nu \in \mathcal{P}_p(\mathcal{Z})$ with $\Pi(\mu, \nu)$ being the set of *couplings between $\mu$ and $\nu$* (the infimum is always attained). The metric space $(\mathcal{P}_p(\mathcal{Z}), W_p)$ is Polish and called the $p$-th Wasserstein Space. In the remainder of this section we consider $p = 2$ and $\mathcal{Z} = \mathbb{R}^D$.

We recall central objects for the sequel, including Lions' derivative [9, 47], popularized in mean-field games (see e.g. [10, 12, 15, 38]) and shown (in [32]) to coincide with the Wasserstein gradient ([1]):

**Definition 3** (**Linear Functional Derivative and Intrinsic Derivative**). *Given $F : \mathcal{P}_2(\mathcal{Z}) \to \overline{\mathbb{R}}$, its linear functional derivative is the function (if it exists) $\frac{\partial F}{\partial \mu} : Dom(F) \times \mathcal{Z} \to \mathbb{R}$ such that $\forall \mu, \nu \in Dom(F)$, $\lim_{h \to 0} \frac{F((1-h)\mu + h\nu) - F(\mu)}{h} = \int_\mathcal{Z} \frac{\partial F}{\partial \mu}(\mu, z) d(\nu - \mu)(z)$ and $\int_\mathcal{Z} \frac{\partial F}{\partial \mu}(\mu, z) d\mu(z) = 0$. The function $F' : \mu \in \mathcal{P}_2(\mathcal{Z}) \mapsto \frac{\partial F}{\partial \mu}(\mu, \cdot)$ is known as the first variation of $F$ at $\mu$. Moreover, if $\frac{\partial F}{\partial \mu}$ exists and is differentiable in its second argument, we define the intrinsic derivative of $F$ at $\mu$ to be: $D_\mu F(\mu, z) = \nabla_z \left( \frac{\partial F}{\partial \mu}(\mu, z) \right)$. Abusing notation, we will write $\frac{\partial F}{\partial \mu} : \mathcal{P}_2(\mathcal{Z}) \times \mathcal{Z} \to \mathbb{R}$ and $D_\mu F : \mathcal{P}_2(\mathcal{Z}) \times \mathcal{Z} \to \mathcal{Z}$, even if they are only partially defined.*

This allows us to define next a Wasserstein Gradient Flow (following e.g. [1, 16]):

**Definition 4** (**Wasserstein Gradient Flow**). *Let $\varsigma : \mathbb{R}_+ \to \mathbb{R}_+$ be a regular scalar function and $F : \mathcal{P}_2(\mathcal{Z}) \to \overline{\mathbb{R}}$ be a **convex** functional for which the intrinsic derivative $D_\mu F$ is defined. We define a **Wasserstein Gradient Flow (WGF) for** $F$ (shortened **WGF(F)**) as any absolutely continuous trajectory $(\mu_t)_{t \geq 0}$ in $\mathcal{P}_2(\mathcal{Z})$ that satisfies, distributionally on $[0, \infty) \times \mathcal{Z}$ :*

$$\partial_t \mu_t = \varsigma(t) \operatorname{div}(D_\mu F(\mu_t, \cdot) \mu_t). \tag{3}$$

Several authors ([1, 16, 63, 71], among others) have proven under various sets of assumptions that, given an initial condition $\mu_0 \in \mathcal{P}_2(\mathcal{Z})$, the **WGF**$(F)$ admits a unique (weak) solution, $(\mu_t)_{t \geq 0}$. In a sense, **WGF**$(F)$ *'follows the negative gradient'* of $F$. Unfortunately, even for convex $F$, stationary points of **WGF**$(F)$ need not be global minima, see [16].

We are interested in the case where $F$ is the following convex, **entropy-regularized population risk**: $R^{\tau,\beta}(\mu) := R(\mu) + \tau \int r d\mu + \beta H_\lambda(\mu)$, where $\tau, \beta \geq 0$, $\lambda$ is the Lebesgue Measure on $\mathcal{Z}$, $r : \mathcal{Z} \to \mathbb{R}_+$ is a *penalization*, and $H_\lambda$ defined as $H_\lambda(\mu) := \int \log(\frac{d\mu}{d\lambda}(z)) d\mu(z)$ if $\mu \lll \lambda$ or $+\infty$ otherwise, is the *Boltzmann entropy* of $\mu$. In this case, **WGF**$(R^{\tau,\beta})$ reads as the PDE:

$$\partial_t \mu_t = \varsigma(t) \left[ \operatorname{div}((D_\mu R(\mu_t, \cdot) + \tau \nabla_\theta r) \mu_t) + \beta \Delta \mu_t \right], \tag{4}$$

known as McKean-Vlasov equation in the probability and PDE communities (see the classic references [54, 70], and the recent review [11]) and popularized as 'distributional dynamics' in NN literature (e.g. [53]). When $\beta > 0$, a solution to (4) has a density w.r.t. $\lambda$ and is actually strong. Under rather simple technical assumptions (see SuppMat-D.3, or [12, 15, 38, 57, 69]), when $\tau, \beta > 0$ it is known that the **WGF**$(R^{\tau,\beta})$ $W_2$-converges to a (unique) minimizer. When $\tau, \beta = 0$ a sort of converse holds (see [16]): if **WGF**$(R)$ converges in $W_2$, then the limit minimizes $R$.

Proven by [16, 53, 62, 67] and later refined e.g. by [14, 22, 23, 51, 66, 69], the main result in the **MF** Theory of overparametrized shallow NNs states that SGD *training* for a shallow NN, in the right *scaling* limit as $N \to \infty$, approximates a **WGF** :

**Theorem 1** (**Mean-Field limit, sketch**). *For each $T > 0$, under relevant technical assumptions including regularity of $\sigma_*$ and a proper asymptotic behaviour of $\varepsilon_N \to 0$ as $N \to \infty$, the rescaled empirical process given by $\mu^N := (\nu^N_{\lfloor t/\varepsilon_N \rfloor})_{t \in [0,T]}$ converges in law (in the Skorokhod space $D_{\mathcal{P}(\mathcal{Z})}([0,T])$) to $\mu := (\mu_t)_{t \in [0,T]}$ given by the **unique WGF**$(R^{\tau,\beta})$ starting at $\mu_0$.*

Despite the **MF** limit of NNs being a theoretical approximation, the behavior it predicts can effectively be observed in practice, even for finite, not too large $N$ (see the numerical experiments in many of the aforementioned works and below). Moreover, it is the asymptotic regime that most closely describes the actual feature-learning behavior observed in large, overparametrized NNs during training (as compared e.g. to the lazy-training regime described by the Neural Tangent Kernel approximation [17, 41]). Note that, for $\beta > 0$, the entropy term $H_\lambda$ in **WGF**$(R^{\tau,\beta})$ (as well as the Laplace operator in equation (4)) is approximated, in practice, by the Gaussian noise term in the SGLD (1), as $N \to \infty$.

## 2.3 Symmetry-leveraging techniques

We next discuss mathematical formulations of the main techniques to leverage posited distributional symmetries of the data at the training or architecture levels. We henceforth fix a *compact group* $G$ of normalized Haar measure $\lambda_G$, acting on $\mathcal{X}$ and $\mathcal{Y}$, which we denote $G \circlessgtr_\rho \mathcal{X}$, $G \circlessgtr_{\hat{\rho}} \mathcal{Y}$. [2] A function $f : \mathcal{X} \to \mathcal{Y}$ is termed *equivariant* if $\forall g \in G$, $\hat{\rho}_{g^{-1}}.f(\rho_g.x) = f(x)$ $d\pi_\mathcal{X}(x)$-a.s. We further say that the data $(X, Y) \sim \pi$ is *equivariant*, and write $\pi \in \mathcal{P}^G(\mathcal{X} \times \mathcal{Y})$, if $\forall g \in G$, $(\rho_g.X, \hat{\rho}_g.Y) \sim \pi$ (this is not enforced unless stated). The space of functions $f : \mathcal{X} \to \mathcal{Y}$ square-integrable (in Bochner sense) w.r.t $\pi_\mathcal{X} = Law(X)$ is called $L^2(\mathcal{X}, \mathcal{Y}; \pi_\mathcal{X})$. Further relevant concepts are introduced as needed.

**Data Augmentation (DA):** This training scheme considers $\{g_k\}_{k \in \mathbb{N}} \overset{i.i.d.}{\sim} \lambda_G$ independent from the $\{(X_k, Y_k)\}_{k \in \mathbb{N}}$ in (1), and carries out SGD on samples $\{(\rho_{g_k}.X_k, \hat{\rho}_{g_k}.Y_k)\}_{k \in \mathbb{N}}$. **DA** and the *vanilla* training scheme would thus be equivalent if $\pi \in \mathcal{P}^G(\mathcal{X} \times \mathcal{Y})$. One can show (see [13, 48]) that, performing SGD with **DA**, results in an optimization scheme for the *symmetrized population risk*, $R^{DA}(\theta) := \mathbb{E}_\pi \left[ \int_G \ell \left( \Phi^N_\theta(\rho_g.X), \hat{\rho}_g.Y \right) d\lambda_G(g) \right]$. Despite being effective in practice, **DA** gives no guarantee that the resulting model will be equivariant. For deeper insights, see [13, 21, 39, 46, 48, 52].

---

[2]w.l.o.g. by compactness, all $G-$actions are assumed to be via orthogonal representations

**Feature Averaging (FA):** Instead of focusing on the data, **FA** works with *symmetrized* versions of the *vanilla* NN models $\Phi_\theta^N$ at hand, averaging model copies over all possible translations through the group action. This amounts to constructing (or approximating) the *symmetrization operator* over $L^2(\mathcal{X}, \mathcal{Y}; \pi_\mathcal{X})$ defined as $(\mathcal{Q}_G.f)(x) := \int_G \hat{\rho}_{g^{-1}}.f(\rho_g.x)d\lambda_G(g)$ (see [27]), and trying to minimize $R^{FA}(\theta) := \mathbb{E}_\pi \left[ \ell \left( (\mathcal{Q}_G.\Phi_\theta^N)(X), Y \right) \right]$ (see [13, 21, 46, 48]). The resulting model will be equivariant, however, **FA** is inefficient, as $\approx |G|$ times more evaluations are needed for training and inference.

**Equivariant Architectures (EA):** Following [8], **EA** in multilayer NNs are configurations yielding models equivariant between each of the hidden layers (where $G$ is assumed to act). As stated in [29, 30, 61, 64, 65, 75] and [2, 18, 44, 45, 73], once the (equivariant) activation functions between the different layers have been fixed, **EA**s are plainly parameter-sharing schemes (determined by the space of *intertwiners/group convolutions* between layers). In our context, assuming that $G \circlearrowright_M \mathcal{Z}$ is some group action, we require that $\sigma_* : \mathcal{X} \times \mathcal{Z} \to \mathcal{Y}$ is *jointly equivariant*, namely, $\forall (g, x, z) \in G \times \mathcal{X} \times \mathcal{Z}$, $\sigma_*(\rho_g.x, M_g.z) = \hat{\rho}_g \sigma_*(x, z)$; to ensure $G$-actions over different spaces are properly related. Introducing the set of fixed points for $G \circlearrowright_M \mathcal{Z}$, $\mathcal{E}^G := \{z \in \mathcal{Z} \; : \; \forall g \in G, \; M_g.z = z\}$, a shallow NN model $\Phi_\theta^N$ thus has an **EA** if $\theta \in (\mathcal{E}^G)^N$. Under the right choices of $\sigma_*$ and $M$, the obtained **EA**s can encode interesting and widely applied architectures, such as CNNs [19] and DeepSets [77] (see SuppMat-C.1 for further discussion). We call $\mathcal{E}^G$ the **subspace of invariant parameters**, which is a closed linear subspace of $\mathcal{Z}$, with unique *orthogonal projection* $P_{\mathcal{E}^G} : \mathcal{Z} \to \mathcal{E}^G$, explicitly given by $P_{\mathcal{E}^G}.z := \int_G M_g.z \, d\lambda_G(g)$ for $z \in \mathcal{Z}$ (see [30]). We are thus led to solve: $\min_{\theta \in (\mathcal{E}^G)^N} R(\theta)$ or, equivalently, to find the best *projected* model $\Phi_\theta^{N,EA} := \langle \sigma_*, P_{\mathcal{E}^G} \# \nu_\theta^N \rangle$, by minimizing $R^{EA}(\theta) := \mathbb{E}_\pi \left[ \ell \left( \Phi_\theta^{N,EA}(X), Y \right) \right]$. This can considerably reduce the parameter space dimension; however **EA** might generally yield a decreased expressivity or approximation capacity.

# 3 Symmetries in overparametrized neural networks: main results

## 3.1 Two notions of symmetries for parameter distributions

The following notions regarding distributions from $\mathcal{P}(\mathcal{Z})$ are central to our work:

**Definition 5.** *Given $\mu \in \mathcal{P}(\mathcal{Z})$, we respectively define its **symmetrized** and **projected** versions as $\mu^G := \int_G (M_g \# \mu) d\lambda_G$ and $\mu^{\mathcal{E}^G} := P_{\mathcal{E}^G} \# \mu$. Moreover, we introduce two subspaces of $\mathcal{P}(\mathcal{Z})$: $\mathcal{P}^G(\mathcal{Z}) := \{\mu \in \mathcal{P}(\mathcal{Z}) \; : \; \forall g \in G, \; M_g \# \mu = \mu\}$ and $\mathcal{P}(\mathcal{E}^G) := \{\mu \in \mathcal{P}(\mathcal{Z}) \; : \; \mu(\mathcal{E}^G) = 1\}$.*

**Example.** *For $G = \{\pm 1\}$ acting multiplicatively on $\mathcal{Z} = \mathbb{R}$, one has $\mathcal{E}^G = \{0\}$, hence $\mathcal{P}(\mathcal{E}^G) = \{\delta_0\}$, while $\mathcal{P}^G(\mathcal{Z}) = \{\frac{1}{2}(\nu + \nu(-\cdot)) : \nu \in \mathcal{P}(\mathbb{R}_+)\}$. In particular, for $z \in \mathcal{Z}$, $(\delta_z)^G = \frac{1}{2}(\delta_z + \delta_{-z})$.*

**Definition 6 (Invariant Probability Measures).** *We say that $\mu \in \mathcal{P}(\mathcal{Z})$ is:*

$$\textit{Weakly-Invariant (WI) if } \mu = \mu^G \quad \text{and} \quad \textit{Strongly-Invariant (SI) if } \mu = \mu^{\mathcal{E}^G}.$$

We notice that: $\mathcal{P}(\mathcal{E}^G) \subseteq \mathcal{P}^G(\mathcal{Z})$, $\mu^G \in \mathcal{P}^G(\mathcal{Z})$ and $\mu^{\mathcal{E}^G} \in \mathcal{P}(\mathcal{E}^G)$. Thus, **SI** implies **WI**. Next result relates the symmetrization operation on $\mathcal{P}(\mathcal{Z})$ with the one on shallow models $\mathcal{F}_{\sigma_*}(\mathcal{P}(\mathcal{Z}))$:

**Proposition 1.** *Let $\Phi_\mu \in \mathcal{F}_{\sigma_*}(\mathcal{P}(\mathcal{Z}))$ with $\sigma_* : \mathcal{X} \times \mathcal{Z} \to \mathcal{Y}$ jointly equivariant. Then:*

$$(\mathcal{Q}_G \Phi_\mu) = \Phi_{\mu^G}.$$

*That is to say, the **closest equivariant function** (in $L^2(\mathcal{X}, \mathcal{Y}; \pi_\mathcal{X})$) to $\Phi_\mu$ is given by the shallow model associated to the **symmetrized** version of $\mu$.*

**Remark.** *In particular, $\Phi_\mu$ is equivariant as soon as $\mu$ is **WI** only. Conversely, if $\Phi_\mu : \mathcal{X} \to \mathcal{Y}$ is an equivariant function, then $\Phi_\mu = \Phi_{\mu^G}$, i.e. it can be expressed in terms of a **WI** distribution. This highlights a priority role of **WI** distributions on $\mathcal{Z}$ in representing invariant shallow models.*

The alternative, 'projected model' $\Phi_{\mu^{\mathcal{E}^G}}$, in turn, is never the closest equivariant shallow model, to $\Phi_\mu$ in $L^2(\mathcal{X}, \mathcal{Y}, \pi_\mathcal{X})$, unless equal to $\Phi_{\mu^G}$. The latter rarely is the case (unlike commonly implied in the literature). In fact, the symmetrized version $\mathcal{Q}_G \Phi_\theta^N$ of a shallow NN model $\Phi_\theta^N$ involves $(\nu_\theta^N)^G = \frac{1}{N} \sum_{i=1}^N \varphi_{\theta_i}$, where $\forall z \in \mathcal{Z}$, $\varphi_z$ is the orbit measure of the action,[3] and has $N \cdot |G|$

---

[3]defined as: $\varphi_z = T_z \# \lambda_G$, where $T_z := [g \in G \mapsto M_g.z \in \mathcal{Z}]$

$\mathcal{Z}$-valued parameters (possibly with $|G| = \infty$). This sharply contrasts $(\nu_\theta^N)^{\mathcal{E}^G} = \frac{1}{N} \sum_{i=1}^N \delta_{P_{\mathcal{E}^G}.\theta_i}$, which has $\leq N$ distinct parameters, all living in $\mathcal{E}^G$. So, in general, depending on $\sigma_*$ and $G$, one might have $\langle \sigma_*, (\nu_\theta^N)^{\mathcal{E}^G} \rangle \neq \mathcal{Q}_G \Phi_\theta^N$. A notable case in which the equality holds is the class of linear models, which is discussed in SuppMat-E.1.

**Example.** *In the previous example, for $\mu = \delta_z$ and $\sigma_*$ jointly equivariant, we have $\Phi_\mu = \sigma_*(\cdot, z)$, $\Phi_{\mu^G} = \mathcal{Q}_G \Phi_\mu = \frac{1}{2}(\sigma_*(\cdot, z) + \sigma_*(\cdot, -z))$ and $\Phi_{\mu^{\mathcal{E}^G}} = \sigma_*(\cdot, 0)$ which are generally distinct if $z \neq 0$. Notice that $\Phi_{\mu^G}$ is an equivariant function without any of its 'parameters' living in $\mathcal{E}^G$.*

### 3.2 Invariant functionals on $\mathcal{P}(\mathcal{Z})$ and their optima

In the same spirit as when defining the population risk $R : \mathcal{P}(\mathcal{Z}) \to \overline{\mathbb{R}}$ in (2), the risk functions associated with SL-techniques from Section 2.3 can be lifted to functionals over $\mathcal{P}(\mathcal{Z})$, namely to: $R^{DA}(\mu) := \mathbb{E}_\pi \left[ \int_G \ell \left( \Phi_\mu(\rho_g.X), \hat{\rho}_g.Y \right) d\lambda_G(g) \right]$, $R^{FA}(\mu) := \mathbb{E}_\pi \left[ \ell \left( \mathcal{Q}_G(\Phi_\mu)(X), Y \right) \right]$ and $R^{EA}(\mu) := \mathbb{E}_\pi \left[ \ell \left( \Phi_{\mu^{\mathcal{E}^G}}(X), Y \right) \right]$, respectively. This will allow us to study these SL-techniques, in the overparametrized regime, under a common **MF** framework. We need the following assumption:

**Assumption 1.** *$\pi \in \mathcal{P}_2(\mathcal{X} \times \mathcal{Y})$; $\ell : \mathcal{Y} \times \mathcal{Y} \to \mathbb{R}$ is convex, jointly invariant and differentiable with $\nabla_1 \ell$ linearly growing; and $\sigma_* : \mathcal{X} \times \mathcal{Z} \to \mathcal{Y}$ is bounded, jointly equivariant and differentiable.*

The *quadratic loss* $\ell(y, \hat{y}) = \frac{1}{2} ||y - \hat{y}||^2$ is an example of such $\ell$. Having $\sigma_*$ bounded and differentiable is a simplifying assumption, usually made in the **MF** literature, when establishing key results such as *global convergence of NN* (see e.g. [12, 38, 53]); relaxing this condition to include further commonly-used functions $\sigma_*$ seems feasible, up to some additional technicalities (see SuppMat-A.2 for further discussion). Finally, having $\sigma_*$ be jointly equivariant (as defined in section 2.3) isn't a truly restrictive assumption: under the right choice of $\sigma_*$ and $M$, any usual single-hidden-layer NN architecture can be made to satisfy it (see SuppMat-C.1 for a deeper discussion). We also need:

**Definition 7.** *A functional $F : \mathcal{P}(\mathcal{Z}) \to \mathbb{R}$ is invariant if $F(M_g \# \mu) = F(\mu) \, \forall (g, \mu) \in G \times \mathcal{P}(\mathcal{Z})$; equivalently, if it equals its symmetrized version $F^G(\mu) := \int_G F(M_g \# \mu) d\lambda_G(g)$.*

**Proposition 2.** *Under Assumption 1, $R^{DA}$, $R^{FA}$ and $R^{EA}$ are invariant (and convex) and we have:*

$$R^{DA}(\mu) = R^G(\mu), \quad R^{FA}(\mu) = R(\mu^G) \quad and \quad R^{EA}(\mu) = R(\mu^{\mathcal{E}^G}).$$

*In particular, $R = R^{DA}$ if $R$ is invariant. Moreover, $\forall \mu \in \mathcal{P}^G(\mathcal{Z})$, $R(\mu) = R^{DA}(\mu) = R^{FA}(\mu)$. Last, if $\pi \in \mathcal{P}^G(\mathcal{X} \times \mathcal{Y})$ (the data distribution is equivariant), then $R$ is invariant.*

The proof relies on Proposition 1 and calculations as in [30], see SuppMat-E.2. Next result is a general property of functionals over $\mathcal{P}(\mathcal{Z})$, which is key for the forthcoming analysis:

**Proposition 3** (**Optimality for Invariant Functionals**)**.** *Let $F : \mathcal{P}(\mathcal{Z}) \to \overline{\mathbb{R}}$ be convex, $\mathcal{C}^1$ and invariant. Then: $\forall \mu \in \mathcal{P}(\mathcal{Z})$, $F(\mu^G) \leq F(\mu)$; and so, $\inf_{\mu \in \mathcal{P}^G(\mathcal{Z})} F(\mu) = \inf_{\mu \in \mathcal{P}(\mathcal{Z})} F(\mu)$. In particular, if $F$ has a unique minimizer over $\mathcal{P}(\mathcal{Z})$, it **must be WI**.*

The proof relies on an ad-hoc version of Jensen's inequality. Next, we state that optimizing under **DA** and **FA** is essentially equivalent, and corresponds to optimizing $R$ exclusively over **WI** measures:

**Theorem 2** (**Equivalence of DA and FA**)**.** *Under assumption 1, we have:*

$$\inf_{\mu \in \mathcal{P}(\mathcal{Z})} R^{DA}(\mu) = \inf_{\mu \in \mathcal{P}^G(\mathcal{Z})} R^{DA}(\mu) = \inf_{\mu \in \mathcal{P}^G(\mathcal{Z})} R(\mu) = \inf_{\mu \in \mathcal{P}^G(\mathcal{Z})} R^{FA}(\mu) = \inf_{\mu \in \mathcal{P}(\mathcal{Z})} R^{FA}(\mu).$$

Note that, on the other hand, $R^{EA}$ only satisfies: $\inf_{\mu \in \mathcal{P}(\mathcal{Z})} R^{EA}(\mu) = \inf_{\mu \in \mathcal{P}(\mathcal{E}^G)} R(\mu)$. In the case of the quadratic loss, Theorem 2 can be made more explicit:

**Corollary 1.** *Under Assumption 1, when the loss is quadratic and $\pi_{\mathcal{X}}$ is invariant, we have:*

$$\inf_{\mu \in \mathcal{P}^G(\mathcal{Z})} R(\mu) = R_* + \inf_{\mu \in \mathcal{P}^G(\mathcal{Z})} \| \Phi_\mu - f_* \|^2_{L^2(\mathcal{X}, \mathcal{Y}; \pi_{\mathcal{X}})} = \tilde{R}_* + \inf_{\mu \in \mathcal{P}^G(\mathcal{Z})} \| \Phi_\mu - \mathcal{Q}_G . f_* \|^2_{L^2(\mathcal{X}, \mathcal{Y}; \pi_{\mathcal{X}})}.$$

*where $f_* = \mathbb{E}_\pi[Y|X = \cdot]$, and $R_*$, $\tilde{R}_*$ are constants only depending on $\pi$ and $f_*$. That is, optimizing under **DA** and **FA** corresponds to approximating the **symmetrized** version of $f_*$.*

Under equivariance of the data distribution $\pi$, the following general result also holds:

**Corollary 2.** *Let Assumption 1 hold and suppose $\pi \in \mathcal{P}^G(\mathcal{X} \times \mathcal{Y})$. Then, $R$ is invariant and therefore:* $\inf_{\mu \in \mathcal{P}(\mathcal{Z})} R(\mu) = \inf_{\mu \in \mathcal{P}^G(\mathcal{Z})} R(\mu) = \inf_{\mu \in \mathcal{P}(\mathcal{Z})} R^{DA}(\mu) = \inf_{\mu \in \mathcal{P}(\mathcal{Z})} R^{FA}(\mu)$.

**Remark.** *Consequently, equivariant data allow us to globally optimize the population risk by only considering **WI** measures. It also shows that **DA** and **FA** provide no advantage for this optimization.*

The same unfortunately is not true for **SI** measures (answering a question in [27]), as shown by the following result, which constructs a simple example in which $\mathcal{E}^G$ is trivial:

**Proposition 4.** *Even with a finite group $G$ acting orthogonally on $\mathcal{X} = \mathbb{R}^d$, $\mathcal{Y} = \mathbb{R}$ and $\mathcal{Z} = \mathbb{R}^{(d+2)}$; with $\pi$ being compactly-supported and equivariant; with $\ell$ being quadratic; and with $\sigma_*$ being $\mathcal{C}^\infty$, bounded and jointly equivariant; we can have:* $\inf_{\mu \in \mathcal{P}(\mathcal{Z})} R(\mu) < \inf_{\nu \in \mathcal{P}(\mathcal{E}^G)} R(\nu)$.

In fact, even if $R$ is invariant, when $\mathcal{E}^G$ is *too restrictive*, it might become impossible to globally optimize $R$ over **SI** measures (which amounts to using $R^{EA}$ as a proxy for $R$). This subtlety has to be considered when deciding to use **EA**s on problems where symmetries exist. Nevertheless, if $\mathcal{E}^G$ has good *universality* properties, a true **SI** solution to the learning problem can be sought for:

**Proposition 5.** *Let Assumption 1 hold, $\ell$ be quadratic and $\pi \in \mathcal{P}_2^G(\mathcal{X} \times \mathcal{Y})$. If $\mathcal{F}_{\sigma_*}(\mathcal{P}(\mathcal{E}^G))$ is dense in $L_G^2(\mathcal{X}, \mathcal{Y}; \pi_{\mathcal{X}}) := \mathcal{Q}_G(L^2(\mathcal{X}, \mathcal{Y}; \pi_{\mathcal{X}}))$, then:* $\inf_{\mu \in \mathcal{P}(\mathcal{Z})} R(\mu) = \inf_{\nu \in \mathcal{P}(\mathcal{E}^G)} R(\nu) = R_*$.

**Remark.** *See e.g. [50, 60, 76, 77] for conditions on $\mathcal{E}^G$ and $\sigma_*$ guaranteeing this 'restricted' universality on $L_G^2(\mathcal{X}, \mathcal{Y}; \pi_{\mathcal{X}})$. These allow for effectively solving the problem in fewer dimensions, which is key in successful **EA** like CNNs and DeepSets. See SuppMat-E.2.5 for a deeper discussion.*

## 3.3 Symmetries and SL training dynamics in the overparametrized regime

We now study the **MFL** of the various training dynamics when $\mathcal{Z} = \mathbb{R}^D$. We begin with the general:

**Theorem 3.** *Let $F : \mathcal{P}(\mathcal{Z}) \to \overline{\mathbb{R}}$ be an invariant functional such that **WGF**$(F)$ is well defined and has a unique (weak) solution $(\mu_t)_{t \geq 0}$. If $\mu_0 \in \mathcal{P}_2^G(\mathcal{Z})$, then, for dt-a.e. $t \geq 0$ we have $\mu_t \in \mathcal{P}_2^G(\mathcal{Z})$.*

The proof of Theorem 3 relies on $D_\mu F$ being equivariant (in a suitable sense) and $(M_g \mu_t)_{t \geq 0}$ satisfying also, as a consequence, **WGF**$(F)$ (See SuppMat-E.3 for the details). Note that $\mu_0 \in \mathcal{P}_2^G(\mathcal{Z})$ is simply verified, e.g. by a standard Gaussian in $\mathcal{Z}$. Specializing this result, we get:

**Corollary 3.** *Let Assumption 1 and technical assumptions (as in [16]) hold. Then, if $R$ and $r$ are invariant, **WGF**$(R^{\tau,\beta})$ starting from $\mu_0 \in \mathcal{P}_2^G(\mathcal{Z})$ satisfies: for dt-a.e. $t \geq 0$, $\mu_t \in \mathcal{P}_2^G(\mathcal{Z})$. If moreover $\beta > 0$, each $\mu_t$ has a density function invariant with respect to $G \circlearrowright_M \mathcal{Z}$.*

**Remark.** *If $\pi$ is equivariant, $R$ is invariant, and this result is valid for a **freely-trained NN, without employing SL-techniques**. In a way, **MFL** incorporates these symmetries from infinite SGD iterations.*

Theorem 3 and Corollary 3 can thus be seen as significant generalizations of Proposition 2.1 from [35], which addresses the case of wide 2-layer ReLU networks with a target function that's symmetric w.r.t. a single orthogonal transformation. The fact that strong solutions to **WGF**$(R^{\tau,\beta})$ can be sought among invariant functions to reduce the complexity when $\pi$ is equivariant, was also first hinted in [53]. The natural domain of invariant functions is in fact the quotient space of $G \circlearrowright_M \mathcal{Z}$ (and not $\mathcal{E}^G$, which is strictly embedded in it).

Comparing the different training dynamics at the **MF** level and applying Proposition 2, we also get:

**Theorem 4.** *Under assumptions of Corollary 3, if $\mu_0 \in \mathcal{P}_2^G(\mathcal{Z})$, **WGF**$(R^{DA})$ and **WGF**$(R^{FA})$ solutions are equal. If further $R$ is invariant, the **WGF**$(R)$ solution coincides with them too.*

**Remark.** *In particular, with equivariant data (i.e. invariant $R$), training with **DA** or **FA** is essentially the same, at least at the **MF** level, as **using no SL-technique whatsoever**. Hence, a relevant, practical open question, is: how do the convergence rates to the **MFL** compare in all three cases, as $N \to \infty$?*

We will now see that similar results hold for $\mathcal{P}(\mathcal{E}^G)$ instead of $\mathcal{P}^G(\mathcal{Z})$. Notice that the *entropy-regularized* risk forces each $\mu_t$ to have a density w.r.t. $\lambda$ in $\mathcal{Z}$ if $\beta > 0$. Therefore, if $G \circlearrowright_M \mathcal{Z}$ is non-trivial (thus $\mathcal{E}^G$ is a strict subspace), we always have $\mu_t \notin \mathcal{P}(\mathcal{E}^G)$. It thus seems natural to *restrain* the noise in equation (1) to stay in $\mathcal{E}^G$; namely, to consider the *projected noisy SGD* dynamic:

$$\theta_i^{k+1} = \theta_i^k - s_k^N \left( \nabla_z \sigma_*(X_k, \theta_i^k) \cdot \nabla_1 \ell(\Phi_{\theta^k}^N(X_k), Y_k) + \tau \nabla r(\theta_i^k) \right) + \sqrt{2\beta s_k^N} P_{\mathcal{E}^G} \xi_i^k. \quad (5)$$

Note that projecting *only* the noise in (5) doesn't force $\theta_i^{k+1}$ to be in $\mathcal{E}^G$, even if $\theta_i^k$ was. Introducing the related **projected-regularized functional**: $R_{\mathcal{E}^G}^{\tau,\beta}(\mu) := R(\mu) + \tau \int r d\mu + \beta H_{\lambda_{\mathcal{E}^G}}(\mu^{\mathcal{E}^G})$, with $\lambda_{\mathcal{E}^G}$ the Lebesgue Measure over $\mathcal{E}^G$, we get the following analogue of Corollary 3:

**Theorem 5.** *Let Assumption 1 and technical assumptions on $R$ hold. Suppose that $R$ and $r$ are invariant. Then, if $\mu_0 \in \mathcal{P}_2(\mathcal{E}^G)$, $(\mu_t)_{t \geq 0}$ solution of $\mathbf{WGF}(R_{\mathcal{E}^G}^{\tau,\beta})$ satisfies $\forall t \geq 0$, $\mu_t \in \mathcal{P}_2(\mathcal{E}^G)$.*

The result holds for $\beta \geq 0$. Its proof is based on pathwise properties of the McKean-Vlasov stochastic differential equation (studied e.g. in [22]) associated with the $\mathbf{WGF}(R_{\mathcal{E}^G}^{\tau,\beta})$, see SuppMat-D.2.

**Remark.** *Theorem 5 is significantly stronger than Corollary 3: it implies that, for equivariant $\pi$, the flow will remain in the set of **SI** distributions all throughout its evolution, despite there being **no explicit constraint on the network parameters during training** (they can all be freely updated), **nor any SL-technique** being used. This is a highly non-intuitive fact, and a large $N$ exclusive phenomenon, as our numerical experiments will show. See SuppMat-D.2 for a deeper discussion.*

**Remark.** *Notice that, despite the computation of $\mathcal{E}^G$ being generally hard (see [29]), $\mu_0 \in \mathcal{P}_2(\mathcal{E}^G)$ can be achieved by simply setting $\mu_0 = \delta_{\overline{0}}$. Moreover, since one can also take $\beta = 0$, 'having access' to the noise projection $P_{\mathcal{E}^G}$ is never explicitly required, allowing for a broader applicability of the result. In particular, as we'll show in our experiments, usual shallow NNs with all parameters initialized at $\{0\}$, freely trained with 'noiseless' SGD, will satisfy Theorem 5 in the **MFL**.*

**Remark.** *Theorem 5 holds too for the invariant functionals $R^{DA}$, $R^{FA}$ and $R^{EA}$ in the role of $R$, **even if $\pi$ is not equivariant**. Notably, **DA**, **FA** and **EA** procedures starting on a **SI** distribution, despite being free to involve all parameters, will keep the distribution **SI** all throughout training.*

Last, we also have:

**Theorem 6.** *Let the conditions for Theorem 5 hold. If $\mu_0 \in \mathcal{P}_2(\mathcal{E}^G)$, the $\mathbf{WGF}(R^{FA})$, $\mathbf{WGF}(R^{DA})$ and $\mathbf{WGF}(R^{EA})$ solutions **coincide**. If $R$ is invariant, $\mathbf{WGF}(R)$ solution coincides with them too.*

## 4 Numerical experiments and architecture-discovery heuristic

To empirically illustrate some of our results from the previous section, we consider synthetic data produced in a **teacher-student** setting (see e.g. [14, 16]). Code necessary for replicating the obtained results, as well as a detailed description of our experimental setting, can be sought in the SuppMat.

We study a simple setting with: $\mathcal{X} = \mathcal{Y} = \mathbb{R}^2$, $\mathcal{Z} = \mathbb{R}^{2 \times 2} \cong \mathbb{R}^4$, and $G = C_2$ acting on $\mathcal{X}$ and $\mathcal{Y}$ by *permuting the coordinates*; and on $\mathcal{Z}$ via the natural *intertwining* action (for which $\mathcal{E}^G$ is explicit). We take the jointly equivariant activation $\sigma_*(x, z) = \sigma(z \cdot x)$, $\forall (x, z) \in \mathcal{X} \times \mathcal{Z}$ with $\sigma : \mathbb{R} \to \mathbb{R}$ a sigmoidal function applied pointwise; and consider *normally* distributed features, and labels produced from a **teacher model** $f_*$. This teacher is given by a shallow NN model, either $f_* = \Phi_{\theta^*}^{N_*}$ with $N_* = 5$ **arbitrary** particles $\theta^* \in \mathcal{Z}^{N_*}$, or its symmetrized version $f_* = \mathcal{Q}_G.\Phi_{\theta^*}^{N_*}$ (referred to as **WI**), with 10 particles. [4] Notice that the data distribution $\pi$ will be equivariant only if the teacher is. We try to **mimic** such **teacher** with a **student model**, $\Phi_\theta^N$, with the same $\sigma_*$, but different particles $\theta \in \mathcal{Z}^N$ that will be trained to minimize the regularized population risk $R^{\tau,\beta}$ (with quadratic loss and penalization). For this we employ the SGD dynamic given by Equation (1) (or projected, if required, as in Equation (5)), possibly involving **DA**, **FA** or **EA** techniques. We refer to *free training*, with no SL-techniques involved, as **vanilla** training. Each experiment was repeated $N_r = 10$ times to ensure consistency. Explicit values of the fixed training parameters are found in SuppMat-F.

### 4.1 Study for varying $N$

We demonstrate how properties of $\mathbf{WGF}(R^{\tau,\beta})$ stated in Section 3.3 become apparent as $N$ grows. We consider a teacher with $\nu_{\theta^*}^{N_*}$ either **arbitrary** or **WI**, and different training schemes performed over $N_e$ epochs, all initialized with the same particles drawn from given $\mu_0 \in \mathcal{P}_2(\mathcal{Z})$ that is either **SI** or **WI**. Figure 1 displays the behavior of the student's particle distribution, $\nu_{N_e}^N$, after training, in terms of certain 'normalized version' of the $W_2$-distance, which we call simply Relative Measure Distance, or **RMD** for short. [5] We refer to SuppMat-F for further insights and, additionally: a deeper

---

[4]An additional variant, with $f_* = \Phi_{\theta^*}^{N_*}$ having **SI** particles, is given in SuppMat-F.

[5]It is roughly equivalent to $W_2$ when far from the $\delta_0$ measure, see SuppMat-F for details

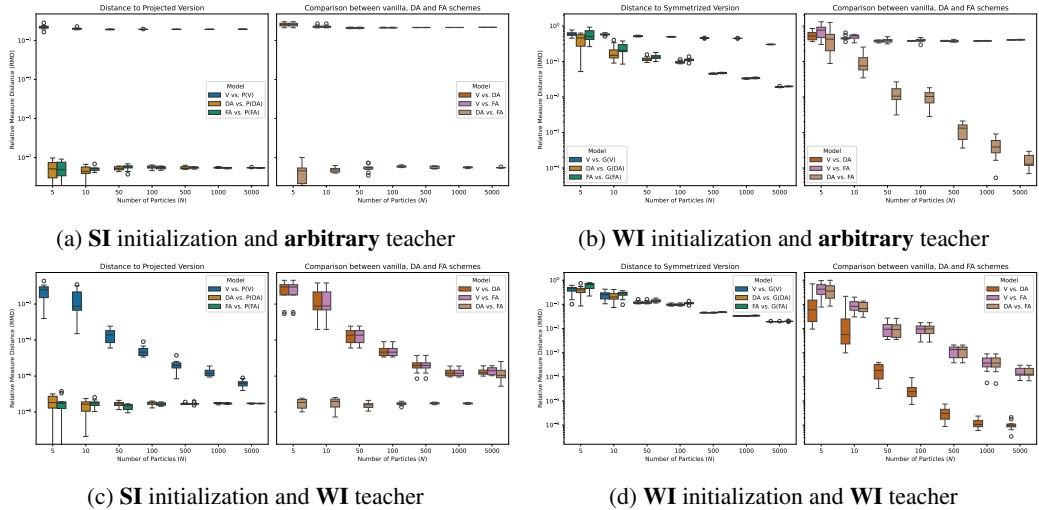

(a) **SI** initialization and **arbitrary** teacher      (b) **WI** initialization and **arbitrary** teacher

(c) **SI** initialization and **WI** teacher      (d) **WI** initialization and **WI** teacher

Figure 1: **RMD**s, at the end of training, for $N = 5, 10, 50, 100, 500, 1000, 5000$ and the **vanilla (V)**, **DA**, **FA** and **EA** schemes. The first plot of each position displays either $\mathbf{RMD}^2(\nu_{N_e}^N, (\nu_{N_e}^N)^{\mathcal{E}^G})$ or $\mathbf{RMD}^2(\nu_{N_e}^N, (\nu_{N_e}^N)^G)$ depending on initialization (either **SI** or **WI**); and the second shows the **RMD** between **DA**, **FA** and **vanilla** schemes.

analysis of the case of $\nu_{\theta^*}^{N_*}$ being **SI**, comparisons between different techniques and **EA**, and $L^2$ comparisons between $\Phi_\theta^N$ and both $f_*$ and $\mathcal{Q}_G.f_*$ (to illustrate Corollary 1).

We first look at the **SI**-initialized training. Though from [30] we know that (exact) **DA** or **FA** during training will *respect* $\mathcal{E}^G$ without needing to pass to the **MFL**. This is certainly not true in general for the **vanilla** scheme, where the symmetry is never *explicit* for the model. We notice in Figure 1, however, that, as $N$ grows big, the **SI**-initialized **vanilla** training scheme, under only a **WI** teacher, does **remain SI** throughout training, as predicted in Theorem 5. This is absolutely remarkable, since there is no intuitive reason why the **vanilla** scheme (were parameters can be updated *freely*) *shouldn't escape* $\mathcal{E}^G$ to better approximate $f_*$. For instance, for an **arbitrary teacher** (with the same particles, but *un-symmetrized*) **vanilla** training readily *leaves* $\mathcal{E}^G$ to better approximate $f_*$. Though this isn't a *predicted behaviour* from our theory, it motivates a heuristic we present in the upcoming section. On the other hand, and as expected, both **DA** and **FA** consistently remain within $\mathcal{E}^G$ almost independently of $N$, and even if $f_*$ isn't equivariant. Finally, as $N$ grows bigger, the end-of-training distribution of the **vanilla** scheme *approaches* that of **DA** and **FA** (as expected from Theorem 4).

Regarding the **WI**-initialized training, unlike the **SI** case, particles sampled *i.i.d.* from a **WI** distribution don't necessarily yield a **WI** empirical distribution $\nu_0^N$. On the one hand, this means we require large $N$ to see $\nu_{N_e}^N$ being (approx.) **WI**; and on the other hand, it means we have no guarantee that **DA** and **FA** will be close unless we look at larger $N$ (where Theorem 4 applies). The second column of Figure 1 precisely shows these behaviours as $N$ grows: both a trend of $\nu_{N_e}^N$ towards becoming **WI**, and a clear coincidence between the **DA**, **FA** and **vanilla** schemes (the latter only for equivariant $f_*$).

### 4.2 Heuristic algorithm for discovering EA parameter spaces

From these experiments, for non-equivariant $f_*$, the **SI**-initialized WGF is seen to eventually **escape** $\mathcal{E}^G$. In turn, for equivariant $f_*$, Figure 2 shows that a training scheme initialized at $E \subsetneq \mathcal{E}^G$ (i.e. $\nu_{\theta_0}^N \in \mathcal{P}(E)$), eventually **leaves** $E$, but **stays within** $\mathcal{E}^G$ (as expected from Theorem 5). These empirical observations hint to an heuristic for **discovering the 'good' EAs for the task at hand**.

Assume indeed $\pi$ equivariant w.r.t. $G$. We want to find the unknown, largest (i.e. most expressive) subspace of $\mathcal{Z}$ supporting **SI** measures. We hence consider some (large) $N$, a shallow NN model with e.g. $\sigma_*(x, z) = \sigma(z.x)$, and numerical thresholds $(\delta_j)_{j \in \mathbb{N}} \subseteq \mathbb{R}_+$. We define $E_0 = \{0\} \leq \mathcal{E}^G$ as an initial subspace and initialize $\nu_{\theta_0}^N = \nu_{\bar{0}}^N \in \mathcal{P}(E_0)$. Then, we iteratively proceed as follows:

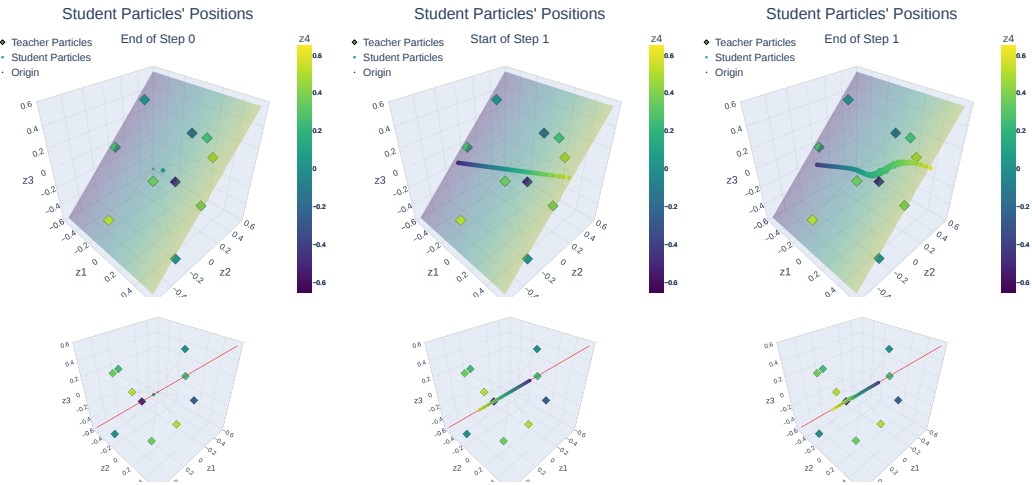

Figure 2: Heuristic method applied on **teacher** (squares) and **student** (dots) particles. *Row 1*: aerial view of hyperplane $\mathcal{E}^G$. *Row 2*: *parallel* view, to verify student particles always remain in $\mathcal{E}^G$ (red line). *Column 1*: step $j = 0$ after training; particles **leave** $E_0 = \{0\}$. *Column 2*: initialization of step $j = 1$ on $E_1 = \langle v_{E_0} \rangle$. *Column 3*: step $j = 1$ after training; particles **leave** $E_1$ (Row 1), but not $\mathcal{E}^G$.

For $j = 0, 1, \ldots$, initialize a model at $\nu_{\theta_0}^N \in \mathcal{P}(E_j)$, train it for $N_e$ epochs, and check whether $\mathbf{RMD}^2(\nu_{N_e}^N, P_{E_j} \# \nu_{N_e}^N) \leq \delta_j$. If that is the case, the training didn't **escape** $E_j$, and one could suppose $\mathcal{E}^G := E_j$. Otherwise, it **left** $E_j$ (so $\mathcal{E}^G \neq E_j$) and one can set e.g. $E_{j+1} := E_j \oplus v_{E_j}$, with $v_{E_j} := \frac{1}{N} \sum_{i=1}^{N} (\theta_i^{N_e} - P_{E_j} . \theta_i^{N_e})$. Allegedly, this scheme would eventually leave all strict subspaces to reach the 'right' $\mathcal{E}^G$. Figure 2 indeed illustrates this behaviour in our simple **teacher-student** example (see SuppMat-F.2 for further details). Notice that we start knowing close to nothing about data symmetries ($E_0 = \{0\}$), and end up *'discovering'* a data-based parameter-sharing scheme ($E_* = \mathcal{E}^G$) that allows for building **SI** NNs. This idea might have potential for real world applications, yet a larger scale experimental analysis and rigorous theoretical guarantees need to be provided.

We refer to [72] for a different approach to this idea of 'discovering the real symmetries of the data'. Their work uses *relaxed* group convolution layers to discover 'data-driven symmetry-breaking' in ML problems. A deeper comparison between both approaches shall be found in SuppMat-F.2.

## 5   Conclusion

In the light of theoretical guarantees given by the **MF** theory of overparametrized shallow NN, we explored their training dynamics when data is possibly equivariant for some group action and/or SL techniques are employed. We thus described how **DA**, **FA** and **EA** schemes can be understood in the limit of infinite internal units, and studied in that setting the qualitative advantages that can be attired from their use. In this **MFL**, **DA** and **FA** are essentially equivalent in terms of the optimization problem they solve and the trajectory of their associated **WGF**s. Moreover, for equivariant data, *freely*-trained NN, in the **MFL**, obey the same **WGF** as **DA/FA**. They also "respect" symmetries during training, as **WI** and **SI** initializations (corresponding to symmetric parameter distributions and **EA** configurations) are preserved throughout, even if potentially all NN parameters can be updated. We also highlighted the prominent role of **WI** laws in representing equivariant models. We illustrated our results with appropriate numerical experiments, which in turn suggested a data-driven heuristic to find appropriate parameter subspaces for **EA**s in a given task. Providing theoretical guarantees for this heuristic is an interesting problem left for future work. A further relevant question to address, is to quantify and compare the speeds at which all studied training schemes approach the **MFL**, as this would a provide a full comparative picture of their performances. Extending the **MF** analysis of symmetries to NNs with more complex inner structures is another interesting line of work.

## Acknowledgements

JM thanks partial financial support from ANID Magister Fellowship 22231325 and CMM-Basal Grant FB210005, ANID-Chile. JF thanks partial financial support from Fondecyt Regular Grants 1201948/1242001 and CMM-Basal Grant FB210005, ANID-Chile . Both authors thank Roberto Cortez, Daniel Remenik and Felipe Tobar for comments and discussions which motivated part of the numerical experiments developed in the paper, as well as the refinement of some ideas. The authors also thank Daniela Bellott for her assistance in running part of the numerical experiments.

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

Table 1: Summary the main abbreviations employed throughout the paper

| Abbreviation | Meaning |
| --- | --- |
| NN / NNs | Neural Network / Neural Networks |
| SL | Symmetry-Leveraging |
| **MF** | Mean Field |
| **MFL** | Mean Field Limit |
| **WGF** | Wasserstein Gradient Flow |
| SGD | Stochastic Gradient Descent |
| **DA** | Data Augmentation |
| **FA** | Feature Averaging |
| **EA** | Equivariant Architectures |
| **WI** | Weakly-Invariant |
| **SI** | Strongly-Invariant |

# A    General Considerations for the Reader

This section presents a summary of recurrent notation, abbreviations and key concepts used in our work, as well as a discussion on its limitations, scope and possible extensions. We thank anonymous Reviewers for suggesting us to add this section to the original manuscript.

## A.1    Summary of recurrent notation, abbreviations and key concepts

In this section we summarize the main abbreviations and notation employed throughout the body of the paper, as well as simple definitions of fundamental concepts from probability theory and algebra that might be useful for understanding our work

Table 1 serves as a glossary for the most used abbreviations in the body of the paper. Table 2 contains a summary of the notation that we recurrently use in our definitions, statements and proofs. It also provides some simple references to mathematical concepts that are key in our work.

For clarity, beyond the contents of both tables, we here also explain a few relevant concepts to the unfamiliar reader:

- **Bochner Integrals:** These correspond to the right generalization of Lebesgue integrals for vector-valued functions (see [24] for further reference).

  Say we have a function $f : \mathcal{X} \rightarrow \mathcal{Y}$ between Hilbert spaces, and such that $\pi_{\mathcal{X}}$ is some measure on $\mathcal{X}$, then we say $f$ is square-integrable (in Bochner sense) if it satisfies: $\int_{\mathcal{X}} \|f(x)\|_{\mathcal{Y}}^2 d\pi_{\mathcal{X}}(x) < \infty$.

  This integral also respects **closed** linear operators, as shown by Hille's theorem (see Theorem II.2.6 in [24]). Namely, if $L : \mathcal{Y} \rightarrow \mathcal{Y}$ is a **closed** linear operator over $\mathcal{Y}$, we have: $\int_{\mathcal{X}} L.f d\pi_{\mathcal{X}} = L. \int_{\mathcal{X}} f d\pi_{\mathcal{X}}$. In particular, this also holds for *bounded* linear operators.

  In general, most of the basic and most intuitive properties of traditional integrals can also be expressed for Bochner integrals.

- **Compact Groups, Group Actions and the Haar measure:** We recurrently talk about *group actions via orthogonal representations* throughout our work, so a due clarification may be required. We assume $G$ to be a **compact** group. Namely, $G$ has a topology that makes it compact, and so that the multiplication and inversion operations are **continuous**. We say that $G$ acts on $\mathcal{Z}$, which we denote $G \circlearrowright \mathcal{Z}$, whenever there exists a map:

$$T : G \times \mathcal{Z} \rightarrow \mathcal{Z}$$
$$(g, z) \mapsto T(g, z)$$

  that satisfies $T(e_G, z) = z$ and $T(g_1, T(g_2, z)) = T(g_1.g_2, z), \ \forall g_1, g_2 \in G, \ \forall z \in \mathcal{Z}$. We always assume the actions to be **continuous**; namely, $T$ is **continuous** with respect to the product topology. Alternatively, by denoting $T_g := T(g, \cdot)$, $T_. : g \in G \mapsto T_g \in \text{Sym}(\mathcal{Z})$ is a **group homomorphism** and, if the action is continuous, $T_g$ is an homeomorphism $\forall g \in G$. If we further assume that $\forall g \in G, \ T_g$ is linear, we call $T_.$ a **group representation**. Further, if $\{T_g\}_{g \in G}$ are **orthogonal** transformations, we call $T_.$ an ***orthogonal* group representation**

Table 2: Summary the notation employed throughout the paper

| Notation | Represented Object |
| --- | --- |
| $\mathcal{X}, \mathcal{Y}, \mathcal{Z}$ | Respectively, feature, label, and parameter spaces (separable and Hilbert). Usually just $\mathcal{X} = \mathbb{R}^d$, $\mathcal{Y} = \mathbb{R}^c$ and $\mathcal{Z} = \mathbb{R}^D$ for $c, d, D \in \mathbb{N}^*$. |
| $\mathcal{P}(\star)$ | Space of Borel probability measures on the metric space $(\star)$ |
| $\pi, \pi_{\mathcal{X}}$ | Data distribution in $\mathcal{P}(\mathcal{X} \times \mathcal{Y})$ and its marginal on $\mathcal{X}$. |
| $\ell$ | **Convex** loss function $\ell : \mathcal{Y} \times \mathcal{Y} \to \mathbb{R}$ |
| $\sigma_*$ | *Activation* function or *unit*, $\sigma_* : \mathcal{X} \times \mathcal{Z} \to \mathcal{Y}$. Not necessarily corresponds to the *usual* activation function from traditional NN implementations. |
| $\Phi_\theta^N$ | Shallow neural network model with $N$ units and parameter $\theta \in \mathcal{Z}^N$ |
| $\delta.$ | Dirac measure on point $\cdot$. Recall that, for a function $\varphi$, integrating gives $\langle \varphi, \delta. \rangle = \varphi(\cdot)$. |
| $\nu_\theta^N$ | Empirical measure of $\theta$ in $\mathcal{P}(\mathcal{Z})$. i.e. $\nu_\theta^N = \frac{1}{N} \sum_{i=1}^N \delta_{\theta_i}$ |
| $\Phi_\mu$ | Shallow model with associated measure $\mu \in \mathcal{P}(\mathcal{Z})$. Also denoted as $\langle \sigma_*, \mu \rangle$ |
| $R(\theta), R(\mu)$ | Generalization error or Population Risk. Abusing notation, it either represents $\mathbb{E}_\pi \left[ \ell(\Phi_\theta^N(X), Y) \right]$ for $\theta \in \mathcal{Z}^N$ or $\mathbb{E}_\pi \left[ \ell(\Phi_\mu(X), Y) \right]$ for $\mu \in \mathcal{P}(\mathcal{Z})$ |
| $\mathcal{F}_{\sigma_*}(\mathcal{P}(\mathcal{Z}))$ | Space of shallow models with unit $\sigma_*$ and measures in $\mathcal{P}(\mathcal{Z})$ |
| $\mathcal{P}_p(\mathcal{Z})$ | Probability measures $\mu \in \mathcal{P}(\mathcal{Z})$ with finite $p$-th moment: $\int_{\mathcal{Z}} \|\theta\|^p \mu(d\theta) < +\infty$. |
| $W_p$ | $p$-th Wasserstein distance, defined as $W_p(\mu, \nu) := \left[ \inf_{\gamma \in \Pi(\mu, \nu)} \mathbb{E}_\gamma[\|X - Y\|^p] \right]^{\frac{1}{p}}$, $\forall \mu, \nu \in \mathcal{P}_p(\mathcal{Z})$, where $\Pi(\mu, \nu)$ is the set of *couplings between $\mu$ and $\nu$*. |
| $\frac{\partial F}{\partial \mu}$ | Linear Functional Derivative of $F : \mathcal{P}_2(\mathcal{Z}) \to \overline{\mathbb{R}}$ as in Definition 3 |
| $D_\mu F$ | Intrinsic Derivative of $F : \mathcal{P}_2(\mathcal{Z}) \to \overline{\mathbb{R}}$, as in Definition 3 |
| $\mathbf{WGF}(F)$ | Wasserstein Gradient Flow for $F : \mathcal{P}_2(\mathcal{Z}) \to \overline{\mathbb{R}}$, as in Definition 4 |
| $r$ | Some penalization function $r : \mathcal{Z} \to \mathbb{R}_+$ |
| $\lambda$ | Lebesgue Measure on $\mathcal{Z}$ |
| $\mu \lll \lambda$ | $\mu \in \mathcal{P}(\mathcal{Z})$ is absolutely continuous w.r.t. $\lambda$. |
| $H_\lambda$ | Boltzmann entropy of $\mu$; $H_\lambda(\mu) := \int \log(\frac{d\mu}{d\lambda}(z)) d\mu(z)$ if $\mu \lll \lambda$ or $+\infty$ otherwise. |
| $R^{\tau, \beta}$ | Penalized population risk, given by $R^{\tau, \beta}(\mu) = R(\mu) + \tau \int r d\mu + \beta H_\lambda(\mu)$, with $\tau, \beta \geq 0$ the *regularization parameters*. |
| $L^2(\mathcal{X}, \mathcal{Y}; \pi_{\mathcal{X}})$ | Space of functions $f : \mathcal{X} \to \mathcal{Y}$ square-integrable (in Bochner sense) w.r.t $\pi_{\mathcal{X}}$ |
| $s_k^N$ | *Step-size* (or *learning rate*) used during training. Parametrized as $s_k^N = \varepsilon_N \varsigma(k \varepsilon_N)$, with a regular function $\varsigma : \mathbb{R}_+ \to \mathbb{R}_+$ and some $\varepsilon_N > 0$; with $k, N \in \mathbb{N}$. |
| $G, \lambda_G$ | **Compact** group and its normalized Haar measure, which is the unique right and left invariant measure w.r.t. the group multiplication, see [43]. |
| $\rho, \hat{\rho}, M$ | Orthogonal representations of the action of $G$ over $\mathcal{X}, \mathcal{Y}$ and $\mathcal{Z}$ respectively. We also denote these actions as $G \circlearrowright_\rho \mathcal{X}$, $G \circlearrowright_{\hat{\rho}} \mathcal{Y}$ and $G \circlearrowright_M \mathcal{Z}$ respectively. |
| $\mathcal{Q}_G$ | *Symmetrization operator* over $L^2(\mathcal{X}, \mathcal{Y}; \pi_{\mathcal{X}})$; $(\mathcal{Q}_G.f)(x) := \int_G \hat{\rho}_{g^{-1}}.f(\rho_g.x) d\lambda_G(g)$ for any $f \in L^2(\mathcal{X}, \mathcal{Y}; \pi_{\mathcal{X}})$. |
| $\mathcal{E}^G$ | Space of fixed points for $G \circlearrowright_M \mathcal{Z}$ (i.e. $z \in \mathcal{Z}$ s.t. $\forall g \in G$ $M_g.z = z$) |
| $P_{\mathcal{E}^G}$ | Orthogonal projection from $\mathcal{Z}$ onto $\mathcal{E}^G$; $P_{\mathcal{E}^G}.z := \int_G M_g.z \, d\lambda_G(g)$ for all $z \in \mathcal{Z}$. |
| $f \# \mu$ | Pushforward of measure $\mu \in \mathcal{P}(\mathcal{Z})$ via $f : \mathcal{Z} \to \tilde{\mathcal{Z}}$, given by $f \# \mu(\cdot) = \mu(f^{-1}(\cdot))$ |
| $\mu^G, \mu^{\mathcal{E}^G}$ | **Symmetrized** and **projected** versions of $\mu \in \mathcal{P}(\mathcal{Z})$: $\mu^G := \int_G (M_g \# \mu) d\lambda_G$ and $\mu^{\mathcal{E}^G} := P_{\mathcal{E}^G} \# \mu$. |
| $\mathcal{P}^G(\star)$ | Space of $G$-invariant measures over space $\star$. e.g. for $G \circlearrowright_M \mathcal{Z}$, the space of all $\mu \in \mathcal{P}(\mathcal{Z})$ s.t. $M_g \# \mu = \mu$ for all $g \in G$. |
| $\mathcal{P}(\mathcal{E}^G)$ | Space of measures concentrated on $\mathcal{E}^G$, i.e. $\mu \in \mathcal{P}(\mathcal{Z})$ s.t. $\mu(\mathcal{E}^G) = 1$. |
| $R^{DA}, R^{FA}, R^{EA}$ | Population risks that are optimized when applying different SL techniques; **DA**, **FA** and **EA** respectively. |
| $L_G^2(\mathcal{X}, \mathcal{Y}; \pi_{\mathcal{X}})$ | Space of functions $f : \mathcal{X} \to \mathcal{Y}$ square-integrable (in Bochner sense) w.r.t $\pi_{\mathcal{X}}$ that are also $G$-equivariant. In other words, shorthand for $\mathcal{Q}_G(L^2(\mathcal{X}, \mathcal{Y}; \pi|_{\mathcal{X}}))$ |
| $dt$-a.e. | Almost everywhere w.r.t. the Lebesgue measure $dt$ on $\mathbb{R}_+$ |
| $\lambda_{\mathcal{E}^G}$ | Shorthand for the Lebesgue Measure over $\mathcal{E}^G$. |
| $R_{\mathcal{E}^G}^{\tau, \beta}$ | Projected-regularized functional given by $R_{\mathcal{E}^G}^{\tau, \beta}(\mu) := R(\mu) + \tau \int r d\mu + \beta H_{\lambda_{\mathcal{E}^G}}(\mu^{\mathcal{E}^G})$ |
| **RMD** | Relative-Measure-Distance; $\mathbf{RMD}^2(\mu, \nu) = \frac{W_2^2(\mu, \nu)}{M_\mu^2 + M_\nu^2}$ where $M_\mu^2 = 2 \int_{\mathcal{Z}} \|z\|^2 d\mu(z)$. |
| $D_{\mathcal{P}(\mathcal{Z})}([0, T])$ | Skorokhod space of càdlàg functions from $[0, T]$ to $\mathcal{P}(\mathcal{Z})$. |

and denote the group action by $G \circlearrowright_T \mathcal{Z}$. This is the case for all of the $G$-actions considered throughout this work.

Working only with group representations is common-practice in the context of symmetries for NNs (see e.g. [13, 48]). Beyond NNs, a whole field of mathematics is of course dedicated to the study of group representations (see e.g. [36] for reference). In this work we borrow some of this theory's terminology, as we refer to **equivariant linear maps** also as **intertwining maps** or *intertwiners*. This is also why we refer to the layer-by-layer action in **EA**s for shallow NNs as an *intertwining action* (see e.g. SuppMat-C.1).

Finally, it is well known that a compact group $G$ admits a **unique normalized Haar measure** $\lambda_G \in \mathcal{P}(G)$ (see [25]), which is **left** and **right** invariant, finite on every compact set, outer regular on Borel sets and inner regular on open sets. It can be interpreted as the *uniform distribution* on $G$, and it is extensively used throughout this work.

For further references in the topic, the curious reader might be interested in [25, 42, 43], among many others.

- **Weak convergence of measures:** We also recall one of the most used notions of convergence in a space of probability measures. Given a sequence $(\mu_n)_{n \in \mathbb{N}} \subseteq \mathcal{P}(\mathcal{Z})$, we say it *weakly* converges to some point $\mu \in \mathcal{P}(\mathcal{Z})$ if, for any continuous and bounded function $f : \mathcal{Z} \to \mathbb{R}$, we have $\langle f, \mu_n \rangle \xrightarrow[n \to \infty]{} \langle f, \mu \rangle$. Notice that this type of convergence is *weaker* than $W_p$ convergence for $p \geq 1$ (which additionally also requires the convergence of $p$-th order moments of the involved measures).

  Notice how we write $\langle f, \mu \rangle$ to denote $\int_{\mathcal{Z}} f \, d\mu$. This notation is heavily used (and abused) throughout the core of our work.

## A.2 Limitations, scope and possible extensions of our work

For convenience of the reader we here present a discussion of some limitations and of the scope of application of the present work. This subsection does not provide any mathematical results required for the sequel.

Some of these limitations are of technical nature, and regard specific assumptions made in order for the specific proofs of our theoretical results to hold. In absence of these conditions, or under less restrictive ones, some of our results (or weaker forms of them) might still hold true, but further research is needed in order to properly establish their validity. Other limitations are the object of more general research questions in this area.

- **On the infinite i.i.d. data sample:** The **MF** theory of NN makes the assumption that it is possible to take an infinite i.i.d. sample from the data distribution $\pi$. This may appear as a limitation of our results, since real-world datasets are naturally finite. We acknowledge that this is in fact an abstraction, but it can nevertheless be a potentially good approximation of the behavior of the SGD algorithm when large datasets are available. Indeed, Theorem 1, which holds for $\varepsilon_N = o(1/N)$, ensures that a sample size of the order of $tN$ is required in order to approximate the **WGF** up to a ('macroscopic') time $t$. When the long-time convergence of the **WGF** can be effectively quantified (an active research question today, see e.g. [57] and references therein), we can estimate how large a data sample will be required in order to attain a predetermined generalization error level. On the other hand, even for $\pi$ with finite support, the **MF** approximation will end up minimizing the empirical-risk. This, as pointed out in Section 2.1, is not the same as minimizing generalization error w.r.t. some underlying data distribution, but it is widely interpreted (in the application of most NN-based machine learning algorithms) as a proxy of doing so. Once again, this approximation can be reasonably good when the data sample size is large enough.

- **On the infinite width of NN models:** Despite the **MFL** being a theoretical tool (which requires that the width $N$ goes to $+\infty$), it is still the asymptotic regime that most closely describes the feature-learning behavior of large NNs during training (see the discussion at the end of Section 2.2). We believe that truly useful insights can be obtained from it for real, moderately wide NNs. For instance, our experiments show that, in quite reasonable practical settings of shallow NNs (with standard pointwise sigmoid activation and objax' default SGD training), the predicted **MF** behavior is seen to emerge in practice, already for finite, not too large $N$ (1000 was generally enough). Actually, most of the insights obtained from the

**MF** analysis of NN are, in fact, unaccessible from a *fixed N* standpoint (as is the case for Theorem 5). Further non-asymptotic and quantitative answers to practical questions (e.g. how many neurons are needed in order to attain a given level of generalization/population error at given computational cost?) could also be obtained from **MF** theory, via quantitative propagation of chaos results (see e.g. [12]).

- **On Assumption 1:** This is the main assumption underlying many of our core results. Although it is generally *simple* and not excessively hard to satisfy in practice, it might seem *constraining* in the context of neural networks. In particular, the technical condition on the gradient of $\ell$, as well as the boundedness of $\sigma_*$, could seem to limit the applicability of our results. However, these conditions can be replaced by alternative properties of the data distribution (e.g. that $\pi$ has compact support, or finite moments up to a given degree). Similarly, lifting some of the rest of the technical assumptions required for establishing the **MFL**, is part of the ongoing research work in the field (see e.g. [12] or [38] for some alternative conditions).

  Regarding the assumption that $\ell$ is $G$-invariant, it is known that traditional loss functions naturally satisfy this condition. The **joint-equivariance** of $\sigma_*$, on the other hand, is much less constraining than it may initially seem. In practice, depending on the choice of $\sigma_*$ and $M$, it might even end up being a trivial constraint. A deeper discussion on this very assumption shall be found in SuppMat-C.1.

- **On the generality of shallow models** $\Phi_\mu \in \mathcal{F}_{\sigma_*}(\mathcal{P}(\mathcal{Z}))$**:** These allow for modeling a wide range of situations (including some variants of multi-layer models). In fact, $\sigma_*$ can by itself encode a complex deep architecture (see SuppMat-C.1) and the resulting shallow model can represent way more interesting structures than single-hidden-layer NNs (e.g. ensembles of such multilayer "units" trained in parallel). Nevertheless, these *shallow* models are still far from including all possibilities in the context of NNs. In fact, a fully unified, satisfactory MF theory for *general* deep NNs is still an open, actively tackled question (for advancements on it see e.g. [4, 55, 68]). We believe that some of our key results (e.g. Theorems 3 and 4) can be extended to some of those settings, which is a question we will leave for future work.

- **Universality guarantees for shallow models** $\Phi_\mu \in \mathcal{F}_{\sigma_*}(\mathcal{P}(\mathcal{Z}))$**:** In this work, we have only explored the simple setting of *tensor-of-order-1* NNs as modelled through $\sigma_*$, see SuppMat-C.1. In particular, it is known from [49] that the desired *universality on equivariant models* is not always possible with these kinds of networks (which is, of course, a limitation to the applicability of Proposition 5). Exploring other interesting situations that could be modeled with our current framework, or plainly reformulating it to account for new variants of NNs (e.g. high-order tensor NNs thay might allow for easier **EA** universality) is undoubtedly part of our future work. See SuppMat- E.2.5 for a related discussion.

- **On $G$-equivariant data distributions:** The assumption of $\pi$ being $G$-equivariant implies that $\pi_\mathcal{X}$ is $G$-invariant as well [6]. This implication can seem a bit *too restrictive* in some settings: e.g. for image classification, it amounts to assuming that *'images can arrive with any possible orientation'* at training time, which is not necessarily the case. However, as extensively discussed in the literature (see [13, 27, 30, 48]), assuming $\pi_\mathcal{X}$ to be $G$-invariant is usually reasonable when the aim is to 'exploit symmetries' of the problem. Not having such assumption means that there are little to no properties from the data that can be exploited in a proof. On the other hand, as mentioned in remarks after Theorems 3 and 5, our proofs don't explicitly rely on having $\pi \in \mathcal{P}^G(\mathcal{X} \times \mathcal{Y})$, but rather on the risk functional $R : \mathcal{P}(\mathcal{Z}) \to \mathbb{R}$ being $G$-invariant. In consequence, one could simply neglect the $G$-equivariance of $\pi$ altogether, and introduce the implied inductive bias by applying **DA** (which forces the marginal on $\mathcal{X}$ to be $G$-invariant), or any other SL technique that achieves the same result.

- **On the numerical experiments:** The suite of experiments that were presented in Section 4 and SuppMat-F, though quite insightful for our theoretical results, are quite limited in reach. In particular, finding ways to illustrate our theoretical results on *less controlled settings*, such as a real-world equivariant datasets, will be a key part of our future work.

  The presented experiments come from a *controlled* setting, mainly looking to avoid heavy practical constraints (e.g. visualizing the huge parameter space of a NN trained over an image dataset can be exceedingly hard), as well as an increased complexity of the involved objects (e.g. even for finite groups, group actions can get severely more complicated than what we experienced). Due to our currently limited computational resources, we

leave these inquiries for future work. Similarly, taking our experiments to a larger scale (e.g. $N = 100.000$) is also a challenge left for future inquiries (our current results, with $N \leq 5000$, might still be considered as *small N*). A significant problem for scaling our experimental verifications is the fact that we relied on calculating Wasserstein distances (which is usually really computationally-expensive) to provide rigorous numerical evidence of our theoretical findings.

Furthermore, we are yet to experiment with architectures that are compatible with **infinite** compact groups $G$; namely for examples coming from physics and NeuralODEs. Different variants of the activation function should also be tested out (e.g. a ReLU, tanh, and many others), as well as variants of the optimization algorithm (e.g. Adam, RMSProp, etc.).

Finally, our heuristic needs to be tested on a larger scale, with more complex datasets and architectures. Also, theoretical guarantees to sustain it shall be provided in future work. We believe that similar arguments as in [72] could be developed for our case; and alternative approaches could be based on understanding the support properties of the McKean-Vlasov SDE studied in Appendix E.3.2.

- **On possible variants of the training dynamic:** In this work we focus mostly on the 'traditional' **WGF** learning dynamics, without delving much into other interesting possible variants of the training process. This might be seen as a possible limitation to the applicability of our work.

    Firstly, the decision to work with the *usual* **WGF** follows from the standards set by [16, 53, 62, 67] among many others. Beyond this fact, we believe that results such as Theorems 3 and 4 are somewhat 'natural', and that they should hold regardless of small differences in the training dynamics.

    For instance, we know that our proofs for Theorems 3 and 4 work straightforwardly in the setting of [22], where the **MFL** is established even with a fixed learning rate that does not necessarily decrease with $N$. For large fixed learning rates, this might shed some light onto the **MF** behaviour of symmetries under 'Edge of Stability' (EoS) dynamics. Similarly, we believe some of our more 'natural' results to be applicable as well for *Wasserstein sub-Gradient Flows* [16], *Underdamped Dynamics* [31], *Annealed Dynamics* [15], among many others. In contrast, Theorems 5 and 6, which involve *stronger* notions of symmetry, don't seem immediate to generalize to many other kinds of asymptotic dynamics. Studying the applicability of our results under different variants of the training process, is surely an interesting question to be tackled as part of our future work.

## B  Symmetries in functions, measures, data and shallow models

In this section we state some useful, basic results on the effect of symmetries acting on measures, functions and data, that will be used in the sequel. We also explain how symmetries of interest can be incorporated into the generalized shallow NN setting from Definition 1, complementing also the discussions given in Section 2.1 and Section 2.3 in that regard.

Recall $\mathcal{X}, \mathcal{Y}$ and $\mathcal{Z}$ are (separable) Hilbert Spaces and $G$ a compact group with Haar measure $\lambda_G$, such that $G \circlearrowright_\rho \mathcal{X}$, $G \circlearrowright_{\hat{\rho}} \mathcal{Y}$ and $G \circlearrowright_M \mathcal{Z}$. Also, let $\mathcal{E}^G \subseteq \mathcal{Z}$ be the linear subspace of parameters that are *fixed points of the action of $G$ over $\mathcal{Z}$*, and $P_{\mathcal{E}^G}$ the orthogonal projection onto it.

### B.1  Differentials and integrals of equivariant functions

The following lemma characterizes the differential of *jointly equivariant* functions with respect to the action of some group $G$. Here we can assume $G$ be a lcsH group w.l.o.g. and consider representations that aren't necessarily orthogonal (we denote them differently to avoid confusion).

**Proposition 6.** *Let $G \circlearrowright_\chi \mathcal{X}$, $G \circlearrowright_{\tilde{\chi}} \mathcal{Z}$, $G \circlearrowright_{\check{\chi}} \mathcal{Y}$ via some representations $\chi, \tilde{\chi}$ and $\check{\chi}$ respectively (not necessarily orthogonal). Let $f : \mathcal{X} \times \mathcal{Z} \to \mathcal{Y}$ be jointly $G$-equivariant with respect to these actions (i.e. $\forall g \in G$, $\forall x \in \mathcal{X}$, $\forall z \in \mathcal{Z}$, $\check{\chi}_g.f(x,z) = f(\chi_g.x, \tilde{\chi}_g.z)$) and Fréchet-differentiable on its first argument. Then:*

$$\forall g \in G, \ \forall x \in \mathcal{X}, \ \forall z \in \mathcal{Z}, \ \ D_x f(\chi_g.x, \tilde{\chi}_g.z) = \check{\chi}_g.D_x f(x,z)\chi_g^{-1}$$

*Proof of Proposition 6.* Indeed, we know that $\forall z \in \mathcal{Z} \ D_x f(\cdot, z)$ is the unique function that satisfies, $\forall \tilde{x} \in \mathcal{X}$:

$$\lim_{h \to 0} \frac{\|f(\tilde{x} + h, z) - f(\tilde{x}) - D_x f(\tilde{x}, z)h\|}{\|h\|} = 0$$

Since we want to prove that $\forall \tilde{x} \in \mathcal{X}, \forall z \in \mathcal{Z}, \ \forall g \in G : D_x f(\chi_g.\tilde{x}, \tilde{\chi}_g.z) = \check{\chi}_g D_x f(\tilde{x}, z)\chi_g^{-1}$, it will be enough to check that:

$$\lim_{h \to 0} \frac{\|f(\chi_g.\tilde{x} + h, \tilde{\chi}_g z) - f(\chi_g.\tilde{x}, \tilde{\chi}_g.z) - \check{\chi}_g D_x f(\tilde{x}, z)\chi_g^{-1}h\|}{\|h\|} = 0$$

which by uniqueness implies the result. Thanks to the joint equivariance of $f$, we have $\forall h \neq 0$:

$$\frac{\|f(\chi_g.\tilde{x} + h, \tilde{\chi}_g z) - f(\chi_g.\tilde{x}, \tilde{\chi}_g.z) - \check{\chi}_g D_x f(\tilde{x}, z)\chi_g^{-1}h\|}{\|h\|}$$

$$= \frac{\|f(\chi_g.(\tilde{x} + \chi_g^{-1}.h), \tilde{\chi}_g z) - f(\chi_g.\tilde{x}, \tilde{\chi}_g.z) - \check{\chi}_g D_x f(\tilde{x}, z)\chi_g^{-1}h\|}{\|h\|}$$

$$= \frac{\|\check{\chi}_g.f(\tilde{x} + \chi_g^{-1}.h, z) - \check{\chi}_g.f(\tilde{x}, z) - \check{\chi}_g D_x f(\tilde{x}, z)\chi_g^{-1}h\|}{\|h\|}$$

$$= \frac{\|\check{\chi}_g.\left[f(\tilde{x} + \chi_g^{-1}.h, z) - f(\tilde{x}, z) - D_x f(\tilde{x}, z)(\chi_g^{-1}h)\right]\|}{\|\chi_g.\chi_g^{-1}.h\|}$$

Now, recall that for every $g \in G$, the operator $\check{\chi}_g$ is bounded, i.e. it has finite operator norm $0 < \|\check{\chi}_g\| < \infty$ (non-zero as $\check{\chi}_g$ is invertible). By defining $\tilde{h} := \chi_g^{-1}.h$, we have:

$$\frac{\|\check{\chi}_g.\left[f(\tilde{x} + \tilde{h}, z) - f(\tilde{x}, z) - D_x f(\tilde{x}, z)\tilde{h}\right]\|}{\|\chi_g \tilde{h}\|} \leq \frac{\|\check{\chi}_g\| \|f(\tilde{x} + \tilde{h}, z) - f(\tilde{x}, z) - D_x f(\tilde{x}, z)\tilde{h}\|}{\|\chi_g.\tilde{h}\|}$$

Multiplying by $1 = \frac{\|\chi_g^{-1}\chi_g \tilde{h}\|}{\|\tilde{h}\|}$ the last term is seen to be bounded by

$$\|\check{\chi}_g\| \|\chi_g^{-1}\| \cdot \frac{\|f(\tilde{x} + \tilde{h}, z) - f(\tilde{x}, z) - D_x f(\tilde{x}, z)\tilde{h}\|}{\|\tilde{h}\|}$$

Since $\chi_g$ and $\chi_g^{-1}$ are bounded operators, we have that: $h \to 0 \iff \tilde{h} = \chi_g^{-1}h \to 0$. Thus

$$\lim_{h \to 0} \frac{\|f(\chi_g.\tilde{x} + h, \tilde{\chi}_g z) - f(\chi_g.\tilde{x}, \tilde{\chi}_g.z) - \check{\chi}_g D_x f(\tilde{x}, z)\chi_g^{-1}h\|}{\|h\|}$$

$$\leq \lim_{h \to 0} \|\check{\chi}_g\| \|\chi_g^{-1}\| \cdot \frac{\|f(\tilde{x} + \tilde{h}, z) - f(\tilde{x}, z) - D_x f(\tilde{x}, z)\tilde{h}\|}{\|\tilde{h}\|} = 0$$

$\square$

This, in particular, allows us to characterize the differential of equivariant functions.

**Corollary 4.** *If $G \circlearrowright_\chi \mathcal{X}$, $G \circlearrowright_{\tilde{\chi}} \mathcal{Y}$, and $f : \mathcal{X} \to \mathcal{Y}$ is a G-equivariant and Fréchet-differentiable function, then:*

$$\forall g \in G, \ \forall x \in \mathcal{X}, \ \ D_x f(\chi_g.x) = \tilde{\chi}_g D_x f(x)\chi_g^T$$

*Proof of Corollary 4.* Direct. $\square$

We can also get some interesting *integral* properties of *jointly* equivariant functions.

**Proposition 7.** *Let $\mathcal{X}, \mathcal{Y}$ and $\mathcal{Z}$ be (separable) Hilbert Spaces and G be a lcsH group. Let $G \circlearrowright_\chi \mathcal{X}$, $G \circlearrowright_{\tilde{\chi}} \mathcal{Z}$, $G \circlearrowright_{\check{\chi}} \mathcal{Y}$ via some representations $\chi, \tilde{\chi}$ and $\check{\chi}$ respectively (not necessarily orthogonal).*

*Let $f : \mathcal{X} \times \mathcal{Z} \to \mathcal{Y}$ be a jointly G-equivariant function (with respect to these actions). Consider a measure $\mu \in \mathcal{P}(\mathcal{Z})$ and let $f$ be Bochner integrable on its second argument with respect to $\mu$. Then:*

$$\forall x \in \mathcal{X}, \ \forall g \in G, \ \check{\chi}_g \langle f(x; \cdot), \mu \rangle = \langle f(\chi_g x; \cdot), \tilde{\chi}_g \# \mu \rangle$$

*Proof of Proposition 7.* Let $\mu \in \mathcal{P}(\mathcal{Z})$ and $f$ be as stated. Notice that, $\forall x \in \mathcal{X}$, $\forall g \in G$, we have:

$$\check{\chi}_g \langle f(x, \cdot), \mu \rangle = \check{\chi}_g \int_{\mathcal{Z}} f(x, \theta) d\mu(\theta) = \int_{\mathcal{Z}} \check{\chi}_g f(x, \theta) d\mu(\theta),$$

where we've used the linearity of the Bochner integral under continuous linear operators (as is $\check{\chi}_g$). It follows, from the joint $G$-equivariance of $f$ and the definition of the *pushforward measure*, that:

$$\int_{\mathcal{Z}} \check{\chi}_g f(x, \theta) d\mu(\theta) = \int_{\mathcal{Z}} f(\chi_g x, \tilde{\chi}_g \theta) d\mu(\theta) = \int_{\mathcal{Z}} f(\chi_g x, \tilde{\theta}) d(\tilde{\chi}_g \# \mu)(\tilde{\theta})$$

We conclude the desired result. $\qquad\square$

## B.2 Properties of symmetric measures

Consider again compact $G$ acting orthogonally over the different spaces. Recall that, given $\mu \in \mathcal{P}(\mathcal{Z})$, we defined $\mu^G := \int_G (M_g \# \mu) d\lambda_G$ as its **symmetrized** version and $\mu^{\mathcal{E}^G} := P_{\mathcal{E}^G} \# \mu$ as its **projected** version. The following two results assumed in the core of the paper are elementary, but we provide their detailed proofs for completeness:

**Lemma 1.** *We have the following inclusion:* $\mathcal{P}(\mathcal{E}^G) \subseteq \mathcal{P}^G(\mathcal{Z})$. *Also, for any* $\mu \in \mathcal{P}(\mathcal{Z})$ *the following equalities hold* $\forall g \in G$: $\mu^{\mathcal{E}^G} = (M_g \# \mu)^{\mathcal{E}^G} = (\mu^G)^{\mathcal{E}^G} = (\mu^{\mathcal{E}^G})^G$ *and* $(M_g \# \mu)^G = \mu^G$.

*Proof of Lemma 1.* Let $\mu \in \mathcal{P}(\mathcal{E}^G)$ (i.e. $\mu(\mathcal{E}^G) = 1$), $g \in G$ and consider any positive measurable $f : \mathcal{Z} \to \mathbb{R}$. We can see:

$$\int_{\mathcal{Z}} f(M_g z) \mu(dz) = \int_{\mathcal{E}^G} f(M_g z) \mu(dz) = \int_{\mathcal{E}^G} f(z) \mu(dz) = \int_{\mathcal{Z}} f(z) \mu(dz).$$

So, that $\forall g \in G$, $\mu = M_g \# \mu$, and thus $\mu \in \mathcal{P}^G(\mathcal{Z})$. Regarding the equalities, consider $\mu \in \mathcal{P}(\mathcal{Z})$ and $A \in \mathcal{B}_{\mathcal{Z}}$ some borel set. Since the $\lambda_G$ is right-invariant, we have $\forall g \in G$, $\forall z \in \mathcal{Z}$ :

$$P_{\mathcal{E}^G} M_g z = \int_G M_h(M_g z) d\lambda_G(h) = \int_G (M_h M_g z) d\lambda_G(h) = \int_G M_{\tilde{h}} z \, d\lambda_G(\tilde{h}) = P_{\mathcal{E}^G} z.$$

Then, for $g \in G$, $M_g^{-1} P_{\mathcal{E}^G}^{-1}(A) = (P_{\mathcal{E}^G} M_g)^{-1}(A) = (P_{\mathcal{E}^G})^{-1}(A)$ and so:

$$(M_g \# \mu)^{\mathcal{E}^G}(A) = \mu(M_g^{-1} P_{\mathcal{E}^G}^{-1}(A)) = \mu(P_{\mathcal{E}^G}^{-1}(A)) = \mu^{\mathcal{E}^G}(A),$$

and:

$$(\mu^G)^{\mathcal{E}^G}(A) = \int_G \mu(M_g^{-1} P_{\mathcal{E}^G}^{-1}(A)) d\lambda_G(g) = \int_G \mu(P_{\mathcal{E}^G}^{-1}(A)) d\lambda_G(g) = \mu(P_{\mathcal{E}^G}^{-1}(A)) = \mu^{\mathcal{E}^G}(A).$$

On the other hand, since $\mu^{\mathcal{E}^G} \in \mathcal{P}^{\mathcal{E}^G}(\mathcal{Z}) \subseteq \mathcal{P}^G(\mathcal{Z})$, $(\cdot)^G$ leaves it unchanged: $(\mu^{\mathcal{E}^G})^G = \mu^{\mathcal{E}^G}$.

For the last equality, let $g \in G$ and $f : \mathcal{Z} \to \mathbb{R}_+$ be measurable. We have:

$$\langle f, (M_g \# \mu)^G \rangle = \int_G \langle f, M_h \# (M_g \# \mu) \rangle d\lambda_G(h) = \int_G \langle f, (M_{\tilde{h}}) \# \mu \rangle d\lambda_G(\tilde{h}) = \langle f, \mu^G \rangle,$$

once again by the right-invariance of $\lambda_G$. Namely, $(M_g \# \mu)^G = \mu^G$.

$\qquad\square$

**Proposition 8.** *For* $\mu \in \mathcal{P}(\mathcal{Z})$, *we have:* $\mu^G \in \mathcal{P}^G(\mathcal{Z})$ *and* $\mu^{\mathcal{E}^G} \in \mathcal{P}(\mathcal{E}^G)$.

*Proof of Proposition 8.* Indeed, let $h \in G$ and $B \in \mathcal{B}_{\mathcal{Z}}$ (borel set of $\mathcal{Z}$), using the properties of $M$ and the left-invariance of $\lambda_G$, we get that:

$$\mu^G(M_h^{-1}(B)) = \int_G \mu(M_g^{-1}(M_h^{-1}(B))) d\lambda_G(g) = \int_G \mu(M_{\tilde{g}}^{-1}(B)) d\lambda_G(\tilde{g}) = \mu^G(B)$$

So, $\forall g \in G$, $\mu^G = M_g \# \mu^G$, implying that $\mu^G \in \mathcal{P}^G(\mathcal{Z})$. On the other hand, by definition we have (as the projection is surjective) $\mu^{\mathcal{E}^G}(\mathcal{E}^G) = \mu(P_{\mathcal{E}^G}^{-1}(\mathcal{E}^G)) = \mu(\mathcal{Z}) = 1$, so that $\mu^{\mathcal{E}^G} \in \mathcal{P}(\mathcal{E}^G)$. $\qquad\square$

**Remark.** *It's not hard to notice that, on $\mathcal{Z} = \mathbb{R}^D$ and with $\lambda$ being the lebesgue measure, if $\mu \in \mathcal{P}(\mathcal{Z})$ is such that $\mu \ll \lambda$ and has density $u : \mathcal{Z} \to \mathbb{R}_+$, then: $\mu^G \in \mathcal{P}^G(\mathcal{Z})$ has density $u^G := \int_G u \circ M_g d\lambda_G(g)$ w.r.t. $\lambda$ (whereas $\mu^{\mathcal{E}^G}$ doesn't have a density w.r.t. $\lambda$ unless the action is trivial). This follows from the $O(D)$-invariance of $\lambda$ and some standard calculations.*

Since we will be working on $\mathcal{P}_2(\mathcal{Z})$ on Section 3.3, it's useful to also notice that:

**Remark.** *If $\mu \in \mathcal{P}_p(\mathcal{Z})$, then $\mu^G, \mu^{\mathcal{E}^G} \in \mathcal{P}_p(\mathcal{Z})$. Indeed, it follows from the fact that $\|M_g\| \leq 1$ $\forall g \in G$ (since the representation is orthogonal) and $\|P_{\mathcal{E}^G}\| \leq 1$ (since $P_{\mathcal{E}^G}$ is a projection).*

Also, we have that:

**Proposition 9.** *$\mathcal{P}_p(\mathcal{E}^G) := \mathcal{P}(\mathcal{E}^G) \cap \mathcal{P}_p(\mathcal{Z})$ and $\mathcal{P}_p^G(\mathcal{Z}) := \mathcal{P}^G(\mathcal{Z}) \cap \mathcal{P}_p(\mathcal{Z})$ are closed and convex subspaces of $\mathcal{P}_p(\mathcal{Z})$ (under the topology induced by $W_p$).*

*Proof of Proposition 9.* Convexity is direct by definition of the involved spaces. For closedness under the Wasserstein topology, recall (see e.g. [1]) that $W_p(\mu_n, \mu) \xrightarrow[n\to\infty]{} 0$ is equivalent to: $\mu_n \xrightarrow[n\to\infty]{} \mu$ (weak convergence) and $\int_{\mathcal{Z}} \|\theta\|^p d\mu_n(\theta) \xrightarrow[n\to\infty]{} \int_{\mathcal{Z}} \|\theta\|^p d\mu(\theta)$. Since for $f \in C_b(\mathcal{Z})$, $f \circ M_g$ (for $g \in G$) is continuous and bounded, $(\mu_n)_{n\in\mathbb{N}} \subseteq \mathcal{P}^G(\mathcal{Z})$ implies $\forall g \in G$, $M_g \# \mu = \mu$ (namely, $\mu \in \mathcal{P}^G(\mathcal{Z})$). Similarly, $f \circ P_{\mathcal{E}^G}$ is continuous and bounded, and so if $(\mu_n)_{n\in\mathbb{N}} \subseteq \mathcal{P}(\mathcal{E}^G)$, then $P_{\mathcal{E}^G} \# \mu = \mu$ (i.e. $\mu \in \mathcal{P}(\mathcal{E}^G)$). $\qquad \square$

## B.3 Properties of equivariant data

We are representing the idea of 'data being symmetric' with respect to the action of $G$, by assuming the data distribution $\pi$ to be equivariant. We will next see that this is a natural generalization of more intuitive, though restrictive, notions of data being symmetric, for instance when $X$ is a r.v. on $\mathcal{X}$ with invariant law $\pi_{\mathcal{X}}$ and $Y = f^*(X) + \xi$ for some equivariant function $f^* : \mathcal{X} \to \mathcal{Y}$ and some centered independent noise $\xi$. Indeed the following results tell us that, assuming $\pi \in \mathcal{P}^G(\mathcal{X} \times \mathcal{Y})$, implies such a structure of the data, but with a more general $\xi = \xi_X$, possibly correlated to $X$ but still 'conditionally' centered given $X$. This result will be required in proving Proposition 5.

**Proposition 10.** *Let $\pi \in \mathcal{P}(\mathcal{X} \times \mathcal{Y})$ be an equivariant data distribution such that $\mathbb{E}_\pi[\|Y\|^2] < \infty$. Then $f^* = \mathbb{E}_\pi[Y|X = \cdot]$ is an equivariant function.*

*Proof of Proposition 10.* Indeed, as $\mathbb{E}[\|Y\|^2] < \infty$, we know the conditional expectation $\mathbb{E}[Y|X]$ is well defined and there exists a measurable $f^* : \mathcal{X} \to \mathcal{Y}$ s.t. $f^*(X) = \mathbb{E}[Y|X]$. Now, by properties of the conditional expectation): Given any $h : \mathcal{X} \to \mathbb{R}$ square integrable, we will show that: $\mathbb{E}_\pi[Yh(X)] = \mathbb{E}_\pi[\int_G \hat{\rho}_g^{-1}.f^*(\rho_g.X)d\lambda_G(g)h(X)]$. Indeed, notice that by *Fubini's theorem* (as $f^* \in L^2(\mathcal{X}, \mathcal{Y}; \pi_{\mathcal{X}})$ and $h$ square integrable), linearity of the integral and $G$-invariance of $\pi$:

$$\mathbb{E}_\pi\left[(\mathcal{Q}_G.f^*)(X)h(X)\right] = \mathbb{E}_\pi\left[\iint_G \hat{\rho}_g^{-1}.f^*(\rho_g.X)d\lambda_G(g)h(X)\right]$$

$$= \int_G \hat{\rho}_g^{-1}.\mathbb{E}_\pi[f^*(X)h(\rho_g.^{-1}.X)]d\lambda_G(g)$$

$$= \int_G \mathbb{E}_\pi[\hat{\rho}_g^{-1}.Yh(\rho_g.^{-1}.X)]d\lambda_G(g)$$

$$= \int_G \mathbb{E}_\pi[Yh(X)]d\lambda_G(g) = \mathbb{E}_\pi[Yh(X)]$$

By uniqueness of the conditional expectation, we know therefore that $f^*(X) \overset{a.s.}{=} \int_G \hat{\rho}_g^{-1}.f^*(\rho_g.X)\lambda_G(g)$. In particular, $\pi_{\mathcal{X}}$-a.e. $f^* = \mathcal{Q}_G(f^*)$, making $f^*$ $G$-equivariant. $\qquad \square$

As a particular example, if we assumed that the data distribution was given by some function $f : \mathcal{X} \to \mathcal{Y}$, i.e. $Y = f(X)$, taking $\pi$ to be equivariant would be equivalent to assuming that $\pi_{\mathcal{X}}$ is invariant and $f$ is an equivariant function (which is the setting of the data simulated in our numerical experiments; see Section 4).

We notice that Proposition 10 can also be used to recover a celebrated result from [27] (later generalized by [40]), in the general setting where data symmetry is encoded by the condition that $\pi \in \mathcal{P}_2(\mathcal{X} \times \mathcal{Y})$. Define the **symmetrization gap** of a learning problem with quadratic loss as:

$$\Delta(f, \mathcal{Q}_G f) := \mathbb{E}_\pi[\|Y - f(X)\|_{\mathcal{Y}}^2] - \mathbb{E}_\pi[\|Y - (\mathcal{Q}_G f)(X)\|_{\mathcal{Y}}^2]$$

The following extension of mentioned statements from [27, 40] is not needed for our results, but we provide a proof of it in SuppMat-G, in view of its potential, independent interest:

**Lemma 2** (Symmetrization Gap Characterization)**.** *Consider the **quadratic loss** and $\pi \in \mathcal{P}(\mathcal{X} \times \mathcal{Y})$ such that $\mathbb{E}_\pi[\|Y\|^2] < \infty$. Also, assume that $\pi|_\mathcal{X}$ is $G$-invariant, but $\pi$ is only $H$-invariant with respect to some $H \leq G$ (closed). Then, the generalization gap satisfies:*

$$\Delta(f, \mathcal{Q}_G f) = -2\langle f^*, f_G^\perp \rangle_{L^2(\mathcal{X}, \mathcal{Y}; \pi_\mathcal{X})} + \|f_G^\perp\|_{L^2(\mathcal{X}, \mathcal{Y}; \pi_\mathcal{X})}^2$$

*where $f^*(x) = \mathbb{E}_\pi[Y|X = x]$ is the conditional expectation function, and $f_G^\perp := f - \mathcal{Q}_G f$.*

*In particular, if $\pi$ is $G$-invariant as well, we get $\Delta(f, \mathcal{Q}_G f) = \|f_G^\perp\|_{L^2(\mathcal{X}, \mathcal{Y}; \pi_\mathcal{X})}^2$*

# C   Concrete realizations of shallow models

The class of models we have introduced in Definition 1 allows for taking an arbitrary, common parameter space $\mathcal{Z}$ for all hidden units, as well as an arbitrary function $\sigma_* : \mathcal{X} \times \mathcal{Z} \to \mathcal{Y}$. As noted in the description of **EA**s in Section 2.3, by requiring only that $\sigma_*$ is jointly-equivariant, we moreover ensure that $G \circlearrowright_M \mathcal{Z}$ is, in some sense, *properly related* to the actions $G \circlearrowright_\rho \mathcal{X}$ and $G \circlearrowright_{\hat{\rho}} \mathcal{Y}$. This abstract property of $\sigma_*$ allows us, in fact, to encode a wide range of situations and interesting results, without delving into the specifics of a particular choice of architecture. We next analyze this notion in the concrete example of the setting of a traditional *single-hidden-layer* shallow NN.

## C.1   Traditional single layer neural networks and large ensembles of multi-layer units

Recall the finite-dimensional setting of single-hidden-layer neural networks. We considered $\mathcal{X} = \mathbb{R}^d$, $\mathcal{Y} = \mathbb{R}^c$ and $\mathcal{Z} = \mathbb{R}^{c \times b} \times \mathbb{R}^{d \times b} \times \mathbb{R}^b$, and defined, for $z = (W, A, B) \in \mathcal{Z}$ and $\sigma : \mathbb{R}^b \to \mathbb{R}^b$, $\sigma_*(x, z) := W\sigma(A^T x + B)$. This allows us to express a *shallow* NN with $N$ hidden units, of parameters $\theta = (\theta_i)_{i=1}^N \in \mathcal{Z}^N$, with $\theta_i = (W_i, A_i, B_i)$, as the function $\Phi_\theta^N : \mathcal{X} \to \mathcal{Y}$ given by:

$$\forall x \in \mathcal{X}, \ \Phi_\theta^N(x) := \frac{1}{N} \sum_{i=1}^N W_i \sigma(A_i^T . x + B_i) = \frac{1}{N} \overline{W} . \sigma(\overline{A}^T . x + \overline{B}), \tag{6}$$

where we write the expression *by blocks*, considering

$$\overline{W} = (W_1, \ldots, W_N) \in \mathbb{R}^{c \times bN}, \ \ \overline{A} = (A_1, \ldots, A_N) \in \mathbb{R}^{d \times bN}, \text{and } \overline{B} = \begin{pmatrix} B_1 \\ \vdots \\ B_N \end{pmatrix} \in \mathbb{R}^{bN}.$$

This corresponds exactly to the usual *single-hidden-layer* setting (see e.g. [23, 53, 62, 67]), only that we allow for the structure to involve the use of *block matrices*. This allows us to translate many relevant **EA**s, such as CNNs (see [19]) or DeepSets (see [77]) into the *shallow NN* framework that we propose (see Appendix C.2 for fully developed examples).

We now also consider a $G$-action on the *intermediate layer*, $G \circlearrowright_\eta \mathbb{R}^b$ (so that $G \circlearrowright_{\eta \otimes \mathrm{Id}_N} (\mathbb{R}^b)^N$). This allows us to define the *natural intertwinning* action[6] of $G$ on $\mathcal{Z}$, which is given by:

$$M_g . z = M_g . (W, A, B) := (\hat{\rho}_g . W \eta_g^T, \rho_g . A \eta_g^T, \eta_g . B), \tag{7}$$

for any $g \in G$ and $z = (W, A, B) \in \mathcal{Z}$. This is exactly the action under which the fixed points (i.e. $\mathcal{E}^G$) will correspond exactly to **EA**s in the traditional sense for this architecture (i.e. each layer being an equivariant function). In particular, we get the following straightforward result:

---

[6]The name is inspired from the traditional definition of 'intertwinning linear maps' in representation theory, see SuppMat-A.1 for an explanation.

**Proposition 11** (**Joint equivariance of $\sigma_*$ for single-hidden-layer NNs**). *In the setting of single-hidden-layer NNs described above, if $\sigma : \mathbb{R}^b \to \mathbb{R}^b$ is G-equivariant (with respect to the action given by $\eta$), then $\sigma_*$ is jointly equivariant.*

*Proof.* This fact follows directly from the specific definition of $\sigma_*$, the linearity and the orthogonality of all the relevant group representations. Indeed, for any equivariant $\sigma : \mathbb{R}^b \to \mathbb{R}^b$, any $g \in G$ and any $z = (W, A, B) \in \mathcal{Z}$, we have:

$$
\begin{aligned}
\sigma_*(\rho_g.x, M_g.z) &= \sigma_*(\rho_g.x, (\hat{\rho}_g.W.\eta_g^T, \rho_g.A.\eta_g^T, \eta_g.B)) \\
&= (\hat{\rho}_g.W.\eta_g^T).\sigma((\rho_g.A.\eta_g^T)^T.(\rho_g.x) + \eta_g.B) \\
&= \hat{\rho}_g.(W.\eta_g^T.\sigma(\eta_g.A^T.(\rho_g^T.\rho_g).x + \eta_g.B)) \\
&= \hat{\rho}_g.(W.\eta_g^T.\sigma(\eta_g.(A^T.x + B))) \\
&= \hat{\rho}_g.(W.\eta_g^T.\eta_g.\sigma(A^T.x + B)) \\
&= \hat{\rho}_g.(W.\sigma(A^T.x + B)) \\
&= \hat{\rho}_g.\sigma_*(x, z)
\end{aligned}
$$

$\square$

Thus, in this particular case, in which $\sigma_*$ represents the unit of a single hidden layer neural network, we require only for $\sigma : \mathbb{R}^b \to \mathbb{R}^b$ to be $G$-equivariant (with respect to the action given by $\eta$ on $\mathbb{R}^b$) in order for the joint equivariance of $\sigma_*$ to hold (and, in consequence, most subsequent results from our work).

In particular, if $\eta$ is chosen to be trivial (i.e. $\eta \equiv id_{\mathbb{R}^b}$), any single-hidden-layer NN is jointly-equivariant,[7] regardless of $\sigma : \mathbb{R}^b \to \mathbb{R}^b$. Namely, all of the results contained in the core of the paper can be applied to an arbitrary single-hidden-layer NN; most importantly, those relating **vanilla**, **DA** and **FA** training. The disadvantage of having a trivial $\eta$ is that the space $\mathcal{E}^G$ might not *encode* very interesting **EA**s. This might make Theorem 5 lose part of its impressiveness, but it takes no credit off the rest of our results (such as Corollary 3 and Theorem 4).

Analogously, if $\eta$ acts via permutations of the coordinates in $\mathbb{R}^b$, it is enough to consider a $\sigma$ that is the *pointwise* application of a scalar function. In practice, this usually isn't a restrictive condition, since most commonly used NN architectures are naturally built following this pattern. Therefore, almost any single-hidden-layer architecture can yield a jointly-equivariant $\sigma_*$ for some of the most common and interesting finite groups (e.g. $\mathcal{S}_n$, $C_n$, among many others; see [30] for further discussion).

For a more complex (possibly infinite) compact group $G$ acting orthogonally on the data and parameters (with non-trivial $\eta$), for $\sigma_*$ to be jointly-equivariant we have to start properly restraining $\sigma : \mathbb{R}^b \to \mathbb{R}^b$. For instance, choosing an $O(b)$-equivariant $\sigma$ (e.g. a Norm-ReLU) would grant Proposition 11 to always hold; but such a restraining choice could potentially harm the model's expressiveness and applicability. We leave a deeper exploration of this more challenging case as future work.

Finally, all of the above discussion (including Proposition 11) readily generalizes to the multilayer case. Namely, if $\sigma_*$ encodes a Multi-Layer Perceptron (MLP) with multiple hidden layers whose parameters live in some linear space $\mathcal{Z}$, we can define $G \circlearrowright_M \mathcal{Z}$ as the *intertwining* action between each successive layer (similar to the previous example). $\sigma_*$ can be made jointly-equivariant by making every activation function on each hidden layer equivariant (see [30]). Then, $\mathcal{E}^G$ corresponds exactly to the parameters that make the entire MLP an equivariant architecture (in the sense that every layer is an equivariant function). With this, $\Phi_\theta^N$ is an ensemble of $N$ such MLPs trained in parallel, to which our results would also apply. Also as before, if the $G$-action on all of the hidden layers (but not on input/output) is made trivial, then any $\sigma_*$ representing a multilayer architecture can be jointly-equivariant.

In the upcoming sections we will further develop these ideas, to show that some of the most relevant and widely applied equivariant architectures can be realized as part of our setting.

---

[7]Notice that this fact is explicitly used in the proof of Proposition 4

## C.2 Shallow DeepSet models, shallow GNNs and shallow CNNs

### C.2.1 DeepSets

An emblematic example of **neural networks with equivariant architecture** are the **Deep Sets**, introduced by [77]; which correspond, in practice, to NN architectures designed to be **invariant/equivariant** to the action of the group of permutations $G = \mathcal{S}_n$.

Consider that our NN processes **sets of size** $n$, which contain real-valued vectors of dimension $\tilde{d} \in \mathbb{N}$. We can represent this input space simply as $\mathcal{X} := \mathbb{R}^{n \times \tilde{d}}$. Say we wanted to build a wide single-hidden-layer network that is invariant/equivariant to the action of $\mathcal{S}_n$, and returns a new set of $n$ vectors, but now of dimension $\tilde{c} \in \mathbb{N}$ (i.e. $\mathcal{Y} := \mathbb{R}^{n \times \tilde{c}}$). For this, let's consider the same architecture as in the previous section (replace $d = n\tilde{d}, c = n\tilde{c}$), where we'll let our intermediate layer be simply another set of $n$ vectors, now of dimension $\tilde{b}$ with $\tilde{b} \in \mathbb{N}$ (such that $b = n\tilde{b}$ above), and repeated $N \in \mathbb{N}$ times (as in the multiple hidden units that we want to achieve). With this, our network flows as follows: $\Phi_\theta^N : \mathbb{R}^{n \times \tilde{d}} \to \mathbb{R}^{(n \times \tilde{b}) \times N} \to \mathbb{R}^{n \times \tilde{c}}$. Notice how $\mathcal{S}_n$ acts *naturally* on each hidden space by simply permuting the vectors of the set (i.e. $\rho$, $\hat{\rho}$ and $\eta$ are defined in this way).

Following the same structure as in the previous section, we will have a parameter space given by: $\mathcal{Z} := \mathbb{R}^{(n \times \tilde{c}) \times (n \times \tilde{b})} \times \mathbb{R}^{(n \times \tilde{d}) \times (n \times \tilde{b})} \times \mathbb{R}^{n \times \tilde{b}}$, and a unit that acts on $z = (W, A, B) \in \mathcal{Z}$ as $\sigma_*(x, z) := W\sigma(A^T x + B)$, with $\sigma : \mathbb{R}^{n \times \tilde{b}} \to \mathbb{R}^{n \times \tilde{b}}$ some usual activation function applied pointwise (which, as mentioned above, will be immediately equivariant to the defined action of $G$ via $\eta$). Notice how building this architecture requires no hard *a priori* knowledge of the underlying symmetry of the problem (beyond the fact that the inputs and outputs are sets).

Under the natural action from this context (i.e. $\rho$, $\hat{\rho}$ and $\eta$ acting on the sets by permuting their elements, and $M$ defined as in SuppMat-C.1), $\mathcal{E}^G$ corresponds exactly to the set of parameters that yield a $\mathcal{S}_n$-equivariant shallow neural network (which can be interpreted as a DeepSet). For the interested reader, we will make this connection explicit in the rest of this section.

As shown in [77], the only way to have an $\mathcal{S}_n$-equivariant affine transformation between two spaces $\mathbb{R}^{n \times d_1}$ and $\mathbb{R}^{n \times d_2}$ is if the parameters $\tilde{A} \in \mathbb{R}^{(n \times d_1) \times (n \times d_2)}$ and $\tilde{B} \in \mathbb{R}^{n \times d_2}$ (from the definition $x \mapsto \tilde{A}^T.x + \tilde{B}$) are of the form:

$$\tilde{A} = \alpha \otimes I + \beta \otimes J, \qquad \tilde{B} = \gamma \otimes (1, \ldots, 1) \tag{8}$$

Where $\alpha, \beta \in \mathbb{R}^{d_1 \times d_2}, \gamma \in \mathbb{R}^{d_2}$ are the truly trainable parameters of the layer; $I = \mathbf{Id}_{n \times n}$ and $J = \vec{1}_n \vec{1}_n^T$ are two $n \times n$ matrices; and $\otimes$ is the usual tensor product. More explicitly, writing the matrices by blocks, this is:

$$\alpha \otimes I = \begin{pmatrix} \alpha & 0 & \ldots & 0 \\ 0 & \alpha & \ldots & 0 \\ \vdots & \ddots & \ddots & 0 \\ 0 & 0 & \ldots & \alpha \end{pmatrix} \in \mathbb{R}^{(n \times d_1) \times (n \times d_2)},$$

$$\beta \otimes J = \begin{pmatrix} \beta & \beta & \ldots & \beta \\ \beta & \beta & \ldots & \beta \\ \vdots & \ddots & \ddots & \beta \\ \beta & \beta & \ldots & \beta \end{pmatrix} \in \mathbb{R}^{(n \times d_1) \times (n \times d_2)}$$

$$\gamma \otimes (1, \ldots, 1) = (\gamma, \ldots, \gamma) \in \mathbb{R}^{(n \times d_2)}$$

In particular, Equation (8) gives us an explicit expression for our space $\mathcal{E}^G$. Indeed, an element $z = (W, A, B) \in \mathcal{Z}$ will live in $\mathcal{E}^G$ if and only if there exists $w_1, w_2 \in \mathbb{R}^{\tilde{c} \times \tilde{b}}$, $a_1, a_2 \in \mathbb{R}^{\tilde{d} \times \tilde{b}}$ and $b_1 \in \mathbb{R}^{\tilde{b}}$ such that:

$$W = w_1 \otimes I + w_2 \otimes J, \ A = a_1 \otimes I + a_2 \otimes J, \text{ and } B = b_1 \otimes (1, \ldots, 1) \tag{9}$$

In a sense, we can think of $\mathcal{E}^G$ simply as being equivalent to $\mathbb{R}^{2(\tilde{c} \times \tilde{b})} \times \mathbb{R}^{2(\tilde{d} \times \tilde{b})} \times \mathbb{R}^{\tilde{b}}$. In particular, we went from having $D = \dim(\mathcal{Z}) = n^2 \cdot \tilde{b} \cdot (\tilde{c} + \tilde{d}) + n \cdot \tilde{b}$ free parameters on each unit, to simply $\tilde{D} = \dim(\mathcal{E}^G) = 2 \cdot \tilde{b} \cdot (\tilde{c} + \tilde{d}) + \tilde{b}$, which should be easier to manage in general.

Now, recall our construction from SuppMat-C.1, and consider the matrices $\overline{W}, \overline{A}$ and $\overline{B}$ from Equation (6). We notice that, in this example, they are of the form:

$$\overline{W} \in \mathbb{R}^{(n \times \tilde{c}) \times (n \times \tilde{b}N)}, \quad \overline{A} \in \mathbb{R}^{(n \times \tilde{d}) \times (n \times \tilde{b}N)}, \text{and} \quad \overline{B} \in \mathbb{R}^{(n \times \tilde{b}N)}.$$

Namely, sensibly replacing $d_1 = \tilde{d}$, $d_2 = \tilde{b}N$ for $\overline{A}$ and $\overline{B}$; and $d_1 = \tilde{c}$, $d_2 = \tilde{b}N$ for $\overline{W}$ in Equation (8); we also get an explicit condition under which $\overline{W}, \overline{A}$ and $\overline{B}$ yield a globally $\mathcal{S}_n$-equivariant architecture. By properly writing these matrices by blocks (as in SuppMat-C.1), and separating the action of each of the $N$ units, one shall notice that the condition for $\overline{W}, \overline{A}$ and $\overline{B}$ to be $\mathcal{S}_n$-equivariant corresponds exactly to every $\theta_i = (W_i, A_i, B_i) \in \mathcal{Z}$, $i = 1, \ldots, N$, being of the form given in Equation (9). That is to say, $\Phi_\theta^N$ has an $\mathcal{S}_n$-equivariant architecture (a DeepSet from [77]) if and only if $\forall i \in \{1, \ldots, N\}, \theta_i \in \mathcal{E}^G$, which is exactly the condition stated in Section 2.3.

**Remark.** *The reader could notice that our previous construction is not truly involving the complete richness of DeepSets. Namely, these architectures often involve using some "pooling" mechanisms, such as a global average pooling (GAP) operation to force invariance into the network (see [8]). Namely, if $\mathcal{A} : \mathbb{R}^{n \times \tilde{b}} \to \mathbb{R}^{\tilde{b}}$ is the usual linear GAP operation, we might want a NN that flows as $\Phi_\theta^N : \mathbb{R}^{n \times \tilde{d}} \to \mathbb{R}^{(n \times \tilde{b}) \times N} \xrightarrow{\mathcal{A}} \mathbb{R}^{\tilde{b}N} \to \mathbb{R}^{\tilde{c}}$. This is no trouble within our framework, since we can simply consider the 'adequate' unit: $\sigma_*(x, z) := W\mathcal{A}(\sigma(A^T x + B))$ for $z = (W, A, B) \in \mathcal{Z} := \mathbb{R}^{\tilde{c} \times \tilde{b}} \times \mathbb{R}^{(n \times \tilde{d}) \times (n \times \tilde{b})} \times \mathbb{R}^{n \times \tilde{b}}$, and all of our results would be applicable.*

*The main disadvantage of doing this is that we are forced to encode some a priori knowledge of the symmetries of the problem into our choice of architecture. While this isn't useful for our heuristic from Section 4.2, all other results relating the **DA**, **FA** and **EA** training dynamics still apply.*

**Remark.** *Similar to the last remark, more complex equivariant NN, with multiple layers and various inner operations involved, can be modeled by choosing $\sigma_*$ properly. Namely, set $\sigma_*$ to be a whole multi-layer structure, with parameters in $\mathcal{Z}$, and $\mathcal{E}^G$ being the subspace of those that make the architecture equivariant (see e.g. Equation (9)).*

*In such case, the shallow model $\Phi_\theta^N$, with $\theta \in \mathcal{Z}^N$, will represent an **ensemble** of $N$ multi-layer units, each one given, for every $i \in \{1, \ldots, N\}$, by $\sigma_*(\cdot, \theta_i) : \mathcal{X} \to \mathcal{Y}$. The output of the ensemble is simply the average of the outputs of each one of the units.*

### C.2.2 GNNs

Generalizing the ideas from DeepSets to GNNs is fairly straightforward. Namely, consider the input of a layer to be a graph, represented by a set of features, coupled with an adjacency matrix that also contains relevant edge-features; and the output to be analogous. Namely, let's say the input space is $\mathcal{X}_1 = \mathbb{R}^{(n \times d_1)} \times \mathbb{R}^{(n \times n) \times d_2}$ and the output space is $\mathcal{X}_2 = \mathbb{R}^{(n \times d_3)} \times \mathbb{R}^{(n \times n) \times d_4}$. Consider also the natural $\mathcal{S}_n$ action acting on these spaces, i.e. permuting the vertices of the graph, acting jointly between vertex features and adjacency matrix. With this in place, one can find an analogous characterization to Equation (8), but for affine layers that are $\mathcal{S}_n$-equivariant between graph spaces (see e.g. [29, 49] for a more explicit construction). From there, it is not hard to construct, following the same steps as for DeepSets, an explicit characterization of $\mathcal{E}^G$ for single-hidden-layer GNNs, analogous to Equation (9). Also as before, more complex GNN structures (with multiple layers, pooling operations, etc.) can be encoded in this setting through the *unit* $\sigma_*$ (with the possibly same drawbacks as in DeepSets). For brevity, we don't delve into GNNs in their full complexity and leave that to the curious reader.

### C.2.3 CNNs

Finally, following the same logic as before, we can model another one of the most emblematic traditional equivariant models: CNNs. We here consider only 1D-CNNs for simplicity, but all arguments can be readily generalized to 2D or 3D CNNs.

In this setting, take an array of $n$ vectors of dimension $\tilde{d} \in \mathbb{N}$ as input ($\mathcal{X} := \mathbb{R}^{n \times \tilde{d}}$) and, analogous to DeepSets, say that the output and hidden layers have the same structure, so that $\mathcal{Y} := \mathbb{R}^{n \times \tilde{c}}$; and $\mathcal{Z} := \mathbb{R}^{(n \times \tilde{c}) \times (n \times \tilde{b})} \times \mathbb{R}^{(n \times \tilde{d}) \times (n \times \tilde{b})} \times \mathbb{R}^{n \times \tilde{b}}$, for $\tilde{b}, \tilde{c} \in \mathbb{N}$. We consider $\sigma_*$ simply as before.

The single main difference with the study of DeepSets is that, in this case, we consider the action of $C_n$ (i.e. the cyclic group of order $n$, also denoted $\mathbb{Z}_n$) instead of $\mathcal{S}_n$. In particular, the *natural action*

of $C_n$ on each space (through $\rho$, $\hat{\rho}$ and $\eta$) consists simply of shifting the array's coordinates in a given direction (modulo $n$). As before, this makes the set $\mathcal{E}^G$ correspond exactly to the set of parameters that yield each layer $C_n$-equivariant (as in a 1D-CNN).

More specifically, we can characterize (analogous to Equation (8)) how a single $C_n$-equivariant layer looks like between two spaces $\mathcal{X}_1 = \mathbb{R}^{(n \times d_1)}$ and $\mathcal{X}_2 = \mathbb{R}^{(n \times d_2)}$ under the *natural* action. It is well known (see [8]) that a $C_n$-equivariant affine layer, $x \mapsto \tilde{A}^T.x + \tilde{B}$, between $\mathcal{X}_1$ and $\mathcal{X}_2$, of parameters $\tilde{A} \in \mathbb{R}^{(n \times d_1) \times (n \times d_2)}$ and $\tilde{B} \in \mathbb{R}^{n \times d_2}$, must be of the form:

$$\tilde{A} = C(\alpha_0, \dots, \alpha_{n-1}) \ \text{ and } \ \tilde{B} = \beta \otimes (1, \dots, 1),$$

with $\beta \in \mathbb{R}^{d_2}$, $\alpha_i \in \mathbb{R}^{d_2 \times d_1}$, $\forall i \in \{0, \dots, n-1\}$, and the associated **circulant matrix** being defined (by blocks, and considering the indices modulo $n$) as:

$$C(\alpha_0, \dots, \alpha_{n-1}) = \begin{pmatrix} \alpha_0 & \alpha_1 & \dots & \dots & \alpha_{-1} \\ \alpha_{-1} & \alpha_0 & \alpha_1 & \dots & \dots \\ \vdots & \vdots & \ddots & \vdots & \vdots \\ \dots & \dots & \alpha_{-1} & \alpha_0 & \alpha_1 \\ \alpha_1 & \dots & \dots & \alpha_{-1} & \alpha_0 \end{pmatrix} \in \mathbb{R}^{(n \times d_2) \times (n \times d_1)}.$$

As a consequence, we get an analog of Equation (9) to explicitly describe $\mathcal{E}^G$: we have that $z = (W, A, B) \in \mathcal{E}^G$ if and only if $\exists (w_i)_{i=0}^{n-1} \subseteq \mathbb{R}^{\tilde{c} \times \tilde{b}}$, $(a_i)_{i=0}^{n-1} \subseteq \mathbb{R}^{\tilde{d} \times \tilde{b}}$ and $\beta \in \mathbb{R}^{\tilde{b}}$ such that:

$$W = C(w_0, \dots, w_{n-1}), \ A = C(a_0, \dots, a_{n-1}), \text{ and } B = \beta \otimes (1, \dots, 1)$$

Thus, the parameter space goes from having $\dim(\mathcal{Z}) = n^2 \cdot \tilde{b} \cdot (\tilde{c} + \tilde{d}) + n \cdot \tilde{b}$ free parameters on each unit, to simply $\dim(\mathcal{E}^G) = n \cdot \tilde{b} \cdot (\tilde{c} + \tilde{d}) + \tilde{b}$. One might also notice that the obtained parameter space $\mathcal{E}^{C_n}$ **contains** $\mathcal{E}^{S_n}$ (from DeepSets), which is expected from the fact that $C_n \leq S_n$. In global terms, the *shallow CNNs* we have modeled here, correspond to models that will grow asymptotically in terms of the *number of different convolutional filters* (encoded by $N$) that are being used.

Finally, as it was also mentioned in previous examples, more complex CNN structures (with multiple layers, pooling operations, etc.) can be encoded in this setting through modifications to the *unit $\sigma_*$*.

# D Further elements from the MF theory of shallow neural networks

In this section we study several theoretical notions required in the MF approach to overparametrized shallow NN. For the purpose of our results, Subsections D.1 and D.2 are the most relevant ones, as we establish therein some useful properties or formula that will be explicitly required. Subsections D.3 and D.4 review some results and recent literature regarding well-posedness and long-time convergence of WGFs, which are relevant to the MF interpretation of the training dynamics of shallow models.

## D.1 Linear functional derivatives and intrinsic derivatives

Let $\mathcal{X}, \mathcal{Y}$ and $\mathcal{Z}$ be separable Hilbert spaces. Recall the definitions of the linear functional derivative and intrinsic derivatives:

**Definition 8** (Linear Functional Derivative (First Variation)). *For a functional $F : \mathcal{P}_2(\mathcal{Z}) \to \mathbb{R}$, its linear functional derivative (lfd) is a function: $\frac{\partial F}{\partial \mu} : \mathcal{P}_2(\mathcal{Z}) \times \mathcal{Z} \to \mathbb{R}$ such that $\forall \mu, \nu \in \mathcal{P}_2(\mathcal{Z})$:*

$$\lim_{h \to 0} \frac{F((1-h)\mu + h\nu) - F(\mu)}{h} = \int_{\mathcal{Z}} \frac{\partial F}{\partial \mu}(\mu, z) d(\nu - \mu)(z), \ \text{ and } \ \int_{\mathcal{Z}} \frac{\partial F}{\partial \mu}(\mu, z) d\mu(z) = 0$$

*The function $F' : \mu \in \mathcal{P}_2(\mathcal{Z}) \mapsto \frac{\partial F}{\partial \mu}(\mu, \cdot)$ is also known as the first variation of $R$ at $\mu$.*

**Definition 9** (Intrinsic Derivative). *Whenever $\frac{\partial F}{\partial \mu} : \mathcal{P}_2(\mathcal{Z}) \times \mathcal{Z} \to \mathbb{R}$ exists and is **differentiable** on its second argument, the intrinsic derivative of $F$ is defined as: $D_\mu F(\mu, z) = \nabla_z \left( \frac{\partial F}{\partial \mu}(\mu, z) \right)$.*

**Example.** *To better illustrate the notion of the linear functional derivative and the intrinsic derivative, consider the following usual examples:*

1. In the important case of the **Boltzmann entropy** $F = H_\lambda$, defined for $\mu \lll \lambda$ by $H_\lambda := \int \log(\frac{d\mu}{d\lambda}(z))d\mu(z)$ (and $+\infty$ otherwise), we have that (modulo an additive constant that doesn't depend on z, see [56]):

$$\frac{\partial F}{\partial \mu}(\mu, z) = \log\left(\frac{d\mu}{d\lambda}(z)\right) + 1 \ \ and \ \ D_\mu F(\mu, z) = \frac{1}{\frac{d\mu}{d\lambda}(z)}\nabla_z \frac{d\mu}{d\lambda}(z).$$

2. Whenever $F(\mu) := \int_{\mathcal{Z}} \phi(z)d\mu(z)$ for some bounded continuously differentiable function $\phi : \mathcal{Z} \to \mathbb{R}$, it is well known that :

$$\frac{\partial F}{\partial \mu}(\mu, z) = \phi(z) - \int \phi d\mu \ \ and \ \ D_\mu F(\mu, z) = \nabla_z \phi(z)$$

The most relevant example, in our case, regards the linear functional and intrinsic derivatives of the **population risk** functional, $R(\mu) = \mathbb{E}_\pi[\ell(\Phi_\mu(X), Y)]$. We can consider the general setting presented in [16], where some Hilbert Space $\mathcal{H}$ is considered, and it is assumed that $F : \mathcal{P}(\mathcal{Z}) \to \mathbb{R}$ can be written as $F(\mu) = L(\langle\Phi, \mu\rangle)$, where $\Phi : \mathcal{Z} \to \mathcal{H}$ is a parametrization of elements in $\mathcal{H}$; $L : \mathcal{H} \to \mathbb{R}$ is some *loss* functional, and the integral $\langle\Phi, \mu\rangle$ is a **Bochner integral** on $\mathcal{H}$. This generalizes the *shallow NN* setting, as one might consider $\mathcal{H} = L^2(\mathcal{X}, \mathcal{Y}, \pi_\mathcal{X})$, $L : \mathcal{H} \to \mathbb{R}$ given by $L(f) = \mathbb{E}_\pi[\ell(f(X), Y)]$ and $\Phi : \mathcal{Z} \to \mathcal{H}$ defined as $\forall z \in \mathcal{Z}, \ \Phi(z) = \sigma_*(\cdot; z)$; so that $R(\mu) = L(\langle\Phi, \mu\rangle)$. In this setting, we can prove the following result:[8]

**Proposition 12.** *Let $\mathcal{H}$ be a separable Hilbert Space and $F(\mu) := L(\langle\Phi, \mu\rangle)$, for some function that's Gateaux-differentiable $L : \mathcal{H} \to \mathbb{R}$ on every direction and of continuous differential; and $\Phi : \mathcal{Z} \to \mathcal{H}$ such that $\forall \mu \in \mathcal{P}(\mathcal{Z}), \ \|\langle\Phi, \mu\rangle\|_\mathcal{H} < \infty$.*

*Then $\forall z \in \mathcal{Z}, \ \forall \mu \in \mathcal{P}(\mathcal{Z})$:*

$$\frac{\partial F}{\partial \mu}(\mu, z) = D_h L(\langle\Phi, \mu\rangle)(\Phi(z)) = \langle\nabla_h L(\langle\Phi, \mu\rangle), \Phi(z)\rangle_\mathcal{H} - C_{F,\mu}$$

$$D_\mu F(\mu, z) = (D_h L(\langle\Phi, \mu\rangle)(D_z\Phi(z)))^* = \nabla_z \Phi(z)(\nabla_h L(\langle\Phi, \mu\rangle)).$$

*Here, $C_{F,\mu} := \langle\nabla_h L(\langle\Phi, \mu\rangle), \langle\Phi, \mu\rangle\rangle_\mathcal{H}$ is exactly the constant needed to avoid ambiguity in the definition; $(\cdot)^*$ denotes the adjoint operator and, in particular, $\nabla_z\Phi(z) = (D_z\Phi(z))^* : \mathcal{H} \to \mathcal{Z}$. When $\mathcal{Z} = \mathbb{R}^D$ this corresponds to the usual definition of the gradient.*

*Proof of Proposition 12.* We know that, $\forall \mu, \nu \in \mathcal{P}(\mathcal{Z}), \ h \in [0, 1]$:

$$\frac{F((1-h)\mu + h\nu) - F(\mu)}{h} = \frac{L(\langle\Phi, (1-h)\mu + h\nu\rangle) - L(\langle\Phi, \mu\rangle)}{h}$$
$$= \frac{L(\langle\Phi, \mu\rangle + h\langle\Phi, \nu - \mu\rangle) - L(\langle\Phi, \mu\rangle)}{h}.$$

Let's denote by $q_\mu := \langle\Phi, \mu\rangle$ (analogously $q_{\nu-\mu} := \langle\Phi, \nu - \mu\rangle$) and $s_{\mu,\nu} := hq_{\nu-\mu}$, so we can write:

$$\frac{F((1-h)\mu + h\nu) - F(\mu)}{h} = \frac{L(q_\mu + s_{\mu,\nu}) - L(q_\mu)}{h}.$$

As $L$ is Gateaux differentiable, we have the following first order Taylor expansion $\forall x, s \in \mathcal{H}, \forall t \in \mathbb{R}$:

$$L(x + t\,s) = L(x) + t\,D_h L(x).s + o(|t|\|s\|),$$

which allows us to write:

$$\frac{F((1-h)\mu + h\nu) - F(\mu)}{h} = \frac{L(q_\mu + h\,q_{\nu-\mu}) - L(q_\mu)}{h} = \frac{h.D_h L(q_\mu).q_{\nu-\mu} + o(|h|\|q_{\nu-\mu}\|)}{h}.$$

As $\|q_{\nu-\mu}\| < \infty$ by hypothesis, we can say that: $o(|h|\|q_{\nu-\mu}\|) = o(h)$. Therefore, taking the limit with $h \to 0$, we get that:

$$\lim_{h\to 0} \frac{F((1-h)\mu + h\nu) - F(\mu)}{h} = D_h L(q_\mu).q_{\nu-\mu} + \lim_{h\to 0} \frac{o(h)}{h} = D_h L(q_\mu).q_{\nu-\mu}$$

---

[8]Where the **gradient** $\nabla_x f(x)$ is the unique vector in $\mathcal{H}$ representing the action of $D_x f(x) : \mathcal{H} \to \mathbb{R}$

Now, developping this last term (using, for instance, the linearity of the Bochner integral, as we know $D_h L(x, \cdot).(\cdot)$ to be linear and bounded), we get that:

$$D_h L(q_\mu).q_{\nu-\mu} = D_h L(q_\mu).\langle \Phi, \nu - \mu \rangle = \langle D_h L(q_\mu)(\Phi), \nu - \mu \rangle,$$

and so by definition of the gradient of $L$:

$$\lim_{h \to 0} \frac{F((1-h)\mu + h\nu) - F(\mu)}{h} = \int_{\mathcal{Z}} \langle \nabla_h L(\langle \Phi, \mu \rangle), \Phi(z) \rangle_{\mathcal{H}} \, d(\nu - \mu)(z).$$

From here we deduce that:

$$\frac{\partial F}{\partial \mu}(\mu, z) = \langle \nabla_h L(\langle \Phi, \mu \rangle), \Phi(z) \rangle_{\mathcal{H}} - C_{F,\mu},$$

where $C_{F,\mu}$ is a fixed constant, given by:

$$C_{F,\mu} = \int_{\mathcal{Z}} \frac{\partial F}{\partial \mu}(\mu, z) d\mu(z) = \int_{\mathcal{Z}} \langle \nabla_h L(\langle \Phi, \mu \rangle), \Phi(z) \rangle_{\mathcal{H}} \, d(\mu)(z) = \langle \nabla_h L(\langle \Phi, \mu \rangle), \langle \Phi, \mu \rangle \rangle_{\mathcal{H}}$$

On the other hand, for the intrinsic derivative, notice that $D_z(\frac{\partial F}{\partial \mu}(\mu, z)) : \mathcal{Z} \to \mathbb{R}$ is a bounded linear functional over $\mathcal{Z}$, so (by Riesz Representation) $\exists D_\mu F(\mu, z) := \nabla_z(\frac{\partial F}{\partial \mu}(\mu, z)) \in \mathcal{Z}$ such that:

$$\forall z \in \mathcal{Z}, \ \left\langle \nabla_z \left( \frac{\partial F}{\partial \mu}(\mu, z) \right), z \right\rangle_{\mathcal{Z}} = D_z \left( \frac{\partial F}{\partial \mu}(\mu, z) \right)(z)$$

However, we can develop the RHS, and as the constant $C_{F,\mu}$ doesn't depend on $z$, we get that:

$$D_z \left( \frac{\partial F}{\partial \mu}(\mu, z) \right)(z) = D_z \left( \langle \nabla_h L(\langle \Phi, \mu \rangle), \Phi(z) \rangle_{\mathcal{H}} \right)(z)$$

Now, by the chain rule and the definition of the *adjoint operator* of $D_z \Phi(z)$:

$$D_z \left( \frac{\partial F}{\partial \mu}(\mu, z) \right)(z) = \langle \nabla_h L(\langle \Phi, \mu \rangle), D_z \Phi(z)(z) \rangle_{\mathcal{H}} = \left\langle (D_z \Phi(z))^* (\nabla_h L(\langle \Phi, \mu \rangle)), z \right\rangle_{\mathcal{Z}}$$

So, as they coincide for every $z \in \mathcal{Z}$, we conclude that:

$$D_\mu F(\mu, z) = (D_z \Phi(z))^* (\nabla_h L(\langle \Phi, \mu \rangle))$$

$\square$

Proposition 12 applies directly to our population risk functional $R(\mu) := \mathbb{E}_\pi \left[ \ell(\Phi_\mu(X), Y) \right]$, by considering:

- The Hilbert space: $\mathcal{H} = L^2(\mathcal{X}, \mathcal{Y}, \pi_{\mathcal{X}})$
- $L : \mathcal{H} \to \mathbb{R}$ given by $L(f) = \mathbb{E}_\pi[\ell(f(X), Y)]$, which is Gateaux-differentiable on every direction in $\mathcal{H}$ if we assume $\ell : \mathcal{Y} \times \mathcal{Y} \to \mathbb{R}$ to be continuously differentiable on its first argument, with $\nabla_1 \ell$ linearly growing. The differential (which is continuous) can be explicitly computed to be:

$$D_h L(f)(h) = \mathbb{E}_\pi \left[ \langle \nabla_1 \ell((f(X), Y), h(X) \rangle_{\mathcal{Y}} \right]$$

- $\Phi : \mathcal{Z} \to \mathcal{H}$ defined as $\forall z \in \mathcal{Z}, \ \Phi(z) = \sigma_*(\cdot; z)$, which satisfies $\forall \mu \in \mathcal{P}(\mathcal{Z}), \ \|\langle \Phi, \mu \rangle\|_{\mathcal{H}} < \infty$ under the assumption of $\sigma_*$ being **bounded** and continuous.

**Corollary 5.** *We can explicitly compute the linear functional derivative and the intrinsic derivative for the learning problem's population risk:*

$$\frac{\partial R}{\partial \mu}(\mu, z) = \mathbb{E}_\pi \left[ \langle \nabla_1 \ell(\langle \sigma_*(X; \cdot), \mu \rangle, Y), \sigma_*(X; z) \rangle_{\mathcal{Y}} \right] + \text{(constant not depending on z)}$$

$$D_\mu R(\mu, z) = \mathbb{E}_\pi \left[ \nabla_z \sigma_*(X; z). \nabla_1 \ell(\langle \sigma_*(X; \cdot), \mu \rangle, Y) \right]$$

Beyond this particular example, the linear functional derivative and the intrinsic derivative behave well when the underlying functional is invariant, as shown by the following result:

**Proposition 13.** *Let $F : \mathcal{P}(\mathcal{Z}) \longrightarrow \mathbb{R}$ be invariant and of class $\mathcal{C}^1$. Then: $\forall z \in \mathcal{Z}$, $\forall \mu \in \mathcal{P}(\mathcal{Z})$, $\forall g \in G$:*

$$\frac{\partial F}{\partial \mu}(M_g \# \mu, M_g.z) = \frac{\partial F}{\partial \mu}(\mu, z) \quad and \quad D_\mu F(M_g \# \mu, M_g.z) = M_g.D_\mu F(\mu, z)$$

*i.e. $\frac{\partial F}{\partial \mu}$ is jointly invariant and $D_\mu F$ jointly equivariant.*

*Proof of Proposition 13.* To prove this, recall that the linear functional derivative of $F$ is the only function $\frac{\partial F}{\partial \mu} : \mathcal{P}(\mathcal{Z}) \times \mathcal{Z} \to \mathbb{R}$ satisfying $\forall \mu, \nu \in \mathcal{P}(\mathcal{Z})$:

$$\lim_{h \to 0} \frac{F((1-h)\mu + h\nu) - F(\mu)}{h} = \int_{\mathcal{Z}} \frac{\partial F}{\partial \mu}(\mu, z)d(\nu - \mu)(z) \quad and \quad \int_{\mathcal{Z}} \frac{\partial F}{\partial \mu}(\mu, z)d\mu(z) = 0.$$

In particular, as $F$ is $G$-invariant (and $M_g$ linear), we can write $\forall \mu, \nu \in \mathcal{P}(\mathcal{Z})$, $\forall h \neq 0$ and $g \in G$:

$$\frac{F((1-h)\mu + h\nu) - F(\mu)}{h} = \frac{F((1-h)(M_g \# \mu) + h(M_g \# \nu)) - F(M_g \# \mu)}{h}.$$

Taking the limit as $h \to 0$ on both sides, we get:

$$\lim_{h \to 0} \frac{F((1-h)\mu + h\nu) - F(\mu)}{h} = \int_{\mathcal{Z}} \frac{\partial F}{\partial \mu}(M_g \# \mu, z)d(M_g \# \nu - M_g \# \mu)(z)$$

$$= \int_{\mathcal{Z}} \frac{\partial F}{\partial \mu}(M_g \# \mu, M_g.z)d(\nu - \mu)(z),$$

and also:

$$\int_{\mathcal{Z}} \frac{\partial F}{\partial \mu}(M_g \# \mu, M_g.z)d\mu(z) = \int_{\mathcal{Z}} \frac{\partial F}{\partial \mu}(M_g \# \mu, z)dM_g \# \mu(z) = 0.$$

So, by uniqueness, we get $\forall g \in G$:

$$\frac{\partial F}{\partial \mu}(\mu, z) = \frac{\partial F}{\partial \mu}(M_g \# \mu, M_g.z),$$

and, from proposition 6 (since $\frac{\partial F}{\partial \mu}$ is jointly invariant), we get $\forall g \in G$:

$$D_\mu F(M_g \# \mu, M_g.z) = \nabla_z \left( \frac{\partial F}{\partial \mu} \right)(M_g.\mu, M_g.z) = M_g.\nabla_z \left( \frac{\partial F}{\partial \mu} \right)(\mu, z) = M_g.D_\mu F(\mu, z)$$

$$\square$$

Finally, for convenience, let us denote by $H^{\mathcal{E}^G}$ the function defined on $\mathcal{P}(\mathcal{Z})$ by $H^{\mathcal{E}^G}(\mu) := H_{\lambda_{\mathcal{E}^G}}(\mu^{\mathcal{E}^G}) = H_{\lambda_{\mathcal{E}^G}} \circ P_{\mathcal{E}^G} \# (\mu)$ (presented in Section 3.3). We can straightforwardly compute its linear functional derivative and its intrinsic derivative in that space:

**Example.** *(LFD and intrinsic derivative for $H^{\mathcal{E}^G}$) By definition of the linear derivative $\frac{\partial H_{\lambda_{\mathcal{E}^G}}}{\partial \eta}$ of $H_{\lambda_{\mathcal{E}^G}}$ on $\mathcal{P}(\mathcal{E}^G)$ and the form we know it takes (see the examples from Appendix D.1), we see that, whenever $\mu^{\mathcal{E}^G}, \nu^{\mathcal{E}^G} \ll \lambda_{\mathcal{E}^G}$, we have:*

$$\lim_{h \to 0} \frac{H_{\lambda_{\mathcal{E}^G}}((1-h)\mu^{\mathcal{E}^G} + h\nu^{\mathcal{E}^G}) - H_{\lambda_{\mathcal{E}^G}}(\mu^{\mathcal{E}^G})}{h} = \int_{\mathcal{E}^G} \frac{\partial H_{\lambda_{\mathcal{E}^G}}}{\partial \eta}(\mu^{\mathcal{E}^G}, x)d(\nu^{\mathcal{E}^G} - \mu^{\mathcal{E}^G})(x)$$

$$= \int_{\mathcal{E}^G} \left( \log \left( \frac{d\mu^{\mathcal{E}^G}}{d\lambda_{\mathcal{E}^G}}(x) \right) + C \right) d(\nu^{\mathcal{E}^G} - \mu^{\mathcal{E}^G})(x)$$

$$= \int_{\mathcal{Z}} \left( \log \left( \frac{d\mu^{\mathcal{E}^G}}{d\lambda_{\mathcal{E}^G}}(P_{\mathcal{E}^G}.z) \right) + C \right) d(\nu - \mu)(z),$$

*which yields that $\frac{\partial H^{\mathcal{E}^G}}{\partial \mu}(\mu, z) = \log \left( \frac{d\mu^{\mathcal{E}^G}}{d\lambda_{\mathcal{E}^G}}(P_{\mathcal{E}^G}.z) \right) + C$ (for $C$ an appropriate constant). A formal expression for the intrinsic derivative follows, which is given by :*

$$D_\mu H^{\mathcal{E}^G}(\mu, z) = \frac{1}{\frac{d\mu^{\mathcal{E}^G}}{d\lambda_{\mathcal{E}^G}}(z)} P_{\mathcal{E}^G}^T \nabla_z \left[ \frac{d\mu^{\mathcal{E}^G}}{d\lambda_{\mathcal{E}^G}} \right] (P_{\mathcal{E}^G}.z)$$

## D.2 Expression for the WGF of the regularized population risk

In the case of the **regularized** population risk functional $R^{\tau,\beta} : \mathcal{P}(\mathcal{Z}) \to \mathbb{R}$, we can explicitly write its intrinsic derivative. Consider a slightly more *general* functional, denoted by $R_\nu^{\tau,\beta}$, where the entropy is calculated against a **Gibbs measure** $\nu \lll \lambda$ such that $\nu(dz) = e^{-U(z)}\lambda(dz)$ for some function $U : \mathcal{Z} \to \mathbb{R}$ (as in [38]). We have $\forall \mu \in \mathcal{P}(\mathcal{Z})$ s.t. $\mu \lll \nu$, $\forall z \in \mathcal{Z}$:

$$D_\mu R_\nu^{\tau,\beta}(\mu, z) = D_\mu R(\mu, z) + \tau \nabla_z r(z) + \beta \nabla_z U(z) + \beta \left( \frac{1}{\mu(z)} \nabla_z \mu(z) \right),$$

so that **WGF**$(R_\nu^{\tau,\beta})$ as in definition 4 reads:

$$\partial_t \mu_t = \varsigma(t) \left[ \mathrm{div} \left( D_\mu R_\nu^{\tau,\beta}(\mu_t, \cdot) \, \mu_t \right) \right] = \varsigma(t) \left[ \mathrm{div} \left( (D_\mu R(\mu_t, \cdot) + \tau \nabla_z r + \beta \nabla_z U) \, \mu_t \right) + \beta \Delta \mu_t \right].$$

We recover the expression for **WGF**$(R^{\tau,\beta})$ in Equation (4) by considering $U \equiv 0$:

$$\partial_t \mu_t = \varsigma(t) \left[ \mathrm{div} \left( (D_\mu R(\mu_t, \cdot) + \tau \nabla_z r) \, \mu_t \right) + \beta \Delta \mu_t \right]. \tag{10}$$

We can see that Equation (4) corresponds to a Fokker-Planck equation, which can be interpreted in terms of a **non-linear SDE** system, representing the behaviour of the *type* parameter: the **McKean-Vlasov SDE** [11, 54, 70] (also known as *the Mean Field Langevin Dynamics (MFLD)* in the NN literature). In the case of $R^{\tau,\beta}$ it reads:

$$dZ_t = \varsigma(t) \left[ -\left( D_\mu R(\mu_t, Z_t) + \tau \nabla_\theta r(Z_t) \right) dt + \sqrt{2\beta} dB_t \right] \quad \text{with} \quad \mu_t = \mathbf{Law}(Z_t), \tag{11}$$

where $(B_t)_{t \geq 0}$ is a $D$-dimensional standard Brownian Motion. It is indeed standard to check (by applying Itô's formula to $\varphi(Z_t, t)$ for $\varphi$ a smooth function, and taking expectation) that $\mu_t = \mathbf{Law}(Z_t)$ is a weak solution to (10). Under mild regularity conditions, both formulations are equivalent. See [11, 54, 70] for details. The previous correspondence also holds true when $\beta = 0$ (in which case (11) is an ODE). The process (11) or variants of it will prove useful to establish some of the relevant results of the paper.

## D.3 Global convergence in the regularized case

For the example we just presented of the entropy-regularized population risk, multiple authors (see [12, 15, 38, 57, 69] among many others) have studied the properties of **WGF**$(R^{\tau,\beta})$, particularly, the global convergence results that can be obtained. For instance, consider the following results from [38] (where we look at $R_\nu^{\tau,\beta}$ for generality). First, define:

**Definition 10.** *We say that a functional $R : \mathcal{P}_p(\mathcal{Z}) \to \mathbb{R}$ is of class $\mathcal{C}^1$ if $\frac{\partial R}{\partial \mu}(\mu, \cdot)$ is well defined and bounded for every $\mu \in \mathcal{P}_p(\mathcal{Z})$, and the function $(\mu, z) \in \mathcal{P}_p(\mathcal{Z}) \times \mathcal{Z} \mapsto \frac{\partial R}{\partial \mu}(\mu, z)$ is **continuous**.*

Now, from [15, 38], we get the following key result. We include the proof for completeness:

**Lemma 3** (as in [38, 15]). *Assume that $R : \mathcal{P}_p(\mathcal{Z}) \to \mathbb{R}$ is **convex** and of class $\mathcal{C}^1$. Then, for any $\mu, \mu' \in P_p(\mathcal{Z})$, we have:*

$$R(\mu') - R(\mu) \geq \int_{\mathcal{Z}} \frac{\partial R}{\partial \mu}(\mu, z) d(\mu' - \mu)(z)$$

*Proof of Lemma 3 (from [38]).* Define $\mu^\epsilon := (1 - \epsilon)\mu + \epsilon\mu'$. Since $R$ is convex, we have

$$\epsilon \left( R(\mu') - R(\mu) \right) \geq R(\mu^\epsilon) - R(\mu) = \int_0^\epsilon \int_{\mathcal{Z}} \frac{\partial R}{\partial \mu}(\mu^s, z) d(\mu' - \mu)(dz) \, ds.$$

Since the map $s \in [0, 1] \mapsto \mu^s$ is continuous, it is of **compact image** (denoted $[\mu, \mu']$). In particular, as $\frac{\partial R}{\partial \mu}$ is continuous and bounded on its second argument, we get that, it is bounded on $[\mu, \mu'] \times \mathcal{Z}$. The dominated convergence and *Lebesgue differentiation* theorems (as $\varepsilon \to 0$) allow us to conclude. $\square$

Consider now the following assumption:

**Assumption 2** (As in [38]). *$U : \mathcal{Z} \to \mathbb{R}$ is assumed to be $\mathcal{C}^\infty$, with $\nabla U$ **Lipschitz** continuous, and such that $\exists C_U > 0$, $\exists C'_U \in \mathbb{R}$ such that $\forall x \in \mathcal{Z}$: $\nabla U(x) \cdot x \geq C_U \|x\|^2 + C'_U$. When required, we will also assume that $r : \mathcal{Z} \to \mathbb{R}$ satisfies these conditions.*

*Notice that these conditions imply that $\exists 0 \leq C' \leq C$ s.t. $\forall x \in \mathcal{Z}$, $C'\|x\|^2 - C \leq U(x) \leq C(1 + \|x\|^2)$ (i.e. $U$ has quadratic growth) and $|\Delta U(x)| \leq C$*

Since $R_\nu^{\tau,\beta}$ includes an **entropy term**, it guarantees **strict convexity, weak lower semicontinuity and compact sublevel sets** (see e.g. [38] or [26]). On the other hand, assumption 2 implies that $U$ (or $r$) will have quadratic growth. Namely, we get (see [38] for a detailed proof):

**Proposition 14** (Existence and Uniqueness of the minimizer (regularized case)). *Let $R$ be **convex**, of class $\mathcal{C}^1$ and bounded from below. Let $\nu$ be the Gibbs measure with potential $U$. Then, $R_\nu^{\tau,\beta}$ has a **unique minimizer**, $\mu^{*,\tau,\beta,\nu} \in \mathcal{P}(\mathcal{Z})$, absolutely continuous with respect to Lebesgue measure $\lambda$. When either $U$ or $r$ satisfies assumption 2, this minimizer also belongs to $\mathcal{P}_2(\mathcal{Z})$.*

For establishing global convergence results further assumptions are requred

**Assumption 3** (Assumptions for well definedness (from [12] and [38])). *Assume that the intrinsic derivative $D_\mu R : \mathcal{P}(\mathcal{Z}) \times \mathcal{Z} \to \mathcal{Z}$ of the functional $R : \mathcal{P}(\mathcal{Z}) \to \mathbb{R}$ exists and satisfies either one of the following:*

1. *(From [38]). Assume:*
   - *$D_\mu R$ is bounded and Lipschitz continuous, i.e. $\exists C_R > 0$ s.t. $\forall z, z' \in \mathcal{Z}$, $\forall \mu, \mu' \in \mathcal{P}_2(\mathcal{Z})$,*
     $$|D_\mu R(\mu, z) - D_\mu R(\mu', z')| \leq C_R[|z - z'| + W_2(\mu, \mu')]$$
   - *$\forall \mu \in \mathcal{P}(\mathcal{Z})$, $D_\mu R(\mu, \cdot) \in C^\infty(\mathcal{Z})$.*
   - *$\nabla D_\mu R : \mathcal{P}(\mathcal{Z}) \times \mathcal{Z} \to \mathcal{Z} \times \mathcal{Z}$ is jointly continuous.*

2. *(From [12], who relax some differentiability conditions at the cost of boundedness assumptions; this allows them to avoid altogether the coercivity condition from assumption 2, which is used in [38]):*
   - *$\forall x \in \mathcal{Z}, \forall m, m' \in \mathcal{P}_2(\mathcal{Z}), |D_\mu R(m, x) - D_\mu R(m', x)| \leq M_{mm}^R W_1(m, m')$ for some constant $M_{mm}^R \geq 0$ (i.e. it is lipschitz on the measure argument).*
   - *Suppose that*
     $$\sup_{\mu \in \mathcal{P}_2(\mathcal{Z})} \sup_{x \in \mathcal{Z}} |\nabla D_\mu R(\mu, x)| \leq M_{mx}^R$$
   *for some constant $M_{mx}^R \geq 0$ i.e. $\nabla D_\mu R(\mu, x)$ is uniformly bounded.*

This allows to establish a traditional global convergence result from the MF Theory of NNs:

**Theorem 7** (from [12] and [38]). *Let $\mu_0 \in \mathcal{P}_2(\mathcal{Z})$, and let assumption 2 and 3 hold; then:*

$$\forall t > 0, \quad \frac{d}{dt}(R_\nu^{\tau,\beta}(\mu_t)) = -\varsigma(t) \int_\mathcal{Z} \left| D_\mu R(\mu_t, z) + \tau \nabla r(z) + \beta \frac{\nabla u_t}{u_t}(z) + \beta \nabla U(z) \right|^2 d\mu_t(z)$$

*where $u_t$ denotes the density of $\mu_t := \textbf{Law}(X_t)$, the solution to equation (4). i.e. following the WGF makes the regularized risk decrease at a known rate. This is known as the **energy dissipation equation**.*

**Remark.** *Notice that this equation can be rewritten using the **Fisher divergence** (or relative Fisher Information) between two measures. This quantity is defined as:*

$$I(\mu\|\nu) := \int_\mathcal{Z} \left\| \nabla \log(\frac{d\mu}{d\nu}(z)) \right\|^2 d\mu(z)$$

*Then, almost by definition, we get:*

$$\frac{d}{dt}(R_\nu^{\tau,\beta}(\mu_t)) = -\beta^2 \varsigma(t) I(\mu_t \| \hat{\mu}_t)$$

*This allows us to characterize the stationary points of the dynamic explicitly, as done in [12, 38, 57, 69].*

Theorem 7 implies that the WGF converges to the unique global optimizer of the regularized problem:

**Theorem 8** (from [38]). *Let $R$ be **convex**, bounded from below and $\mathcal{C}^1$; also assume that assumption 2 and 3 hold. Consider $\mu_0 \in \cup_{p>2}\mathcal{P}_p(\mathcal{Z})$ and let $(\mu_t)_{t\geq 0}$ be the **WGF**$(R_\nu^{\tau,\beta})$ starting from $\mu_0$. Then, the equation has a stationary distribution, $\mu_\infty$, that satisfies:*

$$\mu_\infty := \arg \min_{\mu \in \mathcal{P}(\mathcal{Z})} R_\nu^{\tau,\beta}(\mu) \quad and \quad \lim_{t\to\infty} W_2(\mu_t, \mu_\infty) = 0$$

**Remark.** *Global Convergence Results such as Theorem 7 or Theorem 8 have been established as early as in [53] (for the quadratic loss). However, settings such as those of [12, 38, 57, 69, 15] are of notorious interest to establish essentially the same results under fundamentally more general assumptions.*

Making further *technical* assumptions on our regularized functionals leads to better *convergence results*. Namely, consider the following definition:

**Definition 11.** *We say $\mu \in \mathcal{P}(\mathcal{Z})$ satisfies the Log-Sobolev Inequality with constant $\vartheta > 0$ (in short, LSI($\vartheta$)), if for any $\nu \in \mathcal{P}(\mathcal{Z})$ such that $\nu \lll \mu$, we have:*

$$D(\nu||\mu) := \int_{\mathcal{Z}} \log(\frac{d\nu}{d\mu}(z))d\nu(z) \leq \frac{1}{2\vartheta} \int_{\mathcal{Z}} \left\| \nabla \log(\frac{d\nu}{d\mu}(z)) \right\|^2 d\nu(z) =: \frac{1}{2\vartheta} I(\nu||\mu)$$

*where $D(\nu||\mu)$ is the KL divergence and $I(\mu||\nu)$ is the **Fisher divergence**.*

(see [3, 58] for background on functional inequalities and [15, 12, 57, 69] for applications of it to the NN context). In our setting, as done by most authors in recent years to achieve the desired global convergence results, the following 'uniform-LSI' on the functional $R : \mathcal{P}(\mathcal{Z}) \to \mathbb{R}$ is assumed to hold:

**Assumption 4** (**Uniform LSI** from [15, 12, 57, 69]). *There exists $\vartheta > 0$ such that $\forall \mu \in \mathcal{P}_2(\mathcal{Z})$, $\hat{\mu}$ satisfies LSI($\vartheta$). Here $\hat{\mu}$ is the probability measure with density w.r.t. $\lambda$ given by (slightly abusing notation, and considering $U$ the potential of a Gibbs measure $\nu$ used in the entropy, which is $0$ if $\nu = \lambda$):*

$$\hat{\mu}(z) \propto \exp\left(-\frac{1}{\beta}\frac{\partial R}{\partial \mu}(\mu, z) - \frac{\tau}{\beta}r(z) - U(z)\right)$$

**Remark.** *This **LSI** is a recurrent element in the literature of WGF and Optimal Transport in general. In particular, it implies (see [3]) Poincaré Inequality: $\forall \phi \in \mathcal{C}_b^1(\mathcal{Z})$, $\mathrm{Var}_{\hat{\mu}}(\phi) \leq \frac{1}{2\vartheta}\mathbb{E}_{\hat{\mu}}[|\nabla\phi|^2]$, and ([58]) the Talagrand's $T_2$-transport inequality as well: $\forall \nu \in \mathcal{P}_2(\mathcal{Z})$, $\vartheta W_2^2(\nu, \hat{\mu}) \leq D(\nu||\hat{\mu})$*

Beyond the *characterization* of the decay (from [38]), we have the following guarantee:

**Theorem 9** (from [12, 15]). *Let $R$ be convex, $\mathcal{C}^1$ and bounded from below, and let assumptions 3 and 4 hold. Then, if for some $t_0 \geq 0$, $\mu_{t_0}$ has **finite entropy** and **finite second moment**; then $\forall t \geq t_0$,*

$$D(\mu_t||\mu_\infty) \leq R_\nu^{\tau,\beta}(\mu_t) - R_\nu^{\tau,\beta}(\mu_\infty) \leq (R_\nu^{\tau,\beta}(\mu_{t_0}) - R_\nu^{\tau,\beta}(\mu_\infty))e^{-2\beta\vartheta \int_{t_0}^t \varsigma(s)ds}$$

*where $\mu_\infty = \mu^{\tau,\beta,\nu} = \arg\min_{\mu \in \mathcal{P}(\mathcal{Z})} R_\nu^{\tau,\beta}(\mu)$. That is, the value function following the WGF thus converges exponentially fast to the optimum value of the problem, and this implies an exponential convergence in relative entropy.*

One thus recovers, under the right technical assumptions, a version of Theorem 4 from [53] and actually a quantitative improvement of it. By Talagrand's inequality this also implies *exponential $W_2$* convergence of $\mu_t$ to $\mu_\infty$. We note that the result in [12] is established in the setting with $\tau = 0, \beta = 1$ and $\varsigma \equiv 1$; however, one can show that the result holds as stated by using standard arguments.

## D.4 Conditions for well-posedness of WGF

Most of the technical conditions that will be here presented are directly taken from both [16] and [22]. We only adapt them slightly to fit into our notation.

Regarding the existence of **weak** solutions to the WGF presented in equation (4), [16] are able to guarantee it under the following assumptions (more general assumptions might be sought in [1, 63], but these are relatively standard in the MF context):

**Assumption 5** (Assumptions for existence and uniqueness of the WGF solutions (taken from [16]))**.**
*Consider a setting as described in proposition 12, with $R(\mu) := L(\langle \Phi, \mu \rangle) + V(\mu)$, with $V(\mu) = \tau \int_{\mathcal{Z}} r d\mu$.*

1. *Let $\mathcal{Z}$ to be the closure of a convex open set within some finite-dimensional euclidean space.*

2. *Let $L : \mathcal{H} \to \mathbb{R}^+$ be differentiable, with a differential $dL$ that is **Lipschitz on bounded sets and bounded on sublevel sets**.*

3. *Let $\Phi : \mathcal{Z} \to \mathcal{H}$ be differentiable and $V : \mathcal{Z} \to \mathbb{R}^+$ be **semiconvex** (i.e. $\exists \lambda \in \mathbb{R} : V + \lambda |\cdot|^2$ is convex).*

4. *There exists a family $(Q_r)_{r>0}$ of **nested nonempty closed convex subsets** of $\mathcal{Z}$ such that:*

    (a) $\{u \in \Omega; dist(u, Q_{r'}) \leq r\} \subset Q_{r+r'}$ *for all $r, r' > 0$,*
    (b) $\Phi$ *and $V$ are bounded, and $d\Phi$ is Lipschitz on each $Q_r$*
    (c) $\exists C_1, C_2 > 0$ *such that $\sup_{u \in Q_r}(\|d\Phi(u)\| + \|\partial V(u)\|) \leq C_1 + C_2 r$ for all $r > 0$, where $\|\partial V(u)\|$ stands for the maximal norm of an element in $\partial V(u)$.*

On the other hand, [22] are able to prove (based on Theorem 1.1. from [70]) the existence of **strong solutions with pathwise uniqueness** for **McKean-Vlasov SDE** given by

$$dZ_t = b(t, Z_t, \mu_t)dt + \Sigma(t, Z_t, \mu_t)dB_t$$

where $b$ and $\Sigma$ satisfies the conditions of **B2** (presented below) and for all $t \geq 0$, $\mu_t = \mathbf{Law}(Z_t) \in \mathcal{P}_2(\mathbb{R}^D)$, $(B_t)_{t \geq 0}$ is an $r$-dimensional Brownian motion (with $r \in \mathbb{N}^*$ potentially different from $D \in \mathbb{N}^*$), and $Z_0$ has the (fixed) law $\mu^0 \in \mathcal{P}_2(\mathbb{R}^D)$. For this, consider the following technical assumptions (**B1** and **B2**) which have been taken directly from [22]:

**Assumption 6** (Assumptions for the existence and uniqueness of solutions in [22])**.** *Consider:*

**B1.** *There exist a measurable function $g : \mathbb{R}^D \times \mathcal{W} \to \mathbb{R}$, $M_1 \geq 0$ and $\mu_0 \in \mathcal{P}_2(\mathbb{R}^D)$ such that for any $N \in \mathbb{N}$, the following hold.*

   (a) *For any $w_1, w_2 \in \mathbb{R}^D$ and $z \in \mathcal{W}$ we have*

   $$\|g(w_1, z) - g(w_2, z)\| \leq \zeta(z)\|w_1 - w_2\|, \text{ and } \|g(w_1, z)\| \leq \zeta(z)$$

   *with $\int_{\mathcal{W}} \zeta^2(z) d\pi_{\mathcal{W}}(z) < +\infty$*
   (b) *$b_N \in C(\mathbb{R}_+ \times \mathbb{R}^D \times \mathcal{P}_2(\mathbb{R}^D), \mathbb{R}^D)$ and $\Sigma_N \in C(\mathbb{R}_+ \times \mathbb{R}^D \times \mathcal{P}_2(\mathbb{R}^D), \mathbb{R}^{D \times r})$.*
   (c) *For any $w_1, w_2 \in \mathbb{R}^D$ and $\mu_1, \mu_2 \in \mathcal{P}_2(\mathbb{R}^D)$*

   $$\sup_{t \geq 0} \left\{ \|b_N(t, w1, \mu1) - b_N(t, w2, \mu2)\| + \|\Sigma_N(t, w_1, \mu_1) - \Sigma_N(t, w_2, \mu_2)\| \right\}$$
   $$\leq M_1 \left( \|w_1 - w_2\| + \left( \int_{\mathcal{W}} \int_{\mathbb{R}^D} |\langle g(\cdot, z), \mu_1 \rangle - \langle g(\cdot, z), \mu_2 \rangle|^2 d\pi_{\mathcal{W}}(z) \right)^{1/2} \right),$$
   $$\sup_{t \geq 0} \{\|b_N(t, 0, \mu_0)\| + \|\Sigma_N(t, 0, \mu_0)\|\} \leq M_1.$$

**B2.** *There exist $M_2 \geq 0$, $\kappa > 0$, $b \in C(\mathbb{R}_+ \times \mathbb{R}^D \times \mathcal{P}_2(\mathbb{R}^D), \mathbb{R}^D)$ and $\Sigma \in C(\mathbb{R}_+ \times \mathbb{R}^D \times \mathcal{P}_2(\mathbb{R}^D), \mathbb{R}^{D \times r})$ such that*

   $$\sup_{t \geq 0, w \in \mathbb{R}^D, \mu \in \mathcal{P}_2(\mathbb{R}^D)} \{\|b_N(t, w, \mu) - b(t, w, \mu)\| + \|\Sigma_N(t, w, \mu) - \Sigma(t, w, \mu)\|\} \leq M_2 N^{-\kappa}.$$

**Proposition 15** (Proposition 11 in [22])**.** *Assuming **B1** and **B2**. Given $\mu^0 \in \mathcal{P}_2(\mathbb{R}^D)$ as a fixed initial condition; then, there exists an $(\mathcal{F}_t)_{t \geq 0}$-adapted process $(Z_t)_{t \geq 0}$ that is the **unique (pathwise) strong solution** of the McKean-Vlasov SDE:*

$$dZ_t = b(t, Z_t, \mu_t)dt + \Sigma(t, Z_t, \mu_t)dB_t$$

*Additionally, it satisfies for each $T \geq 0$: $\sup_{t \in [0,T]} \mathbb{E}[\|Z_t\|^2] < \infty$*

# E  Proofs and discussions of main results

## E.1  Proofs of Section 3.1

*Proof of Proposition 1.* By definition of the symmetrization operator, we know that $\forall x \in \mathcal{X}$:

$$(\mathcal{Q}_G \Phi_\mu)(x) = \int_G \hat{\rho}_{g^{-1}} \Phi_\mu(\rho_g x) d\lambda_G(g)$$

For $g \in G$, since $\sigma_*$ is equivariant and $M_g$ is invertible, we can write:

$$\Phi_\mu(\rho_g x) = \langle \sigma_*(\rho_g x, \cdot), \mu \rangle = \langle \sigma_*(\rho_g x, \cdot), M_g \#(M_g^{-1} \# \mu) \rangle = \hat{\rho}_g \langle \sigma_*(x, \cdot), M_g^{-1} \# \mu \rangle$$

where we've used Proposition 7 in the last equality. In turn, we can write (via the inversion-invariance of $\lambda_G$):

$$(\mathcal{Q}_G \Phi_\mu)(x) = \int_G \hat{\rho}_{g^{-1}} \hat{\rho}_g \langle \sigma_*(x, \cdot), M_g^{-1} \# \mu \rangle d\lambda_G(g) = \int_G \langle \sigma_*(x, \cdot), M_{g^{-1}} \# \mu \rangle d\lambda_G(g)$$

$$= \int_G \langle \sigma_*(x, \cdot), M_g \# \mu \rangle d\lambda_G(g) = \langle \sigma_*(x, \cdot), \mu^G \rangle = \Phi_{\mu^G}(x)$$

$\square$

As mentioned in Section 3.1, a simple case where we will have $\Phi_{\mu^G} = \Phi_{\mu^{\mathcal{E}G}}$ is when $\sigma_*$ is *linear*:

**Proposition 16.** *If $\sigma_* : \mathcal{X} \times \mathcal{Z} \to \mathcal{Y}$ is jointly equivariant and $\pi_\mathcal{X}$-a.s. $\forall x \in \mathcal{X}$, $[z \mapsto \sigma_*(x; z)]$ is a bounded linear operator, then, for any $\mu \in \mathcal{P}(\mathcal{Z})$: $\Phi_{\mu^G} = \langle \sigma_*, \mu^G \rangle = \langle \sigma_*, \mu^{\mathcal{E}G} \rangle = \Phi_{\mu^{\mathcal{E}G}}$.*

*Proof of Proposition 16.* A straightforward computation yields (using Fubini's theorem and the linearity of integrals and $\sigma_*$), $\forall \mu \in \mathcal{P}(\mathcal{Z})$, $\forall x \in \mathcal{X}$ ($\pi_\mathcal{X}$-a.s.):

$$\langle \sigma_*(x, \cdot), \mu^G \rangle = \int_G \int_\mathcal{Z} \sigma_*(x, M_g.z) d\mu(z) d\lambda_G(g) = \int_\mathcal{Z} \int_G \sigma_*(x, M_g.z) d\lambda_G(g) d\mu(z)$$

$$= \int_\mathcal{Z} \sigma_*(x, \int_G M_g.z \, d\lambda_G(g)) d\mu(z) = \int_\mathcal{Z} \sigma_*(x, P_{\mathcal{E}G}.z) d\mu(z) = \langle \sigma_*(x, \cdot), \mu^{\mathcal{E}G} \rangle$$

$\square$

**Example.** *Any usual linear model written in terms of some feature function $\vartheta : \mathcal{X} \to \mathcal{Z}$ (where $\mathcal{Z}$ is possibly an Reproducing Kernel Hilbert Space) enters this framework, by defining: $\sigma_*(x, z) = \langle z, \vartheta(x) \rangle$. This won't satisfy Assumption 1, since it is not bounded; but it still serves as an illustration.*

## E.2  Proofs of results in Section 3.2

### E.2.1  Proof of Proposition 2

Now consider, as a shorthand notation, $\forall x \in \mathcal{X}$, $\forall y \in \mathcal{Y}$ the functional $L_{x,y} : \mathcal{P}(\mathcal{Z}) \to \mathbb{R}$ given by $\forall \mu \in \mathcal{P}(\mathcal{Z})$: $L_{x,y}(\mu) = \ell(\Phi_\mu(x), y)$. The following lemma that shall be useful for later stages.

**Lemma 4.** *Let $\sigma_*$ be jointly equivariant and $\ell$ be invariant. Then, $\forall g \in G$, $\forall x \in \mathcal{X}$, $\forall y \in \mathcal{Y}$, $\forall \mu \in \mathcal{P}(\mathcal{Z})$,*

$$L_{\rho_g.x, \hat{\rho}_g.y}(M_g \# \mu) = L_{x,y}(\mu)$$

*Equivalently, the map $L : \mathcal{P}(\mathcal{Z}) \to L^2(\mathcal{X} \times \mathcal{Y}, \pi)$ given by $L(\mu) \mapsto [(x, y) \mapsto L_{x,y}(\mu)]$ is equivariant (under the appropiate[9] G-actions).*

*Proof of Lemma 4.* Using the joint equivariance of $\sigma_*$ (via proposition 7) and the invariance of $\ell$, a straightforward computation yields, for all $x \in \mathcal{X}$, $y \in \mathcal{Y}$, and $g \in G$:

$$L_{\rho_g.x, \hat{\rho}_g.y}(M_g \# \mu) = \ell(\langle \sigma_*(\rho_g.x, \cdot), M_g \# \mu \rangle, \hat{\rho}_g.y)$$

$$= \ell(\hat{\rho}_g.\langle \sigma_*(x, \cdot), \mu \rangle, \hat{\rho}_g.y)$$

$$= \ell(\langle \sigma_*(x, \cdot), \mu \rangle, y) = L_{x,y}(\mu)$$

$\square$

---

[9]In this case, $\forall g \in G$, let $g.\mu = M_g \# \mu$ and $g.f = f^g$ given by $\forall x \in \mathcal{X}$, $\forall y \in \mathcal{Y}$, $f^g(x, y) = f(\rho_g^{-1}.x, \hat{\rho}_g^{-1}.y)$

With this we can prove Proposition 2. Notice that we will basically utilize the equivariance properties of $\sigma_*$ and $\ell$ in Assumption 1 (the convexity of the functions comes directly from the convexity of $R$ when $\ell$ is convex, together with the linearity of $(\cdot)^G$, $(\cdot)^{\mathcal{E}^G}$ and $\int_G (\cdot) d\lambda_G$.).

*Proof of Proposition 2.* We can readily see that: $R^{EA}(\mu) = \mathbb{E}_\pi \left[ \ell \left( \Phi_{\mu^{\mathcal{E}^G}}(X), Y \right) \right] = R(\mu^{\mathcal{E}^G})$.

On the other hand, as $\sigma_*$ is jointly equivariant, from Proposition 1, we have:

$$R^{FA}(\mu) = \mathbb{E}_\pi \left[ \ell \left( \mathcal{Q}_G(\Phi_\mu)(X), Y \right) \right] = \mathbb{E}_\pi \left[ \ell \left( \Phi_{\mu^G}(X), Y \right) \right] = R(\mu^G)$$

Next, using Lemma 4, Fubini's theorem and the inversion-invariance of $\lambda_G$, we get:

$$
\begin{aligned}
R^{DA}(\mu) &= \mathbb{E}_\pi \left[ \int_G \ell \left( \Phi_\mu(\rho_g.X), \hat{\rho}_g.Y \right) d\lambda_G(g) \right] = \mathbb{E}_\pi \left[ \int_G L_{\rho_g.X, \hat{\rho}_g.Y}(\mu) d\lambda_G(g) \right] \\
&= \int_G \mathbb{E}_\pi \left[ L_{\rho_g.X, \hat{\rho}_g.Y}(\mu) \right] d\lambda_G(g) = \int_G \mathbb{E}_\pi \left[ L_{\rho_g.X, \hat{\rho}_g.Y}(M_g \# M_g^{-1} \# \mu) \right] d\lambda_G(g) \\
&= \int_G \mathbb{E}_\pi \left[ L_{X,Y}(M_{g^{-1}} \# \mu) \right] d\lambda_G(g) = \int_G \mathbb{E}_\pi \left[ L_{X,Y}(M_g \# \mu) \right] d\lambda_G(g) \\
&= \int_G R(M_g \# \mu) d\lambda_G(g) = R^G(\mu)
\end{aligned}
$$

From these expressions we can quickly verify that $R^{DA}$, $R^{FA}$ and $R^{EA}$ are invariant. Namely, by Lemma 1, for $g \in G$ and $\mu \in \mathcal{P}(\mathcal{Z})$ we have: $R^{FA}(M_g \# \mu) = R((M_g \# \mu)^G) = R(\mu^G) = R^{FA}(\mu)$, and: $R^{EA}(M_g \# \mu) = R((M_g \# \mu)^{\mathcal{E}^G}) = R(\mu^{\mathcal{E}^G}) = R^{EA}(\mu)$. On the other hand, the right-invariance of $\lambda_G$ implies:

$$R^{DA}(M_g \# \mu) = \int_G R((M_h \# (M_g \# \mu))) d\lambda_G(h) = \int_G R((M_{\tilde{h}} \# \mu)) d\lambda_G(\tilde{h}) = R^{DA}(\mu).$$

We can also see that whenever $R$ is invariant, we have that, for $\mu \in \mathcal{P}(\mathcal{Z})$ and $g \in G$, $R(M_g \# \mu) = R(\mu)$, so that: $R^{DA}(\mu) = R^G(\mu) = \int_G R(M_g \# \mu) d\lambda_G(g) = \int_G R(\mu) d\lambda_G(g) = R(\mu)$.

Also, if $\mu \in \mathcal{P}^G(\mathcal{Z})$, we have, for all $g \in G$, $\mu = \mu^G = M_g \# \mu$, so that: $R^{DA}(\mu) = \int_G R(M_g \# \mu) d\lambda_G(g) = \int_G R(\mu) d\lambda_G(g) = R(\mu) = R(\mu^G) = R^{FA}(\mu)$.

Finally, we verify that our population risk $R : \mathcal{P}(\mathcal{Z}) \to \mathbb{R}$ is invariant whenever $\pi \in \mathcal{P}^G(\mathcal{X} \times \mathcal{Y})$. Indeed, $\forall g \in G$ and $\forall \mu \in \mathcal{P}(\mathcal{Z})$, by the invariance of $\ell$ and $\pi$, together with Proposition 7 from the equivariance of $\sigma_*$, we get:

$$
\begin{aligned}
R(M_g \# \mu) &= \mathbb{E}_\pi \left[ \ell(\langle \sigma_*(X; \cdot), M_g \# \mu \rangle, Y) \right] = \mathbb{E}_\pi \left[ \ell(\hat{\rho}_g \langle \sigma_*(\rho_g^{-1} X; \cdot), \mu \rangle, \hat{\rho}_g \hat{\rho}_g^{-1} Y) \right] \\
&= \mathbb{E}_\pi \left[ \ell(\langle \sigma_*(\rho_g^{-1} X; \cdot), \mu \rangle, \hat{\rho}_g^{-1} Y) \right] = \mathbb{E}_\pi \left[ \ell(\langle \sigma_*(X; \cdot), \mu \rangle, Y) \right] = R(\mu)
\end{aligned}
$$

That is, $R$ is invariant. $\qquad\square$

Notice that the equivariance of the data distribution $\pi$ can also make the *regularized* population risk be invariant, under the right choice of $r$. Namely:

**Corollary 6.** *If $R : \mathcal{P}(\mathcal{Z}) \to \mathbb{R}$ and $r : \mathcal{Z} \to \mathbb{R}$ are invariant (in their respective sense), then $R^{\tau,\beta}$ is invariant.*

The result can be proven for $R_\nu^{\tau,\beta}$ with $\nu$ some $G$-invariant measure (such as $\lambda$ for orthogonal representations). Notice that $r(\theta) = \|\theta\|^2$ is an example of invariant function for *orthogonal representations*.

*Proof of Corollary 6.* It is enough to notice that $V(\mu) = \int_{\mathcal{Z}} r(\theta) d\mu(\theta)$ and $H_\nu(\mu) = \int_{\mathcal{Z}} \log(\frac{d\mu}{d\nu}) d\mu$ (with $\mu \lll \nu$) are invariant when $r : \mathcal{Z} \to \mathbb{R}$ and $\nu \in \mathcal{P}(\mathcal{Z})$ are invariant (in their respective sense):

1. For $V$, notice that for $g \in G$:

$$V(M_g \# \mu) = \int_{\mathcal{Z}} r(\theta) d(M_g \# \mu)(\theta) = \int_{\mathcal{Z}} r(M_g \theta) d\mu(\theta) = \int r(\theta) d\mu(\theta) = V(\mu)$$

thanks to the invariance of $r$. i.e. $V$ is invariant.

2. For $H_\nu$, notice that, for $g \in G$, as $\nu$ is invariant, we know that $\frac{d(M_g \# \mu)}{d\nu}(x) = \frac{d\mu}{d\nu}(M_g^{-1} x)$. Therefore:

$$H_\nu(M_g \# \mu) = \int \log\left(\frac{d(M_g \# \mu)}{d\nu}(\theta)\right) d(M_g \# \mu)(\theta)$$

$$= \int \log\left(\frac{d\mu}{d\nu}(M_g^{-1} \theta)\right) d(M_g \# \mu)(\theta)$$

$$= \int \log\left(\frac{d\mu}{d\nu}(M_g^{-1} M_g \theta)\right) d\mu(\theta) = H_\nu(\mu)$$

Which proves that $H_\nu$ is invariant.

We can readily conclude, since: $R^{\tau,\beta}(\mu) = R(\mu) + \tau \int r d\mu + \beta H_\lambda(\mu)$, for all $\mu \in \mathcal{P}(\mathcal{Z})$. $\qquad \square$

### E.2.2 Proof of Proposition 3

In order to prove Proposition 3, we require a version of Jensen's inequality that's suited for our context. Such a result might exist in the literature, but since we couldn't find a complete proof under our assumptions, we provide our own.

**Proposition 17** (**Jensen's Inequality**). *Let $F : \mathcal{P}(\mathcal{Z}) \longrightarrow \mathbb{R}$ be such that Lemma 3 holds. Let $S$ be some measurable space, $\lambda \in \mathcal{P}(S)$ and $s \in S \mapsto \mu_s \in \mathcal{P}(\mathcal{Z})$ a measurable function. Define $\tilde{\mu} \in \mathcal{P}(\mathcal{Z})$ as the intensity measure: $\tilde{\mu} = \int_S \mu_s \, d\lambda(s) \in \mathcal{P}(\mathcal{Z})$. Then, Jensen's inequality holds:*

$$F(\tilde{\mu}) \leq \int_S F(\mu_s) d\lambda(s)$$

*Proof of Proposition 17.* Since Lemma 3 holds, we have that $\forall \mu_1, \mu_2 \in \mathcal{P}(\mathcal{Z})$:

$$F(\mu_1) \geq F(\mu_2) + \int \frac{\partial F}{\partial \mu}(\mu_2, z) d(\mu_1 - \mu_2)(z)$$

Let $\tilde{s} \in S$ be arbitrary and consider $\mu_2 = \tilde{\mu} := \int \mu_s d\lambda(s)$; and $\mu_1 = \mu_{\tilde{s}}$. Then:

$$F\left(\int \mu_s d\lambda(s)\right) \leq F(\mu_{\tilde{s}}) - \int \frac{\partial F}{\partial \mu}\left(\int \mu_s d\lambda(s), z\right) d\left(\mu_{\tilde{s}} - \int \mu_s d\lambda(s)\right)(z)$$

Integrating the inequality with respect to $\lambda$ (on $\tilde{s}$):

$$\int_S F\left(\int \mu_s d\lambda(s)\right) d\lambda(\tilde{s})$$

$$\leq \triangle := \left[\int_S F(\mu_{\tilde{s}}) d\lambda(\tilde{s}) - \int_S \left(\int_{\mathcal{Z}} \frac{\partial F}{\partial \mu}\left(\int_S \mu_s d\lambda(s), \cdot\right) d\left(\mu_{\tilde{s}} - \int_S \mu_s d\lambda(s)\right)\right) d\lambda(\tilde{s})\right]$$

We notice that the LHS doesn't depend on $\tilde{s}$, so that $\int_S F\left(\int \mu_s d\lambda(s)\right) d\lambda(\tilde{s}) = F\left(\int \mu_s d\lambda(s)\right)$. On the other hand, the right-most term in $\triangle$ can be developed as:

$$\star := \int_S \left(\int_{\mathcal{Z}} \frac{\partial F}{\partial \mu}\left(\int_S \mu_s d\lambda(s), \cdot\right) d\left(\mu_{\tilde{s}} - \int_S \mu_s d\lambda(s)\right)\right) d\lambda(\tilde{s})$$

$$= \int_S \left(\int_{\mathcal{Z}} \frac{\partial F}{\partial \mu}\left(\int_S \mu_s d\lambda(s), \cdot\right) d\mu_{\tilde{s}} - \int_{\mathcal{Z}} \frac{\partial F}{\partial \mu}\left(\int_S \mu_s d\lambda(s), \cdot\right) d\left(\int_S \mu_s d\lambda(s)\right)\right) d\lambda(\tilde{s})$$

$$= \int_S \left(\int_{\mathcal{Z}} \frac{\partial F}{\partial \mu}\left(\int \mu_s d\lambda(s), \cdot\right) d(\mu_{\tilde{s}})\right) d\lambda(\tilde{s})$$

$$- \int_S \left(\int_{\mathcal{Z}} \frac{\partial F}{\partial \mu}\left(\int_S \mu_s d\lambda(s), \cdot\right) d\left(\int_S \mu_s d\lambda(s)\right)\right) d\lambda(\tilde{s})$$

Notice that the *linear functional derivative* is chosen in such a way so that it satisfies $\forall \nu \in \mathcal{P}(\mathcal{Z})$, $\int_{\mathcal{Z}} \frac{\partial F}{\partial \mu}(\nu, z) d\nu(z) = 0$. In particular, the second term of the previous expression vanishes. We get that

$$\star = \int_S \left( \int_{\mathcal{Z}} \frac{\partial F}{\partial \mu} \left( \int_S \mu_s d\lambda(s), z \right) d\mu_{\tilde{s}}(z) \right) d\lambda(\tilde{s})$$

But, by definition: $\forall f : \mathcal{Z} \to \mathbb{R}$ integrable,

$$\langle f, \int_S \mu_s d\lambda(s) \rangle = \int_S \langle f, \mu_s \rangle d\lambda(s) = \int_S \left( \int_{\mathcal{Z}} f(z) d\mu_s(z) \right) d\lambda(s)$$

So this is, by definition, and applying the same convention on the definition of the linear functional derivative[10]:

$$\star = \int_{\mathcal{Z}} \frac{\partial F}{\partial \mu} \left( \int \mu_s, z \right) d \left( \int \mu_{\tilde{s}} d\lambda(\tilde{s}) \right)(z) = 0$$

With this, we conclude that $\triangle = \int_S F(\mu_s) d\lambda(s)$, and so, we get that:

$$F(\tilde{\mu}) = F \left( \int_S \mu_s d\lambda(s) \right) \le \int_S F(\mu_s) d\lambda(s)$$

which corresponds to Jensen's inequality. $\qquad\square$

**Remark.** *We believe that the $\mathcal{C}^1$ hypothesis can be lifted. Understanding what happens when equality holds should be of interest both in our context and in more general scenarios.*

Thanks to this *Jensen inequality*, we readily get the following result:

**Corollary 7.** *If $F : \mathcal{P}(\mathcal{Z}) \longrightarrow \mathbb{R}$ is convex, $\mathcal{C}^1$ and invariant, then $\forall \mu \in \mathcal{P}(\mathcal{Z})$: $F(\mu^G) \le F(\mu)$*

*Proof.* Direct from the definition of $(\cdot)^G$ and Proposition 17. $\qquad\square$

With these results in place, we are ready to prove Proposition 3:

*Proof of Proposition 3.* Evidently, since $\mathcal{P}^G(\mathcal{Z}) \subseteq \mathcal{P}(\mathcal{Z})$, we have:

$$\inf_{\mu \in \mathcal{P}^G(\mathcal{Z})} F(\mu) \ge \inf_{\mu \in \mathcal{P}(\mathcal{Z})} F(\mu)$$

For the other inequality, take $(\mu_n)_{n \in \mathbb{N}} \subseteq \mathcal{P}(\mathcal{Z})$ to be an *infimizing sequence* for $F$; i.e. such that $F(\mu_n) \ge F(\mu_{n+1})$ and $F(\mu_n) \xrightarrow[n \to \infty]{} \inf_{\mu \in \mathcal{P}(\mathcal{Z})} F(\mu)$). Such a sequence always exists. By Corollary 7, we have $\forall n \in \mathbb{N}$, $F(\mu_n^G) \le F(\mu_n)$; thus, $\forall n \in \mathbb{N}$:

$$\inf_{\mu \in \mathcal{P}^G(\mathcal{Z})} F(\mu) \le F(\mu_n^G) \le F(\mu_n)$$

Which allows us to infer, by taking $n \to \infty$, that: $\inf_{\mu \in P^G(\mathcal{Z})} F(\mu) \le \inf_{\mu \in P(\mathcal{Z})} F(\mu)$. In turn, we can conclude that:

$$\inf_{\mu \in \mathcal{P}^G(\mathcal{Z})} F(\mu) = \inf_{\mu \in \mathcal{P}(\mathcal{Z})} F(\mu)$$

Notice that if there was some minimizer $\mu_* \in \arg\min_{\mu \in \mathcal{P}(\mathcal{Z})} F(\mu)$, then by Corollary 7 we would also have $\mu_*^G \in \arg\min_{\mu \in \mathcal{P}(\mathcal{Z})} R(\mu)$. Namely, if such a minimizer was **unique**, then it would satisfy: $\mu_* = \mu_*^G \in \mathcal{P}^G(\mathcal{Z})$. That is, the unique solution would be **WI**. $\qquad\square$

---

[10]Notice that the *convention* on the linear functional derivative's definition isn't truly important for the proof, since in the end $\star$ simply corresponds to the term $\int_{\mathcal{Z}} \frac{\partial F}{\partial \mu} \left( \int \mu_s, z \right) d \left( \int \mu_{\tilde{s}} d\lambda(\tilde{s}) \right)(z)$ being substracted to itself.

### E.2.3 Proof of Theorem 2 and Corollary 1

Notice that, under Assumption 1, from Corollary 5 we know $R$ is of class $\mathcal{C}^1$ (as well as convex). This properties actually transfers to the functionals $R^{DA}, R^{FA}$ and $R^{EA}$, as shown by the following result:

**Proposition 18.** *If $R : \mathcal{P}(\mathcal{Z}) \to \mathbb{R}$ is a convex and $\mathcal{C}^1$ functional, then $R^{DA}$, $R^{FA}$ and $R^{EA}$ are convex and $\mathcal{C}^1$ as well, with linear functional derivatives given by:*

$$\frac{\partial R^{DA}}{\partial \mu}(\mu, z) = \int_G \frac{\partial R}{\partial \mu}(M_g \# \mu, M_g.z) d\lambda_G(g), \quad \frac{\partial R^{FA}}{\partial \mu}(\mu, z) = \int_G \frac{\partial R}{\partial \mu}(\mu^G, M_g.z) d\lambda_G(g)$$

$$and \quad \frac{\partial R^{EA}}{\partial \mu}(\mu, z) = \frac{\partial R}{\partial \mu}(\mu^{\mathcal{E}^G}, P_{\mathcal{E}^G}.z)$$

*And intrinsic derivatives given by (when well defined):*

$$D_\mu R^{DA}(\mu, z) = \int_G M_g^T.D_\mu R(M_g \# \mu, M_g.z) d\lambda_G(g)$$

$$D_\mu R^{FA}(\mu, z) = \int_G M_g^T.D_\mu R(\mu^G, M_g.z) d\lambda_G(g) \quad and \quad D_\mu R^{EA}(\mu, z) = P_{\mathcal{E}^G}^T.D_\mu R(\mu^{\mathcal{E}^G}, P_{\mathcal{E}^G}.z)$$

*In particular, from Proposition 17, we have that: $\forall \mu \in \mathcal{P}(\mathcal{Z})$, $R^{FA}(\mu) \leq R^{DA}(\mu)$*

*Proof of Proposition 18.* We can calculate the linear functional derivatives (l.d.f. for short) as follows. Let $\mu, \nu \in \mathcal{P}(\mathcal{Z})$, and consider:

$$\lim_{h \to 0} \frac{R^{DA}((1-h)\mu + h\nu) - R^{DA}(\mu)}{h}$$

$$= \lim_{h \to 0} \frac{\int_G R(M_g \# ((1-h)\mu + h\nu)) d\lambda_G(g) - \int_G R(M_g \# \mu) d\lambda_G(g)}{h}$$

$$= \lim_{h \to 0} \int_G \frac{R((1-h)M_g \# \mu + h M_g \# \nu) - R(M_g \# \mu)}{h} d\lambda_G(g)$$

$$= \int_G \lim_{h \to 0} \frac{R((1-h)M_g \# \mu + h M_g \# \nu) - R(M_g \# \mu)}{h} d\lambda_G(g)$$

$$= \int_G \int_{\mathcal{Z}} \frac{\partial R}{\partial \mu}(M_g \# \mu, z) d(M_g \# \nu - M_g \# \mu)(z) d\lambda_G(g)$$

$$= \int_G \int_{\mathcal{Z}} \frac{\partial R}{\partial \mu}(M_g \# \mu, M_g.z) d(\nu - \mu)(z) d\lambda_G(g)$$

$$= \int_{\mathcal{Z}} \int_G \frac{\partial R}{\partial \mu}(M_g \# \mu, M_g.z) d\lambda_G(g) d(\nu - \mu)(z).$$

We have used Fubini's theorem, which is applicable[11] thanks to the fact that $R$ is of class $\mathcal{C}^1$, and we've used the definition of the linear functional derivative for $R$. Also, we see that (using Fubini's theorem once again, as well as the definition of the linear functional derivative of $R$):

$$\int_{\mathcal{Z}} \int_G \frac{\partial R}{\partial \mu}(M_g \# \mu, M_g.z) d\lambda_G(g) d\mu(z) = \int_G \int_{\mathcal{Z}} \frac{\partial R}{\partial \mu}(M_g \# \mu, z) d(M_g \# \mu)(z) d\lambda_G(g) = 0.$$

We can then identify:

$$\frac{\partial R^{DA}}{\partial \mu}(\mu, z) = \int_G \frac{\partial R}{\partial \mu}(M_g \# \mu, M_g.z) d\lambda_G(g),$$

and, by taking the gradient:

$$D_\mu R^{DA}(\mu, z) = \int_G M_g^T.D_\mu R(M_g \# \mu, M_g.z) d\lambda_G(g).$$

---

[11]In particular, as for any fixed $\mu \in \mathcal{P}(\mathcal{Z})$ the function $g \in G \mapsto M_g \# \mu$ is continuous (thus, of **compact** image), then the function $(g, z) \in G \times \mathcal{Z} \mapsto \frac{\partial R}{\partial \mu}(M_g \# \mu, M_g.z)$ is **bounded**

We analogously calculate the expression for the l.f.d. of $R^{FA}$; let $\mu, \nu \in \mathcal{P}(\mathcal{Z})$:

$$\lim_{h \to 0} \frac{R^{FA}((1-h)\mu + h\nu) - R^{FA}(\mu)}{h} = \lim_{h \to 0} \frac{R((1-h)\mu^G + h\nu^G) - R(\mu^G)}{h}$$

$$= \int_{\mathcal{Z}} \frac{\partial R}{\partial \mu}(\mu^G, z) d(\nu^G - \mu^G)(z)$$

$$= \int_{\mathcal{Z}} \int_G \frac{\partial R}{\partial \mu}(\mu^G, M_g.z) d\lambda_G(g) d(\nu - \mu)(z),$$

and also:

$$\int_{\mathcal{Z}} \int_G \frac{\partial R}{\partial \mu}(\mu^G, M_g.z) d\lambda_G(g) d\mu(z) = \int_G \int_{\mathcal{Z}} \frac{\partial R}{\partial \mu}(\mu^G, z) d(M_g \# \mu)(z) d\lambda_G(g)$$

$$= \int_{\mathcal{Z}} \frac{\partial R}{\partial \mu}(\mu^G, z) d\mu^G(z) = 0.$$

So, by the definition of the lineal functional derivative, we identify:

$$\frac{\partial R^{FA}}{\partial \mu}(\mu, z) = \int_G \frac{\partial R}{\partial \mu}(\mu^G, M_g.z) d\lambda_G(g),$$

and taking the gradient we get:

$$D_\mu R^{FA}(\mu, z) = \int_G M_g^T . D_\mu R(\mu^G, M_g.z) d\lambda_G(g)$$

Lastly, the l.f.d. of $R^{EA}$ is calculated similarly, noticing that:

$$\lim_{h \to 0} \frac{R^{EA}((1-h)\mu + h\nu) - R^{EA}(\mu)}{h} = \lim_{h \to 0} \frac{R(P_{\mathcal{E}^G} \#((1-h)\mu + h\nu)) - R(P_{\mathcal{E}^G}\mu)}{h}$$

$$= \int_{\mathcal{Z}} \frac{\partial R}{\partial \mu}(\mu^{\mathcal{E}^G}, z) d(P_{\mathcal{E}^G} \# \nu - P_{\mathcal{E}^G} \# \mu)(z)$$

$$= \int_{\mathcal{Z}} \frac{\partial R}{\partial \mu}(\mu^{\mathcal{E}^G}, P_{\mathcal{E}^G} z) d(\nu - \mu)(z),$$

and that:

$$\int_{\mathcal{Z}} \frac{\partial R}{\partial \mu}(\mu^{\mathcal{E}^G}, P_{\mathcal{E}^G}.z) d\mu(z) = \int_{\mathcal{Z}} \frac{\partial R}{\partial \mu}(\mu^{\mathcal{E}^G}, z) d(\mu^{P_{\mathcal{E}^G}})(z) = 0.$$

We can thus identify:

$$\frac{\partial R^{EA}}{\partial \mu}(\mu, z) = \frac{\partial R}{\partial \mu}(\mu^{\mathcal{E}^G}, P_{\mathcal{E}^G}.z) \quad \text{and} \quad D_\mu R^{EA}(\mu, z) = P_{\mathcal{E}^G}^T . D_\mu R(\mu^{\mathcal{E}^G}, P_{\mathcal{E}^G}.z)$$

The last remark is direct from proposition 17. $\qquad\square$

With all of these different elements in place, we are ready to prove Theorem 2.

*Proof of Theorem 2.* Under Assumption 1, $R$ is **convex** and of **class** $\mathcal{C}^1$ from Corollary 5; and so are $R^G$, $R^{FA}$ and $R^{EA}$, from Proposition 18. Since Proposition 2 ensures that the latter are always invariant, Proposition 3 implies that $R^{DA}$, $R^{FA}$ and $R^{EA}$ can all be optimized by only considering **weakly equivariant models** (explaining the first and last equalities). The two middle equalities follow directly from Proposition 2, since $R$, $R^{DA}$ and $R^{FA}$ coincide over $\mathcal{P}^G(\mathcal{Z})$. $\qquad\square$

In the case of the quadratic loss, one can employ the properties of $\mathcal{Q}_G$ from [27] to show Corollary 1.

*Proof of Corollary 1.* Notice that, for $\mu \in \mathcal{P}^G(\mathcal{Z})$, $\Phi_\mu$ is a $G$-invariant function (i.e. $\Phi_\mu \in L^2_G(\mathcal{X}, \mathcal{Y}; \pi_\mathcal{X})$). Also, a simple calculation (see e.g. [5, 51]) allows us to write: $R(\mu) = \mathbb{E}_\pi[\|\Phi_\mu(X) - Y\|^2_\mathcal{Y}] = R_* + \mathbb{E}_{\pi_\mathcal{X}}[\|\Phi_\mu(X) - f_*(X)\|^2_\mathcal{Y}] = R_* + \|\Phi_\mu - f_*\|^2_{L^2(\mathcal{X}, \mathcal{Y}; \pi_\mathcal{X})}$ with

$R_* = \mathbb{E}_\pi[\|Y - f^*(X)\|_{\mathcal{Y}}^2]$ being independent of $\mu$. We can thus write (simplifying subscripts for simplicity):

$$R(\mu) = R_* + \|\Phi_\mu - \mathcal{Q}_G.f_* + \mathcal{Q}_G.f_* - f_*\|_{L^2(\mathcal{X},\mathcal{Y};\pi_\mathcal{X})}^2$$
$$= R_* + \|\Phi_\mu - \mathcal{Q}_G.f_*\|_{L^2}^2 + \|(f_*)_G^\perp\|_{L^2}^2 - 2\langle\Phi_\mu - \mathcal{Q}_G.f_*, (f_*)_G^\perp\rangle_{L^2}$$

where $(f_*)_G^\perp := f_* - \mathcal{Q}_G.f_*$. We notice that, since $\Phi_\mu$ and $\mathcal{Q}_G.f_*$ are $G$-equivariant functions, we have that $\langle(\Phi_\mu - \mathcal{Q}_G.f_*), (f_*)_G^\perp\rangle_{L^2} = 0$. That is, for any $\mu \in \mathcal{P}^G(\mathcal{Z})$, we have $R(\mu) = \tilde{R}_* + \|\Phi_\mu - \mathcal{Q}_G.f_*\|_{L^2}^2$, where $\tilde{R}_* := R_* + \|(f_*)_G^\perp\|_{L^2(\mathcal{X},\mathcal{Y};\pi_\mathcal{X})}^2$ is independent of $\mu$ (and doesn't intervene in the optimization). Finally, we get:

$$\inf_{\mu\in\mathcal{P}^G(\mathcal{Z})} R(\mu) = \tilde{R}_* + \inf_{\mu\in\mathcal{P}^G(\mathcal{Z})} \|\Phi_\mu - \mathcal{Q}_G.f_*\|_{L^2(\mathcal{X},\mathcal{Y};\pi_\mathcal{X})}^2$$

$\square$

### E.2.4 Proof of Corollary 2

When $\pi$ is assumed to be equivariant, we can summon our previous results to prove Corollary 2.

*Proof of Corollary 2.* From Proposition 2 we know that equivariant data implies $R : \mathcal{P}(\mathcal{Z}) \to \mathbb{R}$ is invariant, and also that this makes $R = R^{DA}$. We conclude using Theorem 2. $\square$

We can readily extend this to the regularized case by recalling Corollary 6:

**Corollary 8.** *When $R : \mathcal{P}(\mathcal{Z}) \to \mathbb{R}$ and $r : \mathcal{Z} \to \mathbb{R}$ are $G$-invariant, a minimum for the regularized population risk $R^{\tau,\beta}$ can be found within $\mathcal{P}^G(\mathcal{Z})$. When $\beta > 0$ such **WI minimum is unique**.*

*Proof of Corollary 8.* Direct from Corollary 6, together with Proposition 3 (as in Corollary 2). The uniqueness comes from the strict convexity of the entropy term (see proposition 14). $\square$

### E.2.5 Proof of Proposition 4 and Proposition 5

*Proof of Proposition 4.* Consider the group $G = C_4$ acting on $\mathbb{R}^2$ via 90° rotations. Let $K = B(0,1) \subseteq \mathbb{R}^2$ be a compact set. Consider a random variable $X \sim \mathcal{N}(0, \mathrm{Id}_2)|_K$ (i.e. given by $X = Z\mathbb{1}_{Z\in K}$ for $Z \sim \mathcal{N}(0, \mathrm{Id}_2)$) and set $Y = \|X\|^2$. Notice that $\pi$ defined this way is compactly supported.

Clearly $G$ is finite (thus compact) and it can be seen as its ortogonal representation:

$$\rho_G = \left\{ \begin{pmatrix} 1 & 0 \\ 0 & 1 \end{pmatrix}, \begin{pmatrix} 0 & -1 \\ 1 & 0 \end{pmatrix}, \begin{pmatrix} -1 & 0 \\ 0 & -1 \end{pmatrix}, \begin{pmatrix} 0 & 1 \\ -1 & 0 \end{pmatrix} \right\} \subseteq O(2) \ , \ \hat{\rho}_G = \{\mathrm{Id}_1\} \subseteq O(1) \text{ (trivial repr.)}$$

By the definition of our r.v.s, it is clear that:

- $X \overset{(d)}{=} \rho_g X \ \ \forall g \in G$ because $X \sim N(0, \mathrm{Id}_2)$

- $\forall g \in G, \ (X,Y) \overset{(d)}{=} (\rho_g X, \hat{\rho}_g Y)$ (since $\hat{\rho}$ is the trivial representation, it is enough to notice that $\|\rho_g.X\|^2 = \|X\|^2$ for all $g \in G$.

Therefore, $\pi = \mathrm{Law}(X,Y)$ is $G$-invariant (and compactly supported). Consider a *shallow NN* given by: $\Phi_\theta^N : \mathbb{R}^2 \longrightarrow \mathbb{R}^{N\times b} \longrightarrow \mathbb{R}$ (with $b \in \mathbb{N}$ and some action $G \circlearrowright_\eta \mathbb{R}^b$) as: $\Phi_\theta^N(x) = \frac{1}{N}\sum_{i=1}^N W_i \sigma(A_i^T x + B_i), \ \ \forall x \in \mathbb{R}^d$; where $\theta_i = (W_i, A_i, B_i) \in \mathcal{Z} := \mathbb{R}^{1\times b} \times \mathbb{R}^{2\times b} \times \mathbb{R}^b \cong \mathbb{R}^D$. We let $G \circlearrowright_M \mathcal{Z}$ as described in appendix C.1:

$$M_g.\theta_i = (\hat{\rho}_g W_i \eta_g^T, \rho_g \ A_i \ \eta_g^T, \eta_g.B_i) = (W_i \eta_g^T, \rho_g \ A_i \ \eta_g^T, \eta_g.B_i)$$

We can assume, for instance, that $b = 1$ and $\eta$ is the trivial representation (so that no condition is required for $\sigma_*$ to be jointly $G$-equivariant) and recall that: $\theta_i \in \mathcal{E}^G \iff \forall g \in G, \ M_g\theta_i = \theta_i$.

However, if we assume that: $\forall g \in G, \; \rho_g A_i = A_i$, then, in particular: $\begin{pmatrix} -1 & 0 \\ 0 & -1 \end{pmatrix} \begin{pmatrix} A_i^1 \\ A_i^2 \end{pmatrix} = \begin{pmatrix} A_i^1 \\ A_i^2 \end{pmatrix}$.

This in turn implies, as $A_i^1 = -A_i^1$ and $A_i^2 = -A_i^2$ that $A_i^1 = A_i^2 = 0$. i.e. $A_i \equiv 0$. Thus, any $\theta_i = \begin{pmatrix} w_i \\ A_i \end{pmatrix} \in \mathcal{E}^G$ has $A_i = 0$. Therefore, if we choose any activation $\sigma$ (e.g the *sigmoid* activation or $\sigma = \tanh$, both $\mathcal{C}^\infty$ and bounded) and we choose $N \in \mathbb{N}^*$ and $\theta_i \in \mathcal{E}^G \; \forall \; i = 1, ..., N$; then:

$$\forall x \in \mathbb{R}^2, \; \Phi_\theta^{N, \mathcal{E}^G}(x) = \frac{1}{N} \sum_{1=1}^N W_i \sigma(0^T \cdot x + B_i) = \frac{1}{N} \sum_{i=1}^N W_i \sigma(B_i),$$

which is a constant independent of $x$. i.e. **any equivariant architecture in this context** is a constant function (whereas $Y = \|X\|^2$ is not). Notice, in particular, that any shallow model $\Phi_\nu$ with $\nu \in \mathcal{P}(\mathcal{E}^G)$ will also be a constant function. In particular, notice that we will never do 'better' than minimizing over all posible constants:

$$\inf_{\substack{\theta_i \in \mathcal{E}^G \\ i=1...N \\ N \in \mathbb{N}}} R(\Phi_\theta^{\mathcal{E}^G}) \geq \inf_{\nu \in \mathcal{P}(\mathcal{E}^G)} R(\nu) \geq \inf_{C \in \mathbb{R}} \mathbb{E}[|Y - C|^2] = \inf_{C \in \mathbb{R}} \mathbb{E}[|\|X\|^2 - C|^2].$$

The problem on the right has a known answer, which is $C^* = \mathbb{E}_\pi[\|X\|^2] > 0$. On the other hand, consider a fully conected neuronal network. By the **universal approximation theorem** (which applies for the chosen $\sigma$, as in [37, 20, 5]), as $\pi$ is **compactly supported** (in particular, $\pi_\mathcal{X}(K) = 1$); we consider the **parameters** that approximate the function $f(x) = \|x\|^2$ in $K = B(0,1)$ to precision $\varepsilon > 0$. i.e. For $\varepsilon \in (0, \sqrt{C^*})$, we know: $\Longrightarrow \exists N \in \mathbb{N}, \; \exists a_1, ..., a_N \in \mathbb{R}^2, \; \exists w_1, ..., w_N \in \mathbb{R}^1$ such that:

$$\|\Phi_\theta^N - f\|_{\infty, K} = \sup_{x \in K} |\Phi_\theta^N(x) - f(x)| < \varepsilon < \sqrt{C^*}$$

Then:

$$\mathbb{E}[|Y - \Phi_\theta^N(x)|^2] \leq \mathbb{E}[(\sup_{x \in K} |\Phi_\theta^N(x) - f(x)|)^2] < \mathbb{E}[C^*] = C^*$$

But, in particular, $\exists \nu_\theta^N \in \mathcal{P}(\mathcal{Z})$ such that:

$$\mathbb{E}[|Y - \Phi_\theta^N(x)|^2] < C^*$$

and so:

$$\inf_{\mu \in \mathcal{P}(\mathcal{Z})} R(\mu) \leq \inf_{\theta \in \mathcal{Z}^N} \mathbb{E}[|Y - \Phi_\theta^N(x)|^2] < C^* \leq \inf_{\nu \in \mathcal{P}(\mathcal{E}^G)} R(\nu)$$

In particular, we **can't expect** an optimum of the learning problem to be achieved within $\mathcal{P}(\mathcal{E}^G)$. $\quad \square$

To overcome situations as in Proposition 4, the usual setting is to assume some **universality** condition. This leads to Proposition 5, which we will now prove:

*Proof of Proposition 5.* A standard calculation from the quadratic loss case (see the proof of Corollary 1 for details) yields that, for any $\mu \in \mathcal{P}(\mathcal{Z})$:

$$R(\mu) = \mathbb{E}_\pi[\|Y - \Phi_\mu(X)\|_\mathcal{Y}^2] = R_* + \mathbb{E}_\pi[\|f^*(X) - \Phi_\mu(X)\|_\mathcal{Y}^2]$$

where $f^*(x) := \mathbb{E}_\pi[Y|X = x]$ and $R_*$ is the *Bayes risk* of the problem. From Proposition 10, we know that $f^* \in L_G^2(\mathcal{X}, \mathcal{Y}; \pi_\mathcal{X})$, and so by universality of $\mathcal{F}_{\sigma_*}(\mathcal{P}(\mathcal{E}^G))$ onto that space (as well as that of $\mathcal{F}_{\sigma_*}(\mathcal{P}(\mathcal{Z}))$), we conclude directly that: $\inf_{\nu \in \mathcal{P}(\mathcal{E}^G)} R(\nu) = R_* = \inf_{\mu \in \mathcal{P}(\mathcal{Z})} R(\mu)$. $\quad \square$

**Remark.** *Works such as [50, 60, 76, 77] precisely provide conditions under which **universality** of $\mathcal{F}_{\sigma_*}(\mathcal{P}(\mathcal{E}^G))$ can be guaranteed (modulo some adaptations from their setting to ours).*

*Particularly, our single-hidden-layer NNs, with the width $N \to \infty$, correspond to what is referred to as 'networks of tensor order 1' in the literature. As noted in [50], such kind of equivariant NNs are unable to achieve universality for certain types of group actions (see Theorem 2 from [50]). Despite this, 'first order universality' has already been established for some of the most important examples of equivariant architectures, such as Deep Sets ([76, 77]) and CNNs ([76]).*

*Adapting our setting, in order to eventually allow for arbitrary order tensors in the MF formulation, is part of the future challenges to make our work more broadly applicable.*

### E.3 Proofs of results in Section 3.3

Having laid out all the different relevant elements for our work, we can now procede with the proofs of some of our main results.

#### E.3.1 Proof of Theorem 3, Corollary 3 and Theorem 4

We start by proving Theorem 3 on the general case.

*Proof of Theorem 3.* We know that a family $(\mu_t)_{t \geq 0} \subseteq \mathcal{P}_2(\mathcal{Z})$ satisfies **WGF**$(F)$ in the weak sense if $\forall \varphi \in C_c^\infty(\mathcal{Z} \times (0, T))$:

$$\int_0^T \int_{\mathcal{Z}} (\partial_t \varphi(z, t) - \langle \varsigma(t) D_\mu F(\mu_t, z), \nabla_z \varphi(z, t) \rangle) \, d\mu_t(z) \, dt = 0$$

Now, profiting from the **uniqueness** of the solutions of this equation, it will be enough to show that, given a solution $(\mu_t)_{t \geq 0} \subseteq \mathcal{P}_2(\mathcal{Z})$ of **WGF**$(F)$, then $(\mu_t^G)_{t \geq 0} \subseteq \mathcal{P}_2^G(\mathcal{Z})$ **is also a solution**. Indeed, consider, for $g \in G$, $\tilde{\mu}_t = M_g \# \mu_t$, and notice that for $\varphi \in C_c^\infty(\mathcal{Z} \times (0, T))$:

$$\int_0^T \int_{\mathcal{Z}} (\partial_t \varphi(z, t) - \langle \varsigma(t) D_\mu F(\tilde{\mu}_t, z), \nabla_z \varphi(z, t) \rangle) d\tilde{\mu}_t(z) \, dt$$

$$= \int_0^T \int_{\mathcal{Z}} (\partial_t \varphi(M_g.z, t) - \langle \varsigma(t) D_\mu F(M_g \# \mu_t, M_g.z), \nabla_z \varphi(M_g.z, t) \rangle) \, d\mu_t(z) \, dt =: \star$$

Now, we can define $\varphi^g \in C_c^\infty(\mathcal{Z} \times (0, T))$ given by $\forall (z, t) \in \mathcal{Z} \times (0, T)$ $\varphi^g(z, t) = \varphi(M_g.z, t)$, which satisfies:

$$\partial_t \varphi^g(z, t) = \partial_t \varphi(M_g.z, t) \text{ and } \nabla_z \varphi^g(z, t) = M_g^T \nabla_z \varphi(M_g.z, t)$$

So that, by also using proposition 13 and the orthogonality of the group action, we get:

$$\star = \int_0^T \int_{\mathcal{Z}} (\partial_t \varphi^g(z, t) - \langle M_g.\varsigma(t) D_\mu F(\mu_t, z), M_g \nabla_z \varphi^g(z, t) \rangle) \, d\mu_t(z) \, dt$$

$$= \int_0^T \int_{\mathcal{Z}} (\partial_t \varphi^g(z, t) - \langle \varsigma(t) D_\mu F(\mu_t, z), \nabla_z \varphi^g(z, t) \rangle) \, d\mu_t(z) \, dt = 0$$

Where the last equality comes from the fact that $(\mu_t)_{t \geq 0}$ is a solution to the **WGF**.

In particular, as we also have that $\tilde{\mu}_0 = M_g \# \mu_0 = \mu_0$ (because $\mu_0 \in \mathcal{P}^G(\mathcal{Z})$), by uniqueness we can conclude that this means that $\forall g \in G, \forall t \in (0, T)$ $\lambda$-a.e., $\mu_t = M_g \# \mu_t$.

This may seem *weaker* that what we want to prove. Nevertheless, as our group is compact and has a unique normalized Haar measure, we can proceed as follows: let $f : [0, T] \times \mathcal{Z} \to \mathbb{R}_+$ be any positive and measurable function. Given $g \in G$, take $\Omega_g \subseteq [0, T]$ a full measure set where it holds that $\mu_t = M_g \# \mu_t$. In particular, $f_t = f(t, \cdot) : \mathcal{Z} \to \mathbb{R}$ is positive and measurable, so that:

$$\forall t \in \Omega_g, \ \langle f_t, \mu_t \rangle = \langle f_t, M_g \# \mu_t \rangle = \langle f_t \circ M_g, \mu_t \rangle$$

and we can integrate this equality to get: $\int_0^T \langle f_t, \mu_t \rangle dt = \int_0^T \langle f_t \circ M_g, \mu_t \rangle dt$ Now, by integrating both sides with respect to the Haar measure, and applying Fubini's theorem (because everything is positive) we get:

$$\int_0^T \langle f_t, \mu_t \rangle dt = \int_G \int_0^T \langle f_t, \mu_t \rangle dt d\lambda_G(g) = \int_G \int_0^T \langle f_t \circ M_g, \mu_t \rangle dt d\lambda_G(g) = \int_0^T \langle f_t, \mu_t^G \rangle dt$$

Implying (by a standard argument) that $\forall t \in [0, T]$ a.e. $\mu_t = \mu_t^G$, and therefore: $\forall t \in [0, T]$ a.e. $\mu_t \in \mathcal{P}^G(\mathcal{Z})$.

$\square$

*Proof of Corollary 3.* From Corollary 2, Corollary 6 and Corollary 8, we know that $R^{\tau, \beta}$ is invariant.

On the other hand, from [38] (or [70]) we know that, under our assumptions, a **unique** weak solution to the Fokker-Planck equation exists. Furthermore, this solution is known to be strong if $\beta > 0$.

In particular, theorem 3 applies and allows us to conclude that if $\mu_0 \in \mathcal{P}_2^G(\mathcal{Z})$, then $\forall t \geq 0$ (a.e.) $\mu_t \in \mathcal{P}_2^G(\mathcal{Z})$.

When $\beta > 0$, since solutions are **strong**, we conclude that the densities $(u_t)_{t \geq 0}$ are all $G$-invariant functions ($\lambda$-a.e.). This follows from the remark about *densities of invariant measures* provided in SuppMat-B.2. $\qquad\square$

**Remark.** *When $\beta > 0$, we have a unique weakly-invariant minimizer (from proposition 3 and/or corollary 8); and also, under mild assumptions, a **global convergence** result. That is, independently of the network's initialization, we will converge to the G-invariant solution. An interesting question in this setting is then: At which point does the **WGF** enter the space $\mathcal{P}_2^G(\mathcal{Z})$?*

We can also prove Theorem 4:

*Proof of Theorem 4.* This proof follows from the fact that $\forall z \in \mathcal{Z}, \forall \mu \in \mathcal{P}^G(\mathcal{Z})$:

$$D_\mu R^{DA}(\mu, z) = D_\mu R^{FA}(\mu, z).$$

Indeed, notice that, from proposition 18:

$$D_\mu R^{FA}(\mu, z) = \int_G M_g^T . D_\mu R(\mu^G, M_g.z) d\lambda_G(g) = \int_G M_g^T . D_\mu R(\mu, M_g.z) d\lambda_G(g),$$

while also:

$$D_\mu R^{DA}(\mu, z) = \int_G M_g^T . D_\mu R(M_g \# \mu, M_g.z) d\lambda_G(g) = \int_G M_g^T . D_\mu R(\mu, M_g.z) d\lambda_G(g)$$

Now, let $(\mu_t^{FA})_{t \geq 0}$ and $(\mu_t^{DA})_{t \geq 0}$ be the WGF solutions starting from $\mu_0$ for $R^{FA}$ and $R^{DA}$ respectively. As $R^{FA}$ is $G$-invariant, by corollary 3, $(a.e.)\forall t \geq 0$, $\mu_t^{FA} \in \mathcal{P}^G(\mathcal{Z})$. Now, let's see that this process actually **also satisfies WGF**($R^{DA}$), forcing both processes to coincide by uniqueness.

Indeed, we know that $(\mu_t^{FA})_{t \geq 0}$ satisfies: $\forall \varphi \in C_c^\infty(\mathcal{Z} \times (0, T))$:

$$\int_0^T \int_{\mathcal{Z}} \left( \partial_t \varphi(z, t) - \langle \varsigma(t) D_\mu R^{FA}(\mu_t^{FA}, z), \nabla_z \varphi(z, t) \rangle \right) d\mu_t^{FA}(z) \, dt = 0$$

Now, as $(a.e.)\forall t \geq 0$, $\mu_t^{FA} \in \mathcal{P}^G(\mathcal{Z})$, we have $\forall z \in \mathcal{Z}$: $D_\mu R^{FA}(\mu_t^{FA}, z) = D_\mu R^{DA}(\mu_t^{FA}, z)$. In particular, $(\mu_t^{FA})_{t \geq 0}$ satisfies $\forall \varphi \in C_c^\infty(\mathcal{Z} \times (0, T))$:

$$\int_0^T \int_{\mathcal{Z}} \left( \partial_t \varphi(z, t) - \langle \varsigma(t) D_\mu R^{DA}(\mu_t^{FA}, z), \nabla_z \varphi(z, t) \rangle \right) d\mu_t^{FA}(z) \, dt = 0$$

Implying that $(\mu_t^{FA})_{t \geq 0}$ solves **WGF**($R^{DA}$) starting from $\mu_0$; thus by uniqueness: $(\mu_t^{FA})_{t \geq 0} = (\mu_t^{DA})_{t \geq 0}$.

The last part of the theorem comes from Proposition 2, since if $R$ is invariant, its WGF will exactly coincide with that of $R^{DA}$ (they are the same functional).

$\qquad\square$

**Remark.** *This results tells us the ultimate bottom line: at the MF level, training with **DA** or **FA** results in the exact same dynamic. Furthermore, whenever data is equivariant, they are both essentially equivalent to **applying no technique whatsoever**. Despite this result concerning infinitely wide NNs, it provides meaningful practical insights (as shown Appendix F) for large enough NNs, that could be used in applications.*

**Remark.** *Since $R^{DA}$, $R^{FA}$ and $R$ all coincide on $\mathcal{P}^G(\mathcal{Z})$, one could expect Theorem 4 to hold for $R$ even without assuming $\pi$ to be equivariant. However, the invariance of $R$ is **crucial** for such a result: if $R$ isn't invariant, nothing guarantees that its WGF process will stay within $\mathcal{P}^G(\mathcal{Z})$, whereas **WGF**($R^{DA}$) and **WGF**($R^{FA}$) always do so.*

### E.3.2 Proof of Theorem 5 and Theorem 6

We can now provide the proof for the *stronger result*, Theorem 5, stating that the WGF of an invariant functional will *respect* $\mathcal{E}^G$ all along training. This proof uses the **McKean-Vlasov non-linear SDE** (Equation (11)) presented in Appendix D.2 (see [22] for a reference). Namely, we will consider the following **projected McKean-Vlasov SDE** (with $(B_t)_{t \geq 0}$ a BM on $\mathcal{Z}$), given by:

$$dZ_t = \varsigma(t)[-(D_\mu R(\mu_t, \cdot) + \tau \nabla_\theta r(Z_t))\, dt + \sqrt{2\beta} P_{\mathcal{E}^G} dB_t]\ \ with\ \ \mathrm{Law}(Z_t) = \mu_t, \qquad (12)$$

which corresponds to the MF limit dynamics arising from performing the **projected noisy SGD** scheme from Equation (5). This can be shown by adapting relatively standard arguments from the **MF** literature on NN, see e.g. [22, 23, 53].

*Proof of Theorem 5.* The proof has two steps. The first will consist in showing that the process (12) satisfies $\forall t \geq 0, \ \mu_t \in \mathcal{P}(\mathcal{E}^G)$. In the second step we will check that $(\mu_t)_{t \geq 0} = (\mathrm{Law}(Z_t))_{t \geq 0}$ is a solution to the **WGF** $(R_{\mathcal{E}^G}^{\tau, \beta})$ presented in Section 3.3.

**Step 1:** The (**pathwise unique**) solution of the **projected McKean-Vlasov SDE** (12), $Z = (Z_t)_{t \geq 0}$ satisfies a.s. for all $t \geq 0$:

$$Z_t = Z_0 - \int_0^t \varsigma(s) D_\mu R^\tau(\mu_s, Z_s) ds + \sqrt{2\beta} \int_0^t \varsigma(s) P_{\mathcal{E}^G} dB_s, \ \ and \ \ Z_0 = \xi_0 \ (\text{initial condition})$$
$$(13)$$

Here, $\xi_0$ is such that $\mathrm{Law}(\xi_0) = \mu^0$, and $R^\tau := R + \langle r, \cdot \rangle$ is being used as shorthand notation. We first let $g \in G$ be an arbitrary group element, and we study how the process $\tilde{Z} = (\tilde{Z}_t)_{t \geq 0} := (M_g Z_t)_{t \geq 0}$ satisfies this same equation (13).

Denote $\nu_s := M_g \# \mu_s$ as the law of $\tilde{Z}_s$, we want to show that for all $t \geq 0$:

$$\tilde{Z}_t \stackrel{a.s.}{=} \tilde{Z}_0 - \int_0^t \varsigma(s) D_\mu R^\tau(\nu_s, \tilde{Z}_s) ds + \sqrt{2\beta} \int_0^t \varsigma(s) P_{\mathcal{E}^G} dB_s \qquad (14)$$

Indeed, first notice that:

1. Let $\Omega$ be the full measure set where $\xi_0 \in \mathcal{E}^G$ (which we can do since $\mu_0 \in \mathcal{P}(\mathcal{E}^G)$, or, equivalently: $\mathbb{P}(\xi_0 \in \mathcal{E}^G) = 1$). Then, $\forall \omega \in \Omega, \ Z_0(\omega) = \xi_0(\omega) \in \mathcal{E}^G$. In particular, $\forall \omega \in \Omega, \ \forall g \in G, \tilde{Z}_0(\omega) = M_g Z_0(\omega) = M_g \xi_0(\omega) = \xi_0(\omega) = Z_0(\omega)$. That is, $\tilde{Z}_0 \stackrel{a.s.}{=} Z_0$.

2. Now, the equation is satisfied by $(Z_t)_{t \geq 0}$ and therefore, for $t \geq 0$, we have:

$$\tilde{Z}_t = M_g Z_t = M_g Z_0 - M_g \left( \int_0^t \varsigma(s) D_\mu R^\tau(\mu_s, Z_s) ds \right) + \sqrt{2\beta} M_g \int_0^t \varsigma(s) P_{\mathcal{E}^G} dB_s$$

$$= \tilde{Z}_0 - \int_0^t \varsigma(s) M_g D_\mu R^\tau(\mu_s, Z_s)) ds + \sqrt{2\beta} \int_0^t \varsigma(s) M_g . P_{\mathcal{E}^G} dB_s$$

$$= \tilde{Z}_0 - \int_0^t \varsigma(s) D_\mu R^\tau(M_g \# \mu_s, M_g . Z_s) ds + \sqrt{2\beta} \int_0^t \varsigma(s) P_{\mathcal{E}^G} dB_s$$

$$= \tilde{Z}_0 - \int_0^t \varsigma(s) D_\mu R^\tau(\nu_s, \tilde{Z}_s) ds + \sqrt{2\beta} \int_0^t \varsigma(s) P_{\mathcal{E}^G} dB_s$$

Here, we used the linearity of the integral (and the stochastic integral), the fact that $\forall g \in G, \ M_g P_{\mathcal{E}^G} = P_{\mathcal{E}^G}$, and Proposition 13, which holds for $\forall \theta \in \mathcal{Z}, \forall \mu \in \mathcal{P}(\mathcal{Z})$ (in particular for $\theta = Z_s(\omega), \ \forall \omega \in \Omega$ and $\mu_s = \mathrm{Law}(Z_s)$). Thus, $\forall g \in G$, (14) holds.

By the **pathwise uniqueness** of the solution $(Z_t)_{t \geq 0}$, we have (following, for instance, [28]):

$$\mathbb{P}\left( \sup_{t \geq 0} \| Z_t - \tilde{Z}_t \| = 0 \right) = 1$$

In particular, as $g \in G$ was arbitrary, we have that:

$$\forall g \in G, \ \sup_{t \geq 0} \| Z_t - M_g Z_t \| \stackrel{a.s.}{=} 0 \qquad (15)$$

We now want to be able to **interchange** the $\forall g \in G$ with the probability measure. Fortunately, we are dealing with a compact group with a normalized Haar measure $\lambda_G$. Indeed, from equation (15) we deduce that $\forall g \in G, \ \forall t \geq 0, \ \mathbb{P}(\|Z_t - M_g Z_t\| = 0) = 1$.

Now, notice that, for any $t \geq 0$ and $\omega \in \Omega$:

$$\|Z_t(\omega) - P_{\mathcal{E}^G} Z_t(\omega)\| = \left\| Z_t(\omega) - \int_G M_g.Z_t(\omega) d\lambda_G(g) \right\| \leq \int_G \|Z_t(\omega) - M_g.Z_t(\omega)\| \, d\lambda_G(g)$$

We can integrate both sides by $\mathbb{P}$ to get (using Fubini as functions are positive and measurable):

$$0 \leq \int_\Omega \|Z_t(\omega) - P_{\mathcal{E}^G} Z_t(\omega)\| d\mathbb{P}(\omega) \leq \int_\Omega \int_G \|Z_t(\omega) - M_g.Z_t(\omega)\| \, d\lambda_G(g) d\mathbb{P}(\omega)$$

$$\leq \int_G \int_\Omega \|Z_t(\omega) - M_g.Z_t(\omega)\| \, d\mathbb{P}(\omega) d\lambda_G(g) = 0$$

where in the last step we have used the fact that $\forall g \in G, \ \forall t \geq 0, \ \mathbb{P}(\|Z_t - M_g Z_t\| = 0) = 1$, so that $\forall t \geq 0, \ \forall g \in G, \ \int_\Omega \|Z_t(\omega) - M_g.Z_t(\omega)\| \, d\mathbb{P}(\omega) = 0$.

This implies that $\forall t \geq 0$ $\mathbb{P}$-a.s. $Z_t = P_{\mathcal{E}^G} Z_t$, i.e. $\mathbb{P}(Z_t \in \mathcal{E}^G) = \mu_t(\mathcal{E}^G) = 1$, or, in other words, $\forall t \geq 0, \ \mu_t \in \mathcal{P}(\mathcal{E}^G)$ as required. Note that all arguments work as well in the case that $\beta = 0$

**Step 2:** We now prove that $(\mu_t)_{t \geq 0}$, studied in the previous step, is a weak solution to equation (3); that is, to the **WGF**$(F)$, with (using the previously introduced notation $R^\tau$):

$$F(\mu) = R^{\tau,\beta}_{\mathcal{E}^G}(\mu) = R^\tau(\mu) + \beta H_{\lambda_{\mathcal{E}^G}}(\mu^{\mathcal{E}^G}).$$

Also recall the notation $H^{\mathcal{E}^G}(\mu) := H_{\lambda_{\mathcal{E}^G}}(\mu^{\mathcal{E}^G}) = H_{\lambda_{\mathcal{E}^G}} \circ P_{\mathcal{E}^G} \#(\mu)$ presented in the end of Appendix D.1, as well as the calculations for its intrinsic derivative.

It is standard to check, applying Itô's formula and taking expectation, that the family $(\mu_t)_{t \geq 0} = (\text{Law}(Z_t))_{t \geq 0}$ satisfies, $\forall \varphi \in C_c^\infty(\mathcal{Z} \times (0,T))$:

$$\int_0^T \int_{\mathcal{Z}} \left( \partial_t \varphi(z,t) - \langle \varsigma(t) D_\mu R^\tau(\mu_t, z), \nabla_z \varphi(z,t) \rangle + \beta \, tr[P_{\mathcal{E}^G} D_z^2 \varphi(z,t)] \right) d\mu_t(z) \, dt = 0$$

with $tr$ denoting the trace of a square matrix and $D_z^2$ the Hessian matrix acting on the $z$ variable. Notice that the process $P_{\mathcal{E}^G} B$ is classically a Brownian motion in $\mathcal{E}^G$. Together with the fact that $\mu_t$ is supported on $\mathcal{E}^G$, this implies that, for $\beta > 0$, $\mu_t$ has a density w.r.t. $\lambda_{\mathcal{E}^G}$.

It is clear from the case $\beta = 0$ that the terms involving first order spatial derivatives of $\varphi$ exactly give rise to the terms associated with functional $R^\tau$ in the definition (3) of the **WGF**$(R^{\tau,\beta}_{\mathcal{E}^G})$. Therefore, we just need to check that for all $t \geq 0$, the distribution defined for every $\phi \in C_c^\infty(\mathcal{Z})$ by $\phi \mapsto \int_{\mathcal{Z}} tr[P_{\mathcal{E}^G} D_z^2 \phi(z)] d\mu_t(z)$ and the distribution $\text{div}\left( D_\mu H^{\mathcal{E}^G}(\mu_t, \cdot) \mu_t \right)$ are equal. In fact, for all such $\phi$ we have:

$$-\int_{\mathcal{Z}} \langle \nabla_z \phi(z), D_\mu H^{\mathcal{E}^G}(\mu_t, z) \rangle \, d\mu_t(z)$$

$$= -\int_{\mathcal{Z}} \langle \nabla_z \phi(z), P_{\mathcal{E}^G}^T \nabla_z \left[ \frac{d\mu_t^{\mathcal{E}^G}}{d\lambda_{\mathcal{E}^G}} \right](P_{\mathcal{E}^G}.z) \rangle \left( \frac{d\mu_t^{\mathcal{E}^G}}{d\lambda_{\mathcal{E}^G}}(P_{\mathcal{E}^G}.z) \right)^{-1} d\mu_t(z)$$

Since $\mu_t$ is concentrated in $\mathcal{E}^G$, we have $P_{\mathcal{E}^G} \# \mu_t = \mu_t$. This and the equality $P_{\mathcal{E}^G}^2 = P_{\mathcal{E}^G}$ imply the previous expression is equal to:

$$-\int_{\mathcal{Z}} \langle \nabla_z \phi(P_{\mathcal{E}^G}.z), P_{\mathcal{E}^G}^T \nabla_z \left[ \frac{d\mu_t^{\mathcal{E}^G}}{d\lambda_{\mathcal{E}^G}} \right](P_{\mathcal{E}^G}.z) \rangle \left( \frac{d\mu_t^{\mathcal{E}^G}}{d\lambda_{\mathcal{E}^G}}(P_{\mathcal{E}^G}.z) \right)^{-1} d\mu_t(z)$$

$$= -\int_{\mathcal{E}^G} \langle \nabla_z \phi(x), P_{\mathcal{E}^G}^T \nabla_z \left[ \frac{d\mu_t^{\mathcal{E}^G}}{d\lambda_{\mathcal{E}^G}} \right](x) \rangle \left( \frac{d\mu_t^{\mathcal{E}^G}}{d\lambda_{\mathcal{E}^G}}(x) \right)^{-1} d\mu_t(x)$$

$$= -\int_{\mathcal{E}^G} \langle P_{\mathcal{E}^G}^T \nabla_z \phi(x), P_{\mathcal{E}^G}^T \nabla_z \left[ \frac{d\mu_t^{\mathcal{E}^G}}{d\lambda_{\mathcal{E}^G}} \right](x) \rangle d\lambda_{\mathcal{E}^G}(x)$$

Noticing that $P_{\mathcal{E}^G}^T \nabla_z$ is the gradient calculated on $\mathcal{E}^G$, integrating by parts with respect to $\lambda_{\mathcal{E}^G}$, and using again the fact that $\mu_t$ is concentrated in $\mathcal{E}^G$, a straightforward calculation yields the desired expression $\int_{\mathcal{Z}} tr[P_{\mathcal{E}^G} D_z^2 \phi(z)] d\mu_t(z)$. This concludes the proof. $\qquad\square$

**Remark.** *Notice that, by a.s. continuity of the McKean-Vlasov diffusion* (12) *and the fact that $\mathcal{E}^G$ is closed, Step 1 of the previous proof actually shows that $\mathbb{P}(Z_t \in \mathcal{E}^G, \forall t \geq 0) = 1$.*

Notice also that Theorem 5 bears some resemblance to *Corollary 1* in [30], which states that $\mathcal{E}^G$ is *stable* under the traditional gradient flow of the *augmented risk* ($[\theta \in \mathcal{Z}^N \mapsto R^{DA}(\theta) \in \mathbb{R}]$, as in Section 2.3). Our result shares a similar flavor, but for the MF dynamics of freely-trained NNs with equivariant data.

**Remark.** *Unlike with **WI** distributions, initializing a shallow NN with $\mu_0 \in \mathcal{P}(\mathcal{E}^G)$ isn't as straightforward as using a normal distribution. Effectively (and efficiently) computing the space $\mathcal{E}^G$ is actually quite challenging (as noted in [29]).*

*A natural way to ensure that $\mu_0 \in \mathcal{P}(\mathcal{E}^G)$, independently of the form of $\mathcal{E}^G$, is to initialize all parameters to be $0$. The question of whether under such initialization the parameters will eventually exit $\{0\}$ (or some larger, strict subspace $E \subsetneq \mathcal{E}^G$) and find values over the entire space $\mathcal{E}^G$ is left for future work. Some insights on this behaviour can be sought in our experimental results, see Section 4. If true, this behavior could point towards some type of underlying hypoellipticity of the McKean-Vlasov dynamics (12) (or variants) on $\mathcal{E}^G$, which would be interesting to analyze, in particular in view of potential theoretical guarantees for architecture-discovering heuristics as suggested in Section 4.2.*

Note that there is no need for seeing $\mathcal{E}^G$ as a subspace of an ambient space $\mathcal{Z}$. When training with **EA**s, we simply *force* our parameters to *live on $\mathcal{E}^G$*, since we fix the architechture beforehand. Namely, our 'whole space' is $\tilde{\mathcal{Z}} = \mathcal{E}^G$ (regarded directly as a vector space $\mathcal{E}^G \cong \mathbb{R}^{\tilde{D}}$) rather than $\mathcal{Z}$. Thus, the relevant *population risk* is the *restricted* version of the original: $\tilde{R} := R|_{\mathcal{P}(\tilde{\mathcal{Z}})} : \mathcal{P}(\tilde{\mathcal{Z}}) \to \mathbb{R}$; and we can apply the usual results from the MF Theory when the relevant hypothesis are satisfied by $\tilde{\mathcal{Z}}$ and $\tilde{R}$. Notably, we can have global convergence of $\tilde{R}^{\tau,\beta}$ to $\inf_{\mu \in \mathcal{P}(\tilde{\mathcal{Z}})} \tilde{R}^{\tau,\beta}(\mu) = \inf_{\mu \in \mathcal{P}(\mathcal{E}^G)} R_{\mathcal{E}^G}^{\tau,\beta}(\mu)$

**Remark.** *As shown in [38] (see Proposition 19 in Appendix G for the details) the regularized versions of the involved functionals (i.e. $R^{\tau,\beta}$ and $R_{\mathcal{E}^G}^{\tau,\beta}$) $\Gamma$-converge to the original $R$ as $\tau, \beta \to 0$; meaning that, for small values of the regularization parameters, we should expect the achieved optima to ressemble $\inf_{\mu \in \mathcal{P}^G(\mathcal{Z})} R(\mu)$ and $\inf_{\nu \in \mathcal{P}(\mathcal{E}^G)} R(\nu)$ respectively (or, under Proposition 5, both to $R_*$). We will also leave the exploration of how this approximation behaves as future work.*

Finally, we provide a proof for Theorem 6

*Proof of Theorem 6.* The proof structure is very similar to that of Theorem 4. Namely, it comes from noticing that for $\mu \in \mathcal{P}(\mathcal{E}^G)$ and $z \in \mathcal{E}^G$, we have:

$$D_\mu R^{DA}(\mu, z) = D_\mu R^{FA}(\mu, z) = D_\mu R^{EA}(\mu, z).$$

We already know the first equality, as seen in Theorem 4 (since $\mu \in \mathcal{P}^G(\mathcal{Z})$ from lemma 1). We only need to show the last equality. Indeed, notice that:

$$D_\mu R^{FA}(\mu, z) = \int_G M_g^T . D_\mu R(\mu, z) d\lambda_G(g) = P_{\mathcal{E}^G} . D_\mu R(\mu, z),$$

while also, from Proposition 18:

$$D_\mu R^{EA}(\mu, z) = P_{\mathcal{E}^G}^T . D_\mu R(\mu^{\mathcal{E}^G}, P_{\mathcal{E}^G} . z) = P_{\mathcal{E}^G} . D_\mu R(\mu, z).$$

Knowing this, the rest of the proof is analogous to that of Theorem 4. Let $(\mu_t^{FA})_{t \geq 0}$, $(\mu_t^{DA})_{t \geq 0}$ and $(\mu_t^{EA})_{t \geq 0}$ be the WGF solutions starting from $\mu_0$ for $R^{FA}$, $R^{DA}$ and $R^{EA}$ respectively. Since $\mu_0 \in \mathcal{P}(\mathcal{E}^G) \subseteq \mathcal{P}^G(\mathcal{Z})$, from Theorem 4 we know that $(\mu_t^{FA})_{t \geq 0}$ and $(\mu_t^{DA})_{t \geq 0}$ coincide. Let's see that, w.l.o.g., $(\mu_t^{FA})_{t \geq 0}$ coincides with $(\mu_t^{EA})_{t \geq 0}$.

As $R^{FA}$ is $G$-invariant, by theorem 5, $\forall t \geq 0$, $\mu_t^{FA} \in \mathcal{P}(\mathcal{E}^G)$. Now, let's see that this process **also satisfies WGF**($R^{EA}$), forcing both processes to coincide by uniqueness.

As before, we know that $(\mu_t^{FA})_{t\geq 0}$ satisfies: $\forall \varphi \in C_c^\infty(\mathcal{Z} \times (0,T))$:

$$\int_0^T \int_{\mathcal{Z}} \left(\partial_t \varphi(z,t) - \langle \varsigma(t) D_\mu R^{FA}(\mu_t^{FA}, z), \nabla_z \varphi(z,t)\rangle\right) d\mu_t^{FA}(z)\, dt = 0$$

Now, as $\forall t \geq 0$, $\mu_t^{FA} \in \mathcal{P}(\mathcal{E}^G)$, we can restrict our integral to $\mathcal{E}^G$. Also, we have $\forall z \in \mathcal{E}^G$: $D_\mu R^{FA}(\mu_t^{FA}, z) = D_\mu R^{EA}(\mu_t^{FA}, z)$. With these properties, $(\mu_t^{FA})_{t\geq 0}$ satisfies $\forall \varphi \in C_c^\infty(\mathcal{Z} \times (0,T))$:

$$\int_0^T \int_{\mathcal{E}^G} \left(\partial_t \varphi(z,t) - \langle \varsigma(t) D_\mu R^{EA}(\mu_t^{FA}, z), \nabla_z \varphi(z,t)\rangle\right) d\mu_t^{FA}(z)\, dt = 0$$

Making the integral over $\mathcal{Z}$ once again (we can since $\mu_t^{FA} \in \mathcal{P}(\mathcal{E}^G)$ for all $t \geq 0$), we get that $(\mu_t^{FA})_{t\geq 0}$ solves **WGF**$(R^{EA})$ starting from $\mu_0$; thus by uniqueness: $(\mu_t^{FA})_{t\geq 0} = (\mu_t^{EA})_{t\geq 0}$.

The last part of the theorem comes, once again, from Proposition 2, since if $R$ is invariant, its WGF exactly coincides with that of $R^{DA}$. $\qquad\square$

## F  Experimental setting and further experiments

All the different experiments were run on Python 3.10, on a Google Colab session consisting (by default) of 2 Intel Xeon virtual CPUs (2.20GHz) and with 13GB of RAM.

In order to obtain results that can be visualized, we consider a simple setting where $\mathcal{X} = \mathcal{Y} = \mathbb{R}^2$ and $\mathcal{Z} = \mathbb{R}^{2\times 2} \cong \mathbb{R}^4$. We let $G = C_2$ acting on $\mathcal{X}$ and $\mathcal{Y}$ via the *coordinate transposition action* (i.e. the group generated by the orthogonal matrix $\begin{pmatrix} 0 & 1 \\ 1 & 0 \end{pmatrix}$); and on $\mathcal{Z}$ via the natural intertwining action (i.e. $M_g.z = \hat{\rho}_g.z.\rho_g^T$). We also consider the jointly equivariant activation given by $\sigma_*(x,z) = \sigma(z \cdot x)\ \forall x \in \mathbb{R}^2, \forall z \in \mathbb{R}^{2\times 2}$ with $\sigma : \mathbb{R} \to \mathbb{R}$ a sigmoidal activation function (which is $\mathcal{C}^\infty$ and bounded) applied pointwise. Under this setting, $\mathcal{E}^G$ can be explicitly computed as $\mathcal{E}^G = \left\langle \begin{pmatrix} \frac{1}{\sqrt{2}} & 0 \\ 0 & \frac{1}{\sqrt{2}} \end{pmatrix}, \begin{pmatrix} 0 & \frac{1}{\sqrt{2}} \\ \frac{1}{\sqrt{2}} & 0 \end{pmatrix} \right\rangle$, which is a 2-dimensional subspace of the ambient 4-dimensional space. It's projection operator $P_{\mathcal{E}^G}$ is also explicitly known.

We consider a **teacher** model $f_* = \Phi_{\theta^*}^{N_*}$ with $N_*$ fixed particles, such that $\nu_{\theta^*}^{N_*}$ is either **arbitrary**, **WI** or **SI**. Let $\vartheta = 0.5$ be a scale parameter. The **arbitrary** particles were chosen to be, for $N_* = 5$:

$$\theta_1^* = \vartheta.(-1, 0, 0, 0.5)^T$$
$$\theta_2^* = \vartheta.(0.5, 1, 0, 1)^T$$
$$\theta_3^* = \vartheta.(-0.5, 0.3, 1, 0)^T$$
$$\theta_4^* = \vartheta.(0, -1, -0.5, 1)^T$$
$$\theta_5^* = \vartheta.(0.7, -0.7, 0.5, 0.7)^T.$$

This was fixed in order to make the task non-trivial and *interesting*. The **WI** teacher distribution was simply chosen to be $(\nu_{\theta^*}^{N_*})^G$, with $\theta^*$ as just described, so that the corresponding teacher function resulted to be $f_* = \mathcal{Q}_G.\Phi_{\theta^*}^{N_*}$. In other words, the **WI** distribution has 10 particles, corresponding to each of those of $\theta^*$, together with their image under the $G$-action. The **SI** particles were also fixed, but their chosen coordinates had to be expressed in terms of the basis vector of $\mathcal{E}^G$ (i.e. only providing 2 parameters). Particularly, they were fixed to be $N_* = 5$ and, denoting them by $a^* = (a_i^*)_{i=1}^{N_*}$ to avoid confusion, explicitly described as:

$$a_1^* = \vartheta.(1, 0)^T,\ a_2^* = \vartheta.(0.5, 1)^T,\ a_3^* = \vartheta.(-0.5, 0.3),\ a_4^* = \vartheta.(0, -1),\ a_5^* = \vartheta.(0.7, 0.7).$$

As seen on Section 3.1, the teacher $f_* : \mathcal{X} \to \mathcal{Y}$ will be an equivariant function as soon as its parameter distribution is chosen either **WI** or **SI**. Our data distribution $\pi$ will be such that $(X, Y) \sim \pi$ will satisfy $X \sim \mathcal{N}(0, \sigma_\pi^2.\mathrm{Id}_2)$ (with $\sigma_\pi = 4$), and $Y = f_*(X)$. Namely, $\pi_\mathcal{X}$ will always be $G$-invariant, whereas $\pi$ will only be $G$-invariant if $f_*$ is. This setting allows for testing the different results provided, without losing the properties of $\mathcal{Q}_G$ as a projection (which require $\pi_\mathcal{X}$ to be $G$-invariant, as shown in [27]).

We will try to **mimic** the teacher network by using **student networks**, which will be given by $\Phi_\theta^N$; namely, with the same $\sigma_*$, but varying values of $N$ and $\theta \in \mathcal{Z}^N$. We will train them to minimize the **regularized population risk** $R^{\tau,\beta}$ given by a **quadratic loss**, $\ell(y, \hat{y}) = \|y - \hat{y}\|_{\mathcal{Y}}^2$, and a **quadratic penalization**, $r(z) = \|z\|_{\mathcal{Z}}^2$. For this purpose, we employ a *minibatch* variant of the **SGD training scheme** provided in Equation (1) (possibly **projected**, as in Equation (5), when required). We will also employ the different symmetry-leveraging techniques presented in Section 2.3, such as **DA**, **FA** and **EA**. We refer to the *free training* with no SL-techniques whatsoever as the **vanilla** training.

The training parameters were fixed to be (unless explicitly stated otherwise):

- **Step Size:** $\varsigma \equiv \alpha > 0$ (with $\alpha = 50$ in most experiments), $\varepsilon_N = \frac{1}{N}$, so that $s_k^N = \frac{\alpha}{N}$. This was convenient, since it corresponds to the usual implementation of SGD on most common NN frameworks in Python (namely, **pytorch** and **jax**).

- **Regularization parameters:** $\tau = 10^{-4}$ and $\beta = 10^{-6}$.

- **Batch Size:** It was chosen to be $B = 20$.

- **Number of Training Epochs:** In line with the statement of Theorem 1, to observe phenomena at a MF scale, we need an amount of iterations (commonly known as *epochs* in the ML literature) that is proportional to the number of particles. For this purpose, we fix an *observation time horizon* of $T = 20$. All training schemes were performed for a total of $N_e = N \cdot T$ epochs (iterations). An additional 'granularity' parameter (usually set to be $gr = 5$) is introduced to determine *how often in the dynamic we will observe and save the training losses and particle positions*: we do so every $\lfloor \frac{N_e}{gr} \rfloor$ steps. Notice that $N_e$ depends on $N$, and so models with different values of $N$ were trained for a different amount of epochs.

- **Student Initialization:** The student's particles, $\theta \in \mathcal{Z}^N$, are initialized i.i.d. from some $\mu_0$ that is chosen to be either **WI** or **SI**. When **WI**-initialized, they are sampled from a random gaussian $Z \sim \mathcal{N}(0, \frac{1}{16})$. When **SI**-initialized, particles are taken to be $P_{\mathcal{E}^G}.Z$ with $Z$ as before.

- **Number of Repetitions:** Each experiment was repeated a total of $N_r = 10$ times to ensure consistency. Each repetition, a different random *seed* was employed to: initialize the student's particles, generate the training data, and generate the noise for the SGD iteration. In particular, on a fixed repetition, all models were trained with the **same data** and the **same noise** being applied on SGD updates.

**Remark.** *As $N_e$ is chosen to be proportional to the number of particles, $N$, computational burden and memory requirements quickly became heavy for the simple machines we employed (which didn't even have a dedicated GPU). This is the reason why we don't scale our experiments beyond the $N = 5000$ case. As reference, for $N = 5000$ a single training (with the above hyperparameters) of a single model (either of **vanilla**, **DA**, **FA** or **EA**) took $\approx 15$ minutes (which quickly amounts to large amounts of running time for the $N_r = 10$ repetitions, the $4$ different training schemes and the $6$ possible settings with **WI** or **SI** initialization and **arbitrary**, **WI** or **SI** teacher). This is a clear point of improvement and shall be tackled in future work.*

**Remark.** *As here mentioned, on every fixed 'repetition' of the experiments, the same noise was used during the SGD training iterations for the **vanilla**, **DA** and **FA** schemes. The **EA** scheme, despite using the same seeds for the data and student initializations, didn't have the same noise applied during SGD. This was because, despite using the same seed for the noise generator, noise for our EAs was only $2$-dimensional (since **EAs** are parametrized by $\mathcal{E}^G$), while it was $4$-dimensional for the other schemes. This made the resulting training schemes have an additional layer of noise separating them; and so solving this issue, in order to properly visualize Theorem 6, becomes fundamental.*

**Remark.** *Notice that the $N_r = 10$ performed repetitions were largely enough to allow for plots with error bars (actually, we do boxplots which encode the variability of the different quantities more precisely) that allow for significant analysis of the observed phenomena. We do not go beyond $N_r = 10$ due to the low computational capabilities of our machines, and the already high computational cost of running the experiments (for thousands of hidden units and epochs, in many different settings, and involving the calculation of Wasserstein Distances, as we'll comment below).*

To facilitate the implementation of the ideas behind our **EA** models, we use the *group* and *representations* tools from the **emlp** repository provided as part of [29]. This code is openly available and

has an MIT License for unrestricted access. We employ it to numerically (and efficiently) determine the space $\mathcal{E}^G$ (namely, its basis), as well as $P_{\mathcal{E}^G}$. We do remark that these calculations were correct only up to a precision of $10^{-8}$, which results in a slight burden for our empirical results. On the other hand, regarding the implementation of **EMLP**s provided in the package (**EA**s in our setting), some slight modifications to the source code had to be performed in order to correctly represent our setting.

On a similar note, we can numerically compute the squared Wasserstein-2 distance between two empirical distributions of particles by employing the **pyot** library. This allows us to evaluate *to what extent* our resulting models are close to each other in terms of their particle distribution $\nu_\theta^N$ (which is what the **MF** approach suggests). In order to fix a common scale in which the experiments can be compared, for different values of $N$, and mitigate the effects of fluctuating empirical estimates (mainly for small values of $N$, and in the low dimensions considered), we consider a natural normalization of the Wasserstein-2 distance, which we refer to as the **RMD** (Relative-Measure-Distance). This is defined as: $\mathbf{RMD}^2(\mu, \nu) = \frac{W_2^2(\mu,\nu)}{M_\mu^2 + M_\nu^2}$ where $M_\mu^2 = 2\mathbb{E}[\|Z\|^2]$ for $Z \sim \mu$ (so that $0 \leq \mathbf{RMD} \leq 1$). The **RMD** provided a good metric for the experiments here presented. Notice that, as $N \to \infty$, by the L.L.N. for empirical distribution following from the **MFL** convergence, the **RMD** is expected to stabilize at the corresponding value of the limiting distributions. Therefore, up to a multiplicative quantity approaching a (finite, non null, in our case) constant, we are observing the behavior of the Wasserstein-2 distance. A drawback from using the Wasserstein-2 metric, is that calculating them can be very expensive computationally. This is another one of the reasons why our experiments only get to $N = 5000$ particles.

## F.1   Study for varying $N$

Beyond the analyisis already provided in Section 4, we here provide some meaningful insights. We want to observe to what extent the properties proved in Section 3.3 for the **WGF** of $R^{\tau,\beta}$ can be observed in practice. For this purpose, we observe, for $N \in \{5, 10, 50, 100, 500, 1000, 5000\}$, different relevant quantities to evaluate the different combinations of *teachers* and *students*.

For this set of experiments, **WI**-initialized students were trained with the usual **SGD** scheme from Equation (1); while, **SI**-initialized students, were trained with the **projected SGD dynamics** from Equation (5). Models for the different schemes to be compared, are all initialized with the exact same (random) particles.

Figure 3 displays some comparisons between particle distributions in terms of **RMD** at the end of training, knowing that they were all initialized with the same particles drawn from $\mu_0 \in \mathcal{P}(\mathcal{E}^G)$. From this, we can visually see that, when the teacher distribution is either **WI** of **SI**, the resulting distribution from vanilla training *stays* on $\mathcal{E}^G$ (since $\mathbf{RMD}^2(\nu_{NT}^N, (\nu_{NT}^N)^{\mathcal{E}^G})$ is small) increasingly more as $N$ becomes large. This fact is absolutely remarkable, since, for a **WI** teacher there should be no reason why the **vanilla** training (that's completely *free*, in principle) *shouldn't escape* $\mathcal{E}^G$ to achieve a better approximation of $f_*$. On the other hand, in every single teacher setting, almost independently of $N$, both **DA** and **FA** consistently remain within $\mathcal{E}^G$ (as expected, even if $f_*$ isn't equivariant). For an **arbitrary teacher**, we see that the vanilla training distribution readily *leaves* $\mathcal{E}^G$ to better approximate $f_*$, which isn't a *predicted behaviour* from our theory (we have no guarantees of 'leaving $\mathcal{E}^G$ when data isn't equivariant'), but motivates the heuristic defined in Section 4.

Still from Figure 3, we can see that, as $N$ grows bigger, the end-of-training distribution of the **vanilla** scheme becomes closer and closer to that of **DA** and **FA** (from Theorem 4 we actually expect them to be equal in the limit). A similar result is obtained relating **vanilla**, **DA** and **FA** to the **EA** scheme; the values are however larger than before and *less significantly close* in general. This is possibly due to the *different noises* employed during training (as mentioned in a remark above). We do however notice that for increasing values of $N$, the **EA**, **DA**, **FA** and **vanilla** schemes (the latter only under equivariant $f_*$) tend towards coinciding, which serves to illustrate the constatations from Theorem 6. Finally, notice that the results obtained for **WI** and **SI** teachers present almost no quantitative differences whatsoever between them.

In Figure 4 we present a visualization of the final particle distribution, after an **SI**-initialized training under a **WI** teacher $f_*$, of the **vanilla**, **DA**, **FA** and **EA** schemes (on a single realization of the experiment). At least visually (and macroscopically), it seems like all these regimes followed (approximately) the same flow, as they end up with an approximately equal particle distribution. This isn't a rigorous comparison at all, and providing better quantitative comparisons between the methods

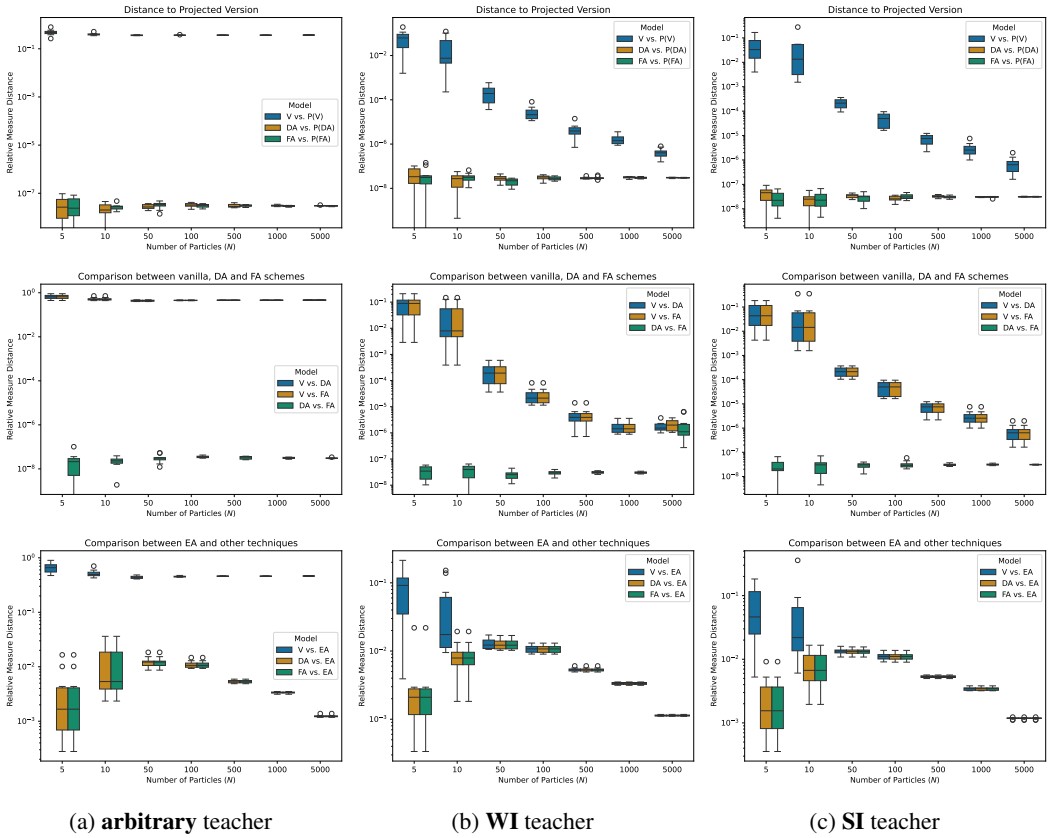

(a) **arbitrary** teacher (b) **WI** teacher (c) **SI** teacher

Figure 3: **RMD** comparisons between training regimes, for different values of $N$, at the end of an **SI**-initialized training for $N_e$ epochs. Each column corresponds to a teacher with, respectively, an **arbitrary**, **WI** and **SI** distribution. *Row 1* displays $\mathbf{RMD}^2(\nu_{N_e}^N, (\nu_{N_e}^N)^{\mathcal{E}^G})$ for the different regimes, in order to evaluate to what extent the training *remained* within $\mathcal{E}^G$. *Row 2* displays the **RMD** between **DA**, **FA** and **vanilla** training regimes; and *Row 3* does the same for each of them against **EA**.

is to be considered for future work. As a counterfactual, we provide in Figure 5 the results of an **SI**-initialized training that is performed under a non-equivariant $f_*$. We can see that the vanilla model readily *leaves* $\mathcal{E}^G$ to achieve a better approximation of $f_*$, while the **DA**, **FA** and **EA** schemes 'stay inside' (roughly coinciding between them as well).

Figure 6 also displays **RMD** comparisons between particle distributions at the end of training, but for an (identic) initialization with particles drawn from $\mu_0 \in \mathcal{P}^G(\mathcal{Z})$. Now, unlike the **SI** case, with particles sampled *i.i.d.* from a **WI** distribution, nothing ensures that the resulting $\nu_0^N$ will be **WI** as well. Namely, in this case the limit as $N \to \infty$ becomes significantly more important to visualize the theoretical results. Indeed, since $\nu_0^N$ isn't necessarily **WI**, we no longer have a guarantee that the finite-$N$ networks trained with **DA**, **FA** or **vanilla** methods will be close to each other in any sense. We do however notice on Figure 6 that, for increasing $N$, the end-of-training distributions of **DA**, **FA** and **vanilla** schemes (the latter only when $f_*$ is equivariant) become increasingly closer to their symmetrized versions (namely, $\mathbf{RMD}^2(\nu_{NT}^N, (\nu_{NT}^N)^G)$ becomes smaller, though never as small as in the **SI**-initialized experiments). Also, as guaranteed by Theorem 4, for large $N$ we see that **DA** and **FA** become indistinguishably close, no matter the teacher's properties; also, when $f_*$ is equivariant, they both 'coincide' with the **vanilla** scheme. A comparison between the **EA** scheme and the result from projecting the **FA** scheme on the last step is also presented. It is used simply to illustrate that, in principle, directly training on $\mathcal{E}^G$ isn't necessarily comparable to performing 'free-training' and *projecting* the resulting particle distribution only on the last step (even when an SL technique such as **FA** is used).

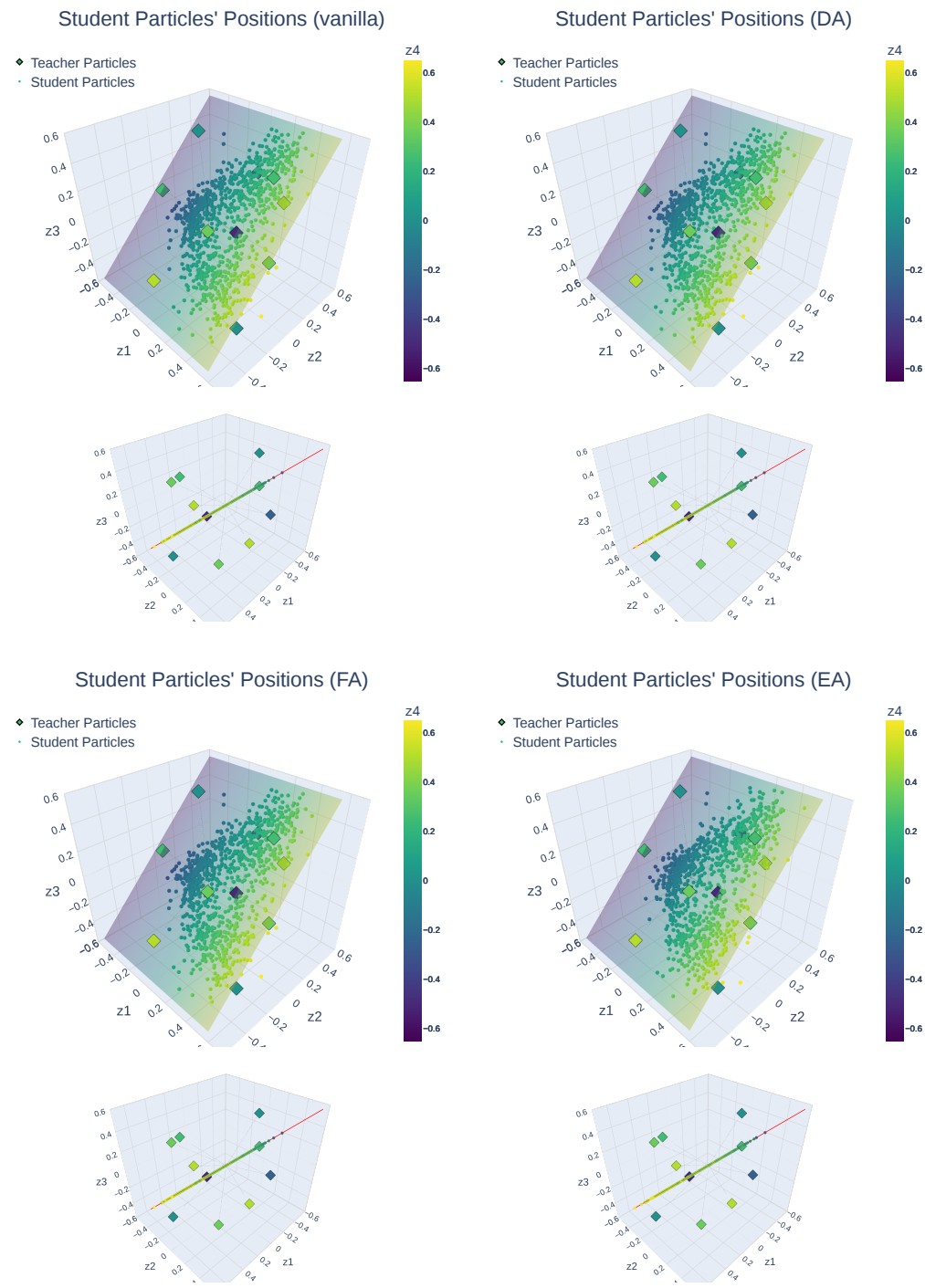

Figure 4: Visualization of the NN particles after training under the **vanilla**, **DA**, **FA** and **EA** schemes, for a single realization of the experiment. Squares represent the *teacher* particles (which are **WI**), dots represent the student particles, and the hyperplane is $\mathcal{E}^G$. The bigger plots show an aerial view of the global particle distribution after training; and the minor plots below them show a viewpoint at the level of (and parallel to) $\mathcal{E}^G$. The student particles were all initialized to be **SI** (and to coincide at initialization between the different schemes), and trained with equation (5) correspondingly applying the proper SL technique.

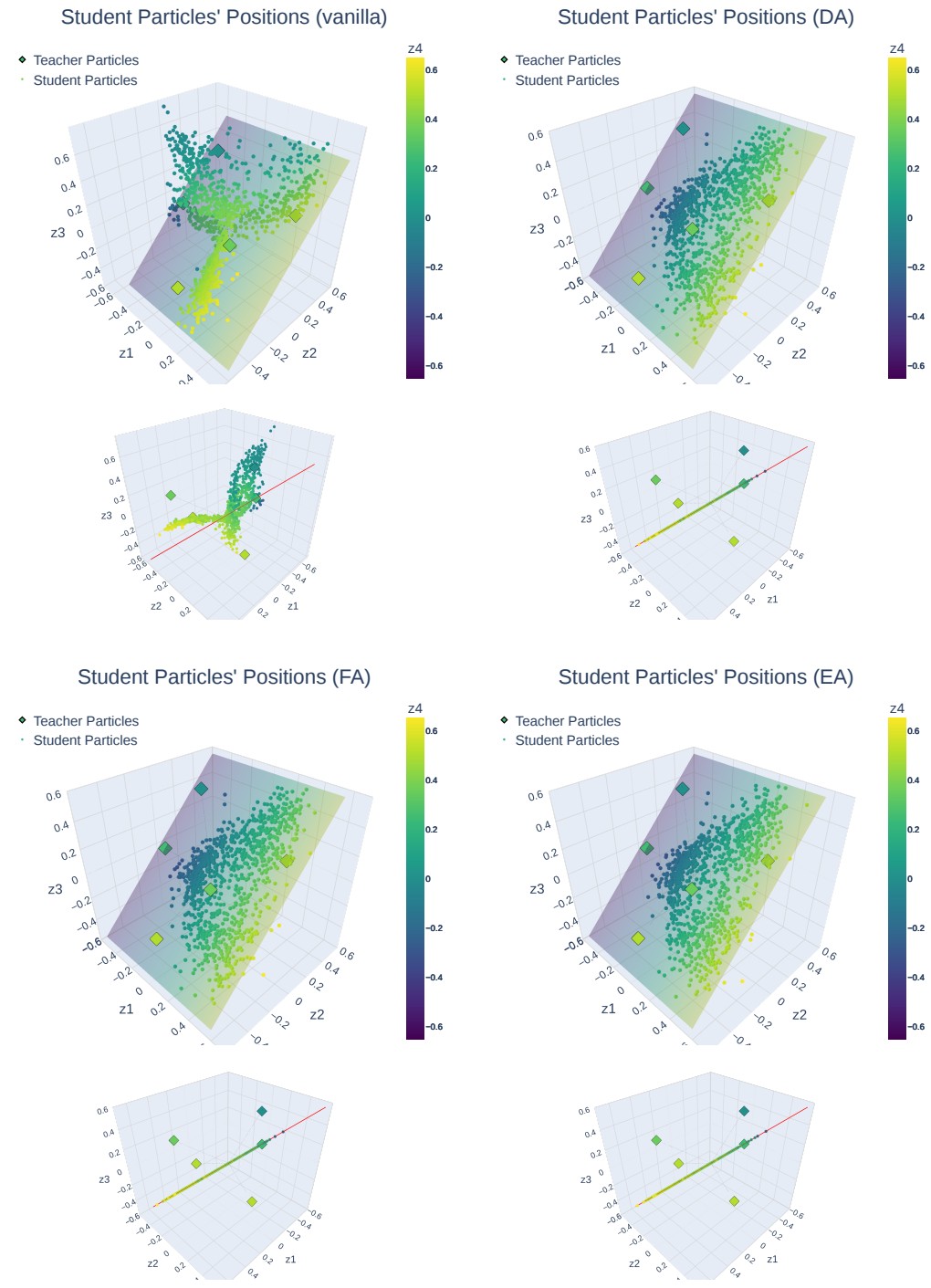

Figure 5: Visualization of the NN particles after training under the **vanilla**, **DA**, **FA** and **EA** schemes for a single realization of the experiment. Squares represent the *teacher* particles (which are **arbitrary**), dots represent the student particles, and the hyperplane is $\mathcal{E}^G$. The bigger plots show an aerial view of the global particle distribution after training; and the minor plots below them show a viewpoint at the level of (and parallel to) $\mathcal{E}^G$. The student particles were all initialized to be **SI** (and to coincide at initialization between the different schemes), and trained with equation (5) correspondingly applying the proper SL technique. Notice how the particles for the **vanilla** scheme readily **leave** $\mathcal{E}^G$ (despite the noise being projected onto it) and seem to *approach* the teacher particles.

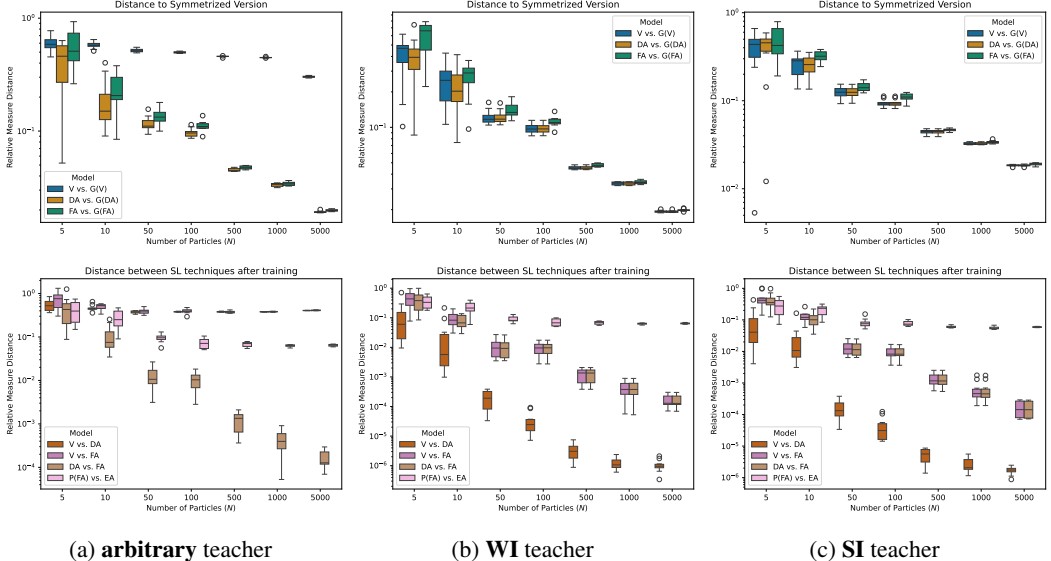

(a) **arbitrary** teacher          (b) **WI** teacher          (c) **SI** teacher

Figure 6: **RMD** comparisons between training regimes, for different values of $N$, at the end of a **WI**-initialized training for $N_e$ epochs. Each column corresponds to a teacher with, respectively, an **arbitrary**, **WI** and **SI** distribution. *Row 1* displays $\mathbf{RMD}^2(\nu_{N_e}^N, (\nu_{N_e}^N)^G)$ for the different regimes, to evaluate to what extent the training *remained* **WI**. *Row 2* displays the **RMD** between **DA**, **FA** and **vanilla** training regimes; as well as a comparison between **EA** and the projected particles of **FA**.

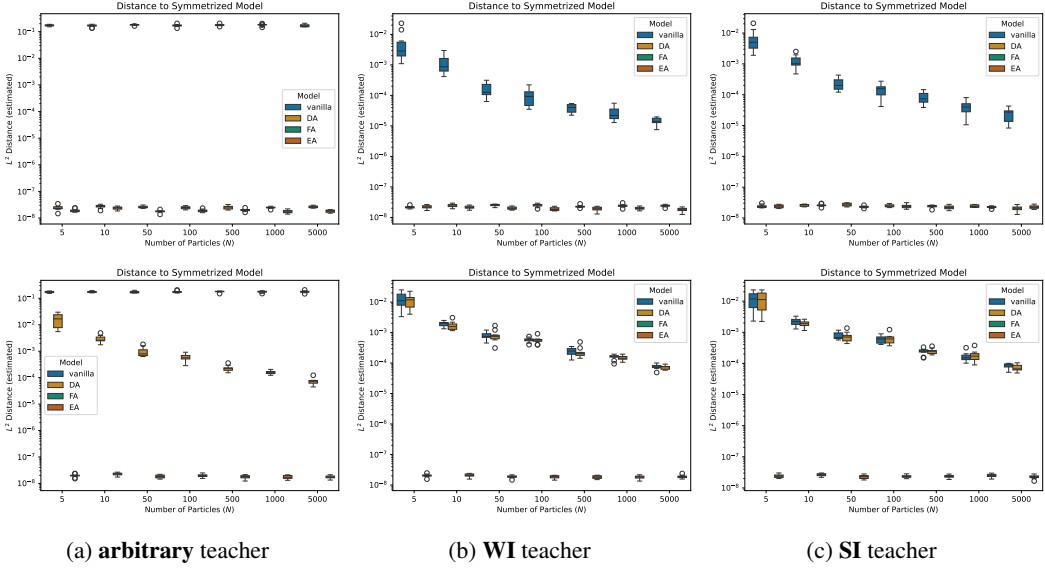

(a) **arbitrary** teacher          (b) **WI** teacher          (c) **SI** teacher

Figure 7: Approximation of $L^2$-distance between each model and its **symmetrized** version for increasing values of $N$. Each column corresponds to a different *teacher* as before. *Row 1* corresponds to the **SI**-initialized experiment and *Row 2* to the **WI**-initialized one.

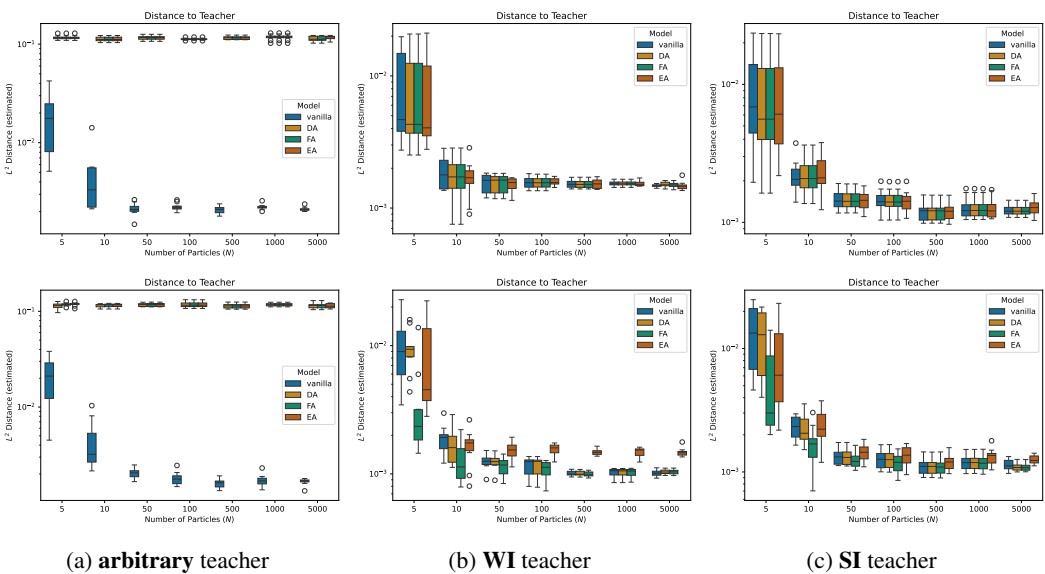

Figure 8: Approximation of $L^2$-distance between each model and the corresponding **teacher network** $f_*$, for increasing values of $N$. Each column corresponds to a different *teacher* as before; *Row 1* corresponds to the **SI**-initialized experiment and *Row 2* to the **WI**-initialized one.

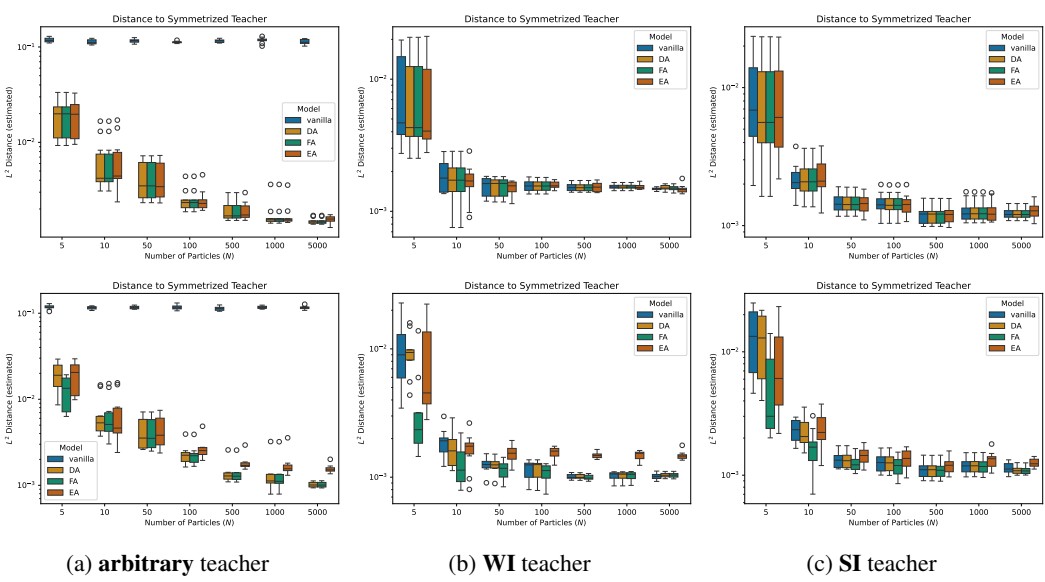

Figure 9: Approximation of $L^2$-distance between each model and the **symmetrized teacher network** $\mathcal{Q}_G.f_*$, for increasing values of $N$. Each column corresponds to a different *teacher* as before; *Row 1* corresponds to the **SI**-initialized experiment and *Row 2* to the **WI**-initialized one.

Now, beyond the analysis of the underlying particle distributions after training, we turn our focus to comparisons of the resulting **models**. We measure some distances in $L^2(\mathcal{X}, \mathcal{Y}; \pi_{\mathcal{X}})$, by approximating $\|\cdot\|_{L^2(\mathcal{X}, \mathcal{Y}; \pi_{\mathcal{X}})}$ with a Monte-Carlo sample of 100 random data points drawn from $\pi$.

Figure 7 shows that the observed behaviour for the underlying *particles*, $\nu_\theta^N$, of each model, is consistent with the behaviour of the obtained model $\Phi_\theta^N$. That is, as particles become *close* to being *symmetric* in some sense, the resulting *shallow model* also becomes increasingly equivariant as well.

Finally, Figure 8 and Figure 9 illustrate quite well the observations from Corollary 1. When our teacher **isn't equivariant**, models trained using any kind of SL technique end up suffering from the **inductive**

**bias** introduced by the symmetric assumption (something that's hinted by the symmetrization gap characterization from Lemma 2 presented in SuppMat-B.3). On the other hand, the **vanilla** model thrives in approximating $f_*$ as it is capable of breaking the assumed symmetry, unlike **DA**, **FA** and **EA**. The training regimes that use SL techniques are effectively approximating $\mathcal{Q}_G.f_*$, as shown in Figure 9 (and proven in Corollary 1). We also notice that, for the **WI**-initialized experiments, **EA**s end up suffering from their constraint of *staying within* $\mathcal{E}^G$, as they can't approximate $f_*$ (or $\mathcal{Q}_G.f_*$) as well as **DA** or **FA** (even when the teacher is **SI**). This isn't the case for the **SI**-initialized experiments, where the performance of **DA**, **FA** and **EA** (and **vanilla** only for equivariant $f_*$) is quite closely comparable (once again, hinting at Theorem 6). We also notice a general trend showing that, for bigger $N$, the approximations of $f_*$ (or, eventually, $\mathcal{Q}_G.f_*$) become increasingly *better* (specially for the **WI**-initialization).

### F.2 Heuristic algorithm for discovering EA parameter spaces

The proposed heuristic that we infer from the results on the previous experimental setting is quite thoroughly described in Section 4.2. We only notice that, for this particular setting, the *learning rate* was chosen to be $\alpha = 20$ (to better approximate the **MFL** conditions). Beyond the description of the proposed heuristic and the simple example visualized in Figure 2, we also provide Figure 10 here, illustrating a possible threshold choice in that setting.

Considering $E_j$ for $j = 0, 1, \ldots$ as the *spaces* that are discovered on each step of the heuristic, Figure 10 displays the values of: $\mathbf{RMD}^2(\nu_\theta^N, P_{E_j} \# \nu_\theta^N)$ and $\mathbf{RMD}^2(\nu_\theta^N, P_{\mathcal{E}^G} \# \nu_\theta^N)$; both before and after training on a given heuristic step $j$. The red line symbolizes a possible value of $\delta$ that could be fixed to **detect** whenever the obtained particle distribution *after training* **stayed** on $E_j$. In the case of this example, on steps 0 and 1 we would decide that *the training left the original space* $E_j$, but we wouldn't do so on step 2, allowing us to fix $\mathcal{E}^G := E_2$. As shown by the values of $\mathbf{RMD}^2(\nu_\theta^N, P_{\mathcal{E}^G} \# \nu_\theta^N)$, we wouldn't be *too far off* with our prediction.

Despite this proposed heuristic being potentially interesting for real-world applications, we acknowledge that the setting where it is applied here might be too *simple*, *synthetic* and *idealized*. On one hand, this provided a *clean-enough* framework, where the underlying phenoma could be easily observed. However, in order to properly validate our heuristic approach, experiments with *more complex settings* (and with larger and more intricate datasets) need to be performed. These should also be coupled with sound theoretical guarantees, whose exploration we leave for future work.

Finally, we here provide some further details on the possible connections of our proposed heuristic, to the 'symmetry-discovery' method presented in [72]. In their work, they employ an architecture based on *relaxed* group convolution layers, which allows to 'detect' breaks of the supposed data symmetry, by observing the *un-alignment of the layer weights*.

As in our work, their method starts with the most possibly constrained architecture: the null space in our case, and the 'perfectly aligned weights' in theirs. This ensures that the model will start respecting symmetry with respect to the largest possible group; only for the training on data to cause these symmetries to 'break' overtime. Their method seemingly works in a single training iteration, while ours iteratively constructs the invariant linear subspace $\mathcal{E}^G$ by adding one new 'symmetry-breaking dimension' at a time. Our Theorem 5 guarantees that, in the **MF** scale, our method **won't leave** $\mathcal{E}^G$; but we have yet to establish symmetry-breaking guarantees for our heuristic, comparable to Proposition 3.1 in [72].

Finally, it's important to note that neither one of the methods is *truly* discovering the "underlying symmetry" of the data. They are both closer to simply "optimizing architectures" which are compatible with data symmetries: either finding the "right" subspace $\mathcal{E}^G$, or the "right" weights for each group convolution filter. Identifying the true underlying structure of data symmetries is a much harder problem that is yet to be tackled in both cases.

## G Further theoretical insights

The following result provides consistency guarantees when the *regularization parameters $\tau$ and $\beta$* are small, and is a slight extension of a result in [38].

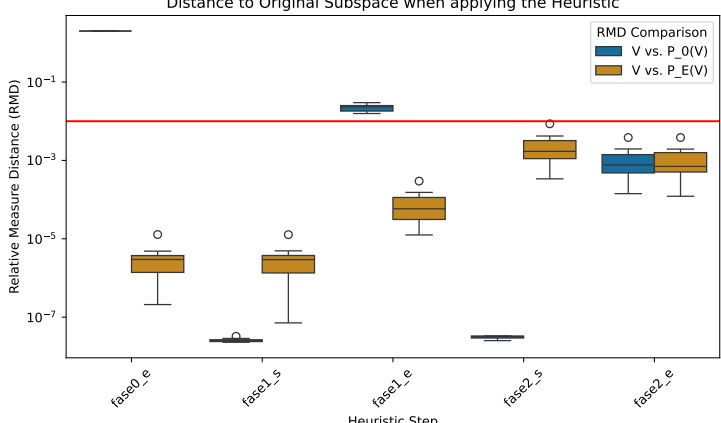

Figure 10: **RMD** comparison between the empirical student particle distribution, $\nu_\theta^N$, to both $P_{E_j} \# \nu_\theta^N$ and $(\nu_\theta^N)^{\mathcal{E}^G}$ (where $j$ is the heuristic step). These are performed at the beginning and the end of training on every fixed heuristic step. The red line is placed at the value $10^{-2}$ and represents a possible *threshold* $\delta$, to be used in the heuristic to determine whether *training left $E_j$* or not.

**Proposition 19** ($\Gamma$-**convergence**, as in ). *Let $\mathcal{Z} = \mathbb{R}^D$. If $R$ is $W_p$-continuous, $\nu$ is a Gibbs measure of potential $U$, and both $U$ and $r$ satisfy assumption 2 (or alternatively, are equal to 0), then $R_\nu^{\tau, \beta}$ $\Gamma$-converges to $R$ when $\tau, \beta \downarrow 0$. Particularly, given $\mu^{*, \tau, \beta, \nu}$ the minimizer of $R_\nu^{\tau, \beta}$, we have*

$$\overline{\lim_{\tau, \beta \to 0}} R(\mu^{*, \tau, \beta, \nu}) = \inf_{\mu \in \mathcal{P}_2(\mathcal{Z})} R(\mu).$$

*In particular, every cluster point of $(\mu^{*, \tau, \beta, \nu})_{\tau, \beta}$ is a minimizer of $R$.*

*Proof of Proposition 19.* We follow the exact same proof structure as [38], employing essentially their same techniques. However, we do adapt it to the case of taking the simultaneous limit of $\tau, \beta \to 0$, and so we do include it for completeness.

Let $(\tau_n)_{n \in \mathbb{N}}$ and $(\beta_n)_{n \in \mathbb{N}}$ be two positive sequences decreasing to 0. On the one hand, since $R$ is continuous (weakly if $p = 0$ or in $W_p$ for other $p \geq 1$) and $H_\nu(\mu) = D(\mu||\nu) \geq 0$, for all $\mu_n \to \mu$ (in the appropiate sense), we have

$$\lim \inf_{n \to +\infty} R_\nu^{\tau_n, \beta_n}(\mu_n) \geq \lim_{n \to +\infty} R(\mu_n) = R(\mu).$$

On the other hand, given $\mu \in \mathcal{P}_p(\mathcal{Z})$, consider $\rho$ to be the heat kernel in $\mathcal{Z} = \mathbb{R}^D$ and $\rho_n(x) := \beta_n^{-D} \nu(x/\beta_n)$. In particular, from [1] (as the heat kernel has finite $p$-th moments) we know that $\mu_n := \mu * \rho_n \xrightarrow[n \to \infty]{} \mu$ in $W_p$ (or weakly if it is the case).

Now, since the function $h(x) := x \log(x)$ is convex, from Jensen's inequality we get that

$$\int_{\mathcal{Z}} h(\mu * \rho_n) dx \leq \int_{\mathcal{Z}} \int_{\mathcal{Z}} h(\rho_n(x-y)) \, \mu(dy) dx = \int_{\mathcal{Z}} h(\rho_n(x)) dx = \int_{\mathcal{Z}} h(\rho(x)) dx - D \log(\sqrt{2\beta_n}),$$

Besides, we have (denoting here $g(x) = e^{-U(x)}$):

$$\int_{\mathcal{Z}} (\mu * \rho_n) \log(g) dx = -\int_{\mathcal{Z}} \mu(dy) \int_{\mathcal{Z}} \rho_n(x) U(x-y) dx \geq -C \left(1 + \int_{\mathcal{Z}} |y|^2 \mu(dy)\right).$$

The last inequality is due to the quadratic growth of $U$; and by the same argument on $r$:

$$\int_{\mathcal{Z}} (\mu * \rho_n) r dx = \int_{\mathcal{Z}} \mu(dy) \int_{\mathcal{Z}} \rho_n(x) r(x-y) dx \leq C \left(1 + \int_{\mathcal{Z}} |y|^2 \mu(dy)\right).$$

Notice that whenever $U \equiv 0$ or $r \equiv 0$, despite them not satisfying assumption 2, we still get the same inequalities (since the leftmost term would be 0).

Now, as $R$ is $W_p$-continuous, $R(\mu_n) \xrightarrow[n\to\infty]{} R(\mu)$, and:

$$\limsup_{n\to+\infty} R_\nu^{\tau_n,\beta_n}(\mu * \nu_n)$$

$$\leq R(\mu) + \limsup_{n\to+\infty} \tau_n \left( \int_{\mathcal{Z}} (\mu * \rho_n) r dx \right)$$

$$+ \limsup_{n\to+\infty} \beta_n \left( \int_{\mathcal{Z}} h(\mu * \rho_n) dx - \int_{\mathcal{Z}} (\mu * \rho_n) \log(g) dx \right)$$

And, as $\lim_{n\to\infty} \beta_n \log(\sqrt{2\beta_n}) = 0$ and the rest of the terms are bounded, we conclude that:

$$\limsup_{n\to+\infty} R_\nu^{\tau_n,\beta_n}(\mu * \rho_n) \leq R(\mu)$$

In particular, denoting by $\mu_*^{\tau,\beta,\nu}$ the unique minimizer of $R_\nu^{\tau,\beta}$, then from the previous expressions we get $\forall n \in \mathbb{N}$ and $\forall \mu \in \mathcal{P}_p(\mathcal{Z})$:

$$R(\mu_*^{\tau_n,\beta_n,\nu}) \leq R_\nu^{\tau_n,\beta_n}(\mu_*^{\tau_n,\beta_n,\nu}) \leq R_\nu^{\tau_n,\beta_n}(\mu * \rho_n)$$

So that,

$$\limsup_{n\to\infty} R(\mu_*^{\tau_n,\beta_n,\nu}) \leq \limsup_{n\to+\infty} R_\nu^{\tau_n,\beta_n}(\mu * \rho_n) \leq R(\mu), \quad \text{for all } \mu \in P_2(\mathcal{Z}).$$

$\square$

Finally, we provide, for completeness, a proof of Lemma 2 :

*Proof of Lemma 2 (based on [27] and [40]).* As $H \leq G$ is a compact group and $\pi$ is $H$-invariant, from proposition 10 we know that $f^* = \mathbb{E}_\pi[Y|X = \cdot]$ lives in $f^* \in L_H^2(\mathcal{X}, \mathcal{Y}; \pi|_\mathcal{X})$. Consider any $f \in L^2(\mathcal{X}, \mathcal{Y}; \pi|_\mathcal{X})$, by Lemma 1 from [27] (which applies since $\pi|_\mathcal{X}$ is $G$-invariant), we can decompose $f$ as $f = \overline{f}_G + f_G^\perp$, where $\overline{f}_G = \mathcal{Q}_G f$ is its symmetric part and $f_G^\perp = f - \mathcal{Q}_G f$ its antisymmetric part. A standard calculation of the population risk under the quadratic loss setting (see the proof of Corollary 1 for further insight) gives: $R(f) = R_* + \|f - f^*\|_{L^2(\mathcal{X},\mathcal{Y};\pi_\mathcal{X})}^2$, and so:

$$\Delta(f, \mathcal{Q}_G.f) = R(f) - R(\mathcal{Q}_G.f) = \mathbb{E}\left[\|f^*(X) - f(X)\|_\mathcal{Y}^2\right] - \mathbb{E}\left[\|f^*(X) - \overline{f}_G(X)\|_\mathcal{Y}^2\right],$$

which can be written as:

$$\Delta(f, \mathcal{Q}_G f) = \mathbb{E}_\pi\left[\|f^*(X) - \overline{f}_G(X)\|_\mathcal{Y}^2 - 2\langle f^*(X) - \overline{f}_G(X), f_G^\perp(X)\rangle_\mathcal{Y} + \|f_G^\perp(X)\|_\mathcal{Y}^2\right]$$

$$- \mathbb{E}_\pi\left[\|f^*(X) - \overline{f}_G(X)\|_\mathcal{Y}^2\right]$$

$$= -2\langle f^* - \overline{f}_G, f_G^\perp\rangle_{L^2(\mathcal{X},\mathcal{Y};\pi_\mathcal{X})} + \|f_G^\perp\|_{L^2(\mathcal{X},\mathcal{Y};\pi_\mathcal{X})}^2$$

$$= -2\langle f^*, f_G^\perp\rangle_{L^2(\mathcal{X},\mathcal{Y};\pi_\mathcal{X})} + \|f_G^\perp\|_{L^2(\mathcal{X},\mathcal{Y};\pi_\mathcal{X})}^2$$

Where we used that $\langle \overline{f}_G, f_G^\perp\rangle_{L^2(\mathcal{X},\mathcal{Y};\pi_\mathcal{X})} = 0$. The first term on the right hand side, $-2\langle f^*, f_G^\perp\rangle_{L^2(\mathcal{X},\mathcal{Y};\pi_\mathcal{X})}$, is what [40] call the *mismatch* between the real *underlying model* (which is only $H$-equivariant) and the *symmetrized* version of our model (which is made entirely $G$-equivariant).

Now, when $\pi$ is $G$-equivariant, by proposition 10, $\mathcal{Q}_G f^* = f^*$, and so: $-2\langle f^*, f_G^\perp\rangle_{L^2(\mathcal{X},\mathcal{Y};\pi_\mathcal{X})} = 0$, giving us the desired result:

$$\Delta(f, \mathcal{Q}_G f) = \|f_G^\perp\|_{L^2(\mathcal{X},\mathcal{Y};\pi_\mathcal{X})}^2$$

$\square$

Lemma 2 essentially says that if we *try to symmetrize* a model with respect to a group that has *'more symmetries'* than what are actually observable in our data (i.e. $\pi$ in itself is only $H$-invariant, but we symmetrize with respect to $G \geq H$); we can either *win* or *lose* generalization power according to the interplay between the two presented terms. In particular, if $\pi$ is $G$ equivariant, there's a *strict generalization benefit* from choosing a *symmetric model* to tackle our *learning problem* (which gives the name to the paper [27]). In particular, whenever $f_G^\perp$ is non-zero (on a strictly positive $\pi|_\mathcal{X}$-measure set) there's a **strict gain** in generalization power from using the symmetrized version of the model.

# NeurIPS Paper Checklist

1. **Claims**

   Question: Do the main claims made in the abstract and introduction accurately reflect the paper's contributions and scope?

   Answer: [Yes]

   Justification: Within the limited space we could consecrate to those sections, both the abstract and the introduction give quite accurate descriptions of the context of the work (Mean-Field approach to overparametrized NN), the questions and problems addressed (role of symmetries and impact of symmetry-leveraging techniques, as seen from the Mean-Field limit), our setting (the type of neural network models considered), the specific results obtained (as precisely as it was possible in words, namely the precise description of the Wasserstein gradient flows describing the limiting training dynamics in different settings), the novelty of the paper (what can be learned from this viewpoint), and the contents of our numerical experiments. The main results are clearly understandable from the abstract, and the introduction gives a rather general description of the paper's goals, since it also leaves room for the context description and the outline of the paper. See Section 1 for details.

   Guidelines:
   - The answer NA means that the abstract and introduction do not include the claims made in the paper.
   - The abstract and/or introduction should clearly state the claims made, including the contributions made in the paper and important assumptions and limitations. A No or NA answer to this question will not be perceived well by the reviewers.
   - The claims made should match theoretical and experimental results, and reflect how much the results can be expected to generalize to other settings.
   - It is fine to include aspirational goals as motivation as long as it is clear that these goals are not attained by the paper.

2. **Limitations**

   Question: Does the paper discuss the limitations of the work performed by the authors?

   Answer: [Yes]

   Justification: We acknowledge and discuss many of the key assumptions required for both our theoretical and experimental results. The need for these assumption limits the applicability and practical impact of our work. We explicitly provide relevant insights in this regard, in a dedicated section, SuppMat-A.2. Also, and without including every single instance, we also discuss our limitations in: assumption 1, after presenting equation (1), section 4, and in many remarks in the SuppMat (e.g. in E.2.5,E.3 or F).

   Guidelines:
   - The answer NA means that the paper has no limitation while the answer No means that the paper has limitations, but those are not discussed in the paper.
   - The authors are encouraged to create a separate "Limitations" section in their paper.
   - The paper should point out any strong assumptions and how robust the results are to violations of these assumptions (e.g., independence assumptions, noiseless settings, model well-specification, asymptotic approximations only holding locally). The authors should reflect on how these assumptions might be violated in practice and what the implications would be.
   - The authors should reflect on the scope of the claims made, e.g., if the approach was only tested on a few datasets or with a few runs. In general, empirical results often depend on implicit assumptions, which should be articulated.
   - The authors should reflect on the factors that influence the performance of the approach. For example, a facial recognition algorithm may perform poorly when image resolution is low or images are taken in low lighting. Or a speech-to-text system might not be used reliably to provide closed captions for online lectures because it fails to handle technical jargon.
   - The authors should discuss the computational efficiency of the proposed algorithms and how they scale with dataset size.

- If applicable, the authors should discuss possible limitations of their approach to address problems of privacy and fairness.
- While the authors might fear that complete honesty about limitations might be used by reviewers as grounds for rejection, a worse outcome might be that reviewers discover limitations that aren't acknowledged in the paper. The authors should use their best judgment and recognize that individual actions in favor of transparency play an important role in developing norms that preserve the integrity of the community. Reviewers will be specifically instructed to not penalize honesty concerning limitations.

3. **Theory Assumptions and Proofs**

Question: For each theoretical result, does the paper provide the full set of assumptions and a complete (and correct) proof?

Answer: [Yes]

Justification: All mathematical statements presented are given complete, rigorous mathematical proofs in the Supplementary Material (mainly in SuppMat-E). We also provide complete sets of assumptions for each result, either in the statement itself or (for some results where the precise assumptions are very technical and lengthy to state) previous to the corresponding proof in the Supplementary Material. Even proofs of some elementary facts or statements are provided for completeness. Furthermore, nearly every mathematical result stated in the main paper is given a comment or discussion regarding its context, its interpretation (beyond the pure mathematical value) and/or an idea of what tools or arguments its proof will rely upon.

Guidelines:

- The answer NA means that the paper does not include theoretical results.
- All the theorems, formulas, and proofs in the paper should be numbered and cross-referenced.
- All assumptions should be clearly stated or referenced in the statement of any theorems.
- The proofs can either appear in the main paper or the supplemental material, but if they appear in the supplemental material, the authors are encouraged to provide a short proof sketch to provide intuition.
- Inversely, any informal proof provided in the core of the paper should be complemented by formal proofs provided in appendix or supplemental material.
- Theorems and Lemmas that the proof relies upon should be properly referenced.

4. **Experimental Result Reproducibility**

Question: Does the paper fully disclose all the information needed to reproduce the main experimental results of the paper to the extent that it affects the main claims and/or conclusions of the paper (regardless of whether the code and data are provided or not)?

Answer: [Yes]

Justification: Together with our submission, we include a ZIP file with the necessary code (and instructions, following the guidelines provided by the conference) for replicating all of our experimental results. We also provide the necessary details to understand and reproduce our work both in Section 4 and (more extensively) in SuppMat-F.

Guidelines:

- The answer NA means that the paper does not include experiments.
- If the paper includes experiments, a No answer to this question will not be perceived well by the reviewers: Making the paper reproducible is important, regardless of whether the code and data are provided or not.
- If the contribution is a dataset and/or model, the authors should describe the steps taken to make their results reproducible or verifiable.
- Depending on the contribution, reproducibility can be accomplished in various ways. For example, if the contribution is a novel architecture, describing the architecture fully might suffice, or if the contribution is a specific model and empirical evaluation, it may be necessary to either make it possible for others to replicate the model with the same dataset, or provide access to the model. In general. releasing code and data is often

one good way to accomplish this, but reproducibility can also be provided via detailed instructions for how to replicate the results, access to a hosted model (e.g., in the case of a large language model), releasing of a model checkpoint, or other means that are appropriate to the research performed.

- While NeurIPS does not require releasing code, the conference does require all submissions to provide some reasonable avenue for reproducibility, which may depend on the nature of the contribution. For example
    (a) If the contribution is primarily a new algorithm, the paper should make it clear how to reproduce that algorithm.
    (b) If the contribution is primarily a new model architecture, the paper should describe the architecture clearly and fully.
    (c) If the contribution is a new model (e.g., a large language model), then there should either be a way to access this model for reproducing the results or a way to reproduce the model (e.g., with an open-source dataset or instructions for how to construct the dataset).
    (d) We recognize that reproducibility may be tricky in some cases, in which case authors are welcome to describe the particular way they provide for reproducibility. In the case of closed-source models, it may be that access to the model is limited in some way (e.g., to registered users), but it should be possible for other researchers to have some path to reproducing or verifying the results.

5. **Open access to data and code**

Question: Does the paper provide open access to the data and code, with sufficient instructions to faithfully reproduce the main experimental results, as described in supplemental material?

Answer: [Yes]

Justification: As mentioned in our previous answer, we include a ZIP file with the necessary code and instructions to faithfully reproduce the main experimental results. We closely follow the NeurIPS guidelines for our code submission, providing the exact commands and environments required to run our experiments. Also both Section 4 and SuppMat-F provide a thorough landscape of the necessary assumptions and parameters to consider when replicating our results.

Guidelines:

- The answer NA means that paper does not include experiments requiring code.
- Please see the NeurIPS code and data submission guidelines (`https://nips.cc/public/guides/CodeSubmissionPolicy`) for more details.
- While we encourage the release of code and data, we understand that this might not be possible, so "No" is an acceptable answer. Papers cannot be rejected simply for not including code, unless this is central to the contribution (e.g., for a new open-source benchmark).
- The instructions should contain the exact command and environment needed to run to reproduce the results. See the NeurIPS code and data submission guidelines (`https://nips.cc/public/guides/CodeSubmissionPolicy`) for more details.
- The authors should provide instructions on data access and preparation, including how to access the raw data, preprocessed data, intermediate data, and generated data, etc.
- The authors should provide scripts to reproduce all experimental results for the new proposed method and baselines. If only a subset of experiments are reproducible, they should state which ones are omitted from the script and why.
- At submission time, to preserve anonymity, the authors should release anonymized versions (if applicable).
- Providing as much information as possible in supplemental material (appended to the paper) is recommended, but including URLs to data and code is permitted.

6. **Experimental Setting/Details**

Question: Does the paper specify all the training and test details (e.g., data splits, hyperparameters, how they were chosen, type of optimizer, etc.) necessary to understand the results?

Answer: [Yes]

Justification: In Section 4 we provide the necessary level of detail for understanding our main experimental results. We provide a thorough and detailed description of the exact parameters (and how they were chosen) employed in our different experiments in SuppMat-F.

Guidelines:

- The answer NA means that the paper does not include experiments.
- The experimental setting should be presented in the core of the paper to a level of detail that is necessary to appreciate the results and make sense of them.
- The full details can be provided either with the code, in appendix, or as supplemental material.

7. **Experiment Statistical Significance**

Question: Does the paper report error bars suitably and correctly defined or other appropriate information about the statistical significance of the experiments?

Answer: [Yes]

Justification: The numerical experiments considered in this work, involved the application of a noisy SGD dynamic (as in equation (1), or equation (5)) that relied on various levels of randomness: the initial parameter configurations, the simulated training data and the gaussian noise in SGD iterations. These factors are taken into account (see SuppMat-F) and thus the different experiments were repeated a total of $10$ times (under different random seeds), which was largely enough to include interpretable error bars in our plots, and to correctly illustrate the true trends of the underlying phenomena (see, again, SuppMat-F).

Guidelines:

- The answer NA means that the paper does not include experiments.
- The authors should answer "Yes" if the results are accompanied by error bars, confidence intervals, or statistical significance tests, at least for the experiments that support the main claims of the paper.
- The factors of variability that the error bars are capturing should be clearly stated (for example, train/test split, initialization, random drawing of some parameter, or overall run with given experimental conditions).
- The method for calculating the error bars should be explained (closed form formula, call to a library function, bootstrap, etc.)
- The assumptions made should be given (e.g., Normally distributed errors).
- It should be clear whether the error bar is the standard deviation or the standard error of the mean.
- It is OK to report 1-sigma error bars, but one should state it. The authors should preferably report a 2-sigma error bar than state that they have a 96% CI, if the hypothesis of Normality of errors is not verified.
- For asymmetric distributions, the authors should be careful not to show in tables or figures symmetric error bars that would yield results that are out of range (e.g. negative error rates).
- If error bars are reported in tables or plots, The authors should explain in the text how they were calculated and reference the corresponding figures or tables in the text.

8. **Experiments Compute Resources**

Question: For each experiment, does the paper provide sufficient information on the computer resources (type of compute workers, memory, time of execution) needed to reproduce the experiments?

Answer: [Yes]

Justification: In SuppMat-F we indicate both the computer resources employed to run the different experiments (see the first paragraph) as well as an estimate of the total computer time for the most complex simulation considered (namely, the $N = 5000$ case). We did not make related experiments prior to the preparation of this submission.

Guidelines:

- The answer NA means that the paper does not include experiments.
- The paper should indicate the type of compute workers CPU or GPU, internal cluster, or cloud provider, including relevant memory and storage.
- The paper should provide the amount of compute required for each of the individual experimental runs as well as estimate the total compute.
- The paper should disclose whether the full research project required more compute than the experiments reported in the paper (e.g., preliminary or failed experiments that didn't make it into the paper).

9. **Code Of Ethics**

Question: Does the research conducted in the paper conform, in every respect, with the NeurIPS Code of Ethics `https://neurips.cc/public/EthicsGuidelines`?

Answer: [Yes]

Justification: This paper rigorously conforms to the NeurIPS Code of Ethics. By its mathematical/theoretical nature, this paper did not require the interaction with humans (besides the authors), and did not expose any living being to any type of harm. Also, no direct or indirect social impact is expected from this work in the short-to-mid term. Since our numerical results come from simulations, there are no major concerns to be had related to the nature of the employed dataset (in terms of privacy, consent, fair use or representative evaluation). Also, Intellectual property in all its forms was thoroughly respected. Finally, all the elements required for reproducibility of our results are disclosed in the material joint to the submission, including software employed and the data-simulation mechanisms.

Guidelines:

- The answer NA means that the authors have not reviewed the NeurIPS Code of Ethics.
- If the authors answer No, they should explain the special circumstances that require a deviation from the Code of Ethics.
- The authors should make sure to preserve anonymity (e.g., if there is a special consideration due to laws or regulations in their jurisdiction).

10. **Broader Impacts**

Question: Does the paper discuss both potential positive societal impacts and negative societal impacts of the work performed?

Answer: [No]

Justification: The research object of this paper is mathematical, as it studies abstract mathematical models of artificial neural networks, and their properties and behaviors in specific, idealized situations and contexts. Therefore, no direct or indirect social impact is expected in the short-to-mid term. It is, however, expected that the theoretical results presented could have a positive impact in the long term, at the levels of development or practical use of safer, more transparent or interpretable machine learning algorithms; or at least in enlarging the corpus of conceptual tools available to researchers and practitioners to better understand the advantages, drawbacks, and potential risks or impacts of their professional activities. Though it might be useful eventually, our work is mostly theoretical and its impact remains limited to academia and pure ML practice. By all these reasons, no potential, positive nor negative, societal impacts of the work performed are addressed in the paper.

Guidelines:

- The answer NA means that there is no societal impact of the work performed.
- If the authors answer NA or No, they should explain why their work has no societal impact or why the paper does not address societal impact.
- Examples of negative societal impacts include potential malicious or unintended uses (e.g., disinformation, generating fake profiles, surveillance), fairness considerations (e.g., deployment of technologies that could make decisions that unfairly impact specific groups), privacy considerations, and security considerations.
- The conference expects that many papers will be foundational research and not tied to particular applications, let alone deployments. However, if there is a direct path to any negative applications, the authors should point it out. For example, it is legitimate

to point out that an improvement in the quality of generative models could be used to generate deepfakes for disinformation. On the other hand, it is not needed to point out that a generic algorithm for optimizing neural networks could enable people to train models that generate Deepfakes faster.

- The authors should consider possible harms that could arise when the technology is being used as intended and functioning correctly, harms that could arise when the technology is being used as intended but gives incorrect results, and harms following from (intentional or unintentional) misuse of the technology.
- If there are negative societal impacts, the authors could also discuss possible mitigation strategies (e.g., gated release of models, providing defenses in addition to attacks, mechanisms for monitoring misuse, mechanisms to monitor how a system learns from feedback over time, improving the efficiency and accessibility of ML).

11. **Safeguards**

Question: Does the paper describe safeguards that have been put in place for responsible release of data or models that have a high risk for misuse (e.g., pretrained language models, image generators, or scraped datasets)?

Answer: [NA]

Justification: As previously mentioned, this work is mainly theoretical, and thus poses no major threats that could require the implementation of such safeguards.

Guidelines:

- The answer NA means that the paper poses no such risks.
- Released models that have a high risk for misuse or dual-use should be released with necessary safeguards to allow for controlled use of the model, for example by requiring that users adhere to usage guidelines or restrictions to access the model or implementing safety filters.
- Datasets that have been scraped from the Internet could pose safety risks. The authors should describe how they avoided releasing unsafe images.
- We recognize that providing effective safeguards is challenging, and many papers do not require this, but we encourage authors to take this into account and make a best faith effort.

12. **Licenses for existing assets**

Question: Are the creators or original owners of assets (e.g., code, data, models), used in the paper, properly credited and are the license and terms of use explicitly mentioned and properly respected?

Answer: [Yes]

Justification: We employ mostly public ML libraries and packages (e.g. **objax**, **pyot**, among others). We do, however, also involve the use of the **emlp** package provided by [29]. These assets are all properly credited in our work (see SuppMat-F), and the corresponding use licenses (particularly for the use of **emlp**) are properly mentioned and respected.

Guidelines:

- The answer NA means that the paper does not use existing assets.
- The authors should cite the original paper that produced the code package or dataset.
- The authors should state which version of the asset is used and, if possible, include a URL.
- The name of the license (e.g., CC-BY 4.0) should be included for each asset.
- For scraped data from a particular source (e.g., website), the copyright and terms of service of that source should be provided.
- If assets are released, the license, copyright information, and terms of use in the package should be provided. For popular datasets, `paperswithcode.com/datasets` has curated licenses for some datasets. Their licensing guide can help determine the license of a dataset.
- For existing datasets that are re-packaged, both the original license and the license of the derived asset (if it has changed) should be provided.

- If this information is not available online, the authors are encouraged to reach out to the asset's creators.

13. **New Assets**

Question: Are new assets introduced in the paper well documented and is the documentation provided alongside the assets?

Answer: [NA]

Justification: Beyond the provided code to reproduce the experiments presented in the paper (which is accompanied with the details and instructions needed to execute it), no assets are introduced in this publication, hence, no specific documentation needs to be provided in that regard.

Guidelines:

- The answer NA means that the paper does not release new assets.
- Researchers should communicate the details of the dataset/code/model as part of their submissions via structured templates. This includes details about training, license, limitations, etc.
- The paper should discuss whether and how consent was obtained from people whose asset is used.
- At submission time, remember to anonymize your assets (if applicable). You can either create an anonymized URL or include an anonymized zip file.

14. **Crowdsourcing and Research with Human Subjects**

Question: For crowdsourcing experiments and research with human subjects, does the paper include the full text of instructions given to participants and screenshots, if applicable, as well as details about compensation (if any)?

Answer: [NA]

Justification: This paper does not involve crowdsourcing nor research with human subjects.

Guidelines:

- The answer NA means that the paper does not involve crowdsourcing nor research with human subjects.
- Including this information in the supplemental material is fine, but if the main contribution of the paper involves human subjects, then as much detail as possible should be included in the main paper.
- According to the NeurIPS Code of Ethics, workers involved in data collection, curation, or other labor should be paid at least the minimum wage in the country of the data collector.

15. **Institutional Review Board (IRB) Approvals or Equivalent for Research with Human Subjects**

Question: Does the paper describe potential risks incurred by study participants, whether such risks were disclosed to the subjects, and whether Institutional Review Board (IRB) approvals (or an equivalent approval/review based on the requirements of your country or institution) were obtained?

Answer: [NA]

Justification: This paper does not involve crowdsourcing nor research with human subjects.

Guidelines:

- The answer NA means that the paper does not involve crowdsourcing nor research with human subjects.
- Depending on the country in which research is conducted, IRB approval (or equivalent) may be required for any human subjects research. If you obtained IRB approval, you should clearly state this in the paper.
- We recognize that the procedures for this may vary significantly between institutions and locations, and we expect authors to adhere to the NeurIPS Code of Ethics and the guidelines for their institution.
- For initial submissions, do not include any information that would break anonymity (if applicable), such as the institution conducting the review.

