# OpenReview forum: "Symmetries in Overparametrized Neural Networks: A Mean Field View"
_NeurIPS.cc/2024/Conference — NeurIPS 2024 spotlight_

### Official Review · Reviewer_aSNH · 2024-06-21

**Soundness:** 4
**Presentation:** 3
**Contribution:** 4
**Rating:** 7
**Confidence:** 2

**Summary:**

This paper studies the Mean-Field limit of generalized shallow neural networks after learning with Wasserstein gradient flow under data augmentation, feature averaging or equivariant architectures as well as standard training. The provided results provide insights into learning with symmetries by covering two notions of invariant laws on the parameter space, varying assumptions on the data distribution and preserved symmetry over the course of training.

**Strengths:**

The paper provides several deep and quite general theoretical results on learning with symmetries in the mean-field limit. Although the results are quite technical, the authors manage to present them in good clarity and embedded in the related literature of mean field limits of shallow models. Considering the examples of Data Augmentation, Feature Averaging and Equivariant Architectures seems natural and provides several interesting insights, for example the equivalence of DA, FA and even free training under symmetric data.

**Weaknesses:**

The paper already seems quite complete and well written and I could not identify significant weaknesses. The following weaknesses are only minor points:

- The description of Figure 1 is hard to parse. Markers like (a)-(d) could reduce ambiguity of what is meant by a column.
- The markers in Figure 2 should be larger and it is unclear to me which additional insight to draw from the second row.

As I am not an expert in the mean-field and Wasserstein gradient flow literature, I did not verify the correctness of all proofs, and did not double-check the claimed novelty.

**Typos:** 22: lastly, 65 and 66: b=N?, 75: Definition name in bold for consistency?, 120: […], 275: will, 329: a heuristic

**Questions:**

- How close is your setting to practical neural networks trained with SGD? In future work, are there hopes to extend your mean-field theory results to deep networks or to SGD with large learning rates and edge of stability dynamics? Similarly is the entropy regularization merely theoretically pleasing or also practically feasible?
- Can GNNs or Transformers also be encoded in your class of equivariant architectures? If so, these example would significantly broaden the target audience.
- In future work, in which ways could Assumption 1 be relaxed?

**Limitations:**

The authors adequately acknowledge the main limitations of the paper. These limitations lie beyond the scope of this paper, but pose interesting directions for future work:

- The provided results are only asymptotic and only consider gradient flow on shallow models as opposed to edge of stability dynamics in finite deep models.
- The practical relevance of the findings and proposed algorithm at finite width for practical architectures, datasets and optimization procedures could be empirically verified. In particular, does vanilla training remain SI throughout training beyond the simple student-teacher setting?

---

> ### Author Rebuttal · Authors · 2024-08-06
>
> We thank the reviewer for their thorough reading and the relevant comments and suggestions.
> Regarding the detected weaknesses of the paper, we will make sure to address them by providing clearer descriptions of the relevant figures, and correcting the detected typos. For instance, we shall complement the description of Figure 2, emphasizing the role of the second row as a “parallel” view of the plane ${\cal E}^G$, that allows us to visually check that particles indeed remain inside it (see Figure 5 for an example where this doesn’t occur).
> We next address the questions posed by the reviewer:
>
> **Q1: Regarding practical NNs, the entropy term in practice, and extensions to deep networks and edge-of-stability (EoS) dynamics.**
> - In our Global Response, we provide a deeper discussion regarding the applicability of our setting to practical NNs. Namely, we stress that the MF limit is closer to real applications than other asymptotic regimes, and that our numerical experiments correspond to quite “practical” settings (a “usual” single-hidden-layer NN architecture trained with the “usual” SGD). That is, despite our results being “asymptotic”, we believe they are applicable to reasonably practical NN implementations (with relatively normal width $N$).
> - Regarding the entropy-regularization, despite it being mostly a theoretical tool to ensure uniqueness and global convergence, it also corresponds in practice to considering a “noisy/Langevin” SGD dynamics. Indeed, when taking the limit $N\to \infty$, the presence of properly scaled noise in the SGD dynamic exactly translates to the appearance of the entropy term in the WGF. Therefore, in a sense, the entropy term is truly realized in practice through the introduction of noise.
> - We most surely hope to extend our results to general deep neural networks, for which we believe that some of the symmetry-related theorems (Section 3.3) might still hold. As discussed in our Global Response, however, this requires a more mature (and unified) mean-field theory of deep neural networks, which is yet to be established.
> - We haven’t specifically looked into SGD with large learning rates and EoS dynamics, but it indeed seems to be an interesting direction for future work. Now, we know that some of our proofs (e.g. for Theorems 3 and 4) can be applied in the setting of works like [18], where the MFL is established under a fixed learning rate (which may be large, and doesn’t necessarily decrease with N). This can be seen as a first preliminary result in the suggested direction.
>
> **Q2: Regarding applicability on GNNs and Transformers.**
>
> As mentioned to other reviewers (see Global Response), very commonly applied equivariant architectures (such as CNNs and GNNs) can be exactly modeled in our setting: either as “shallow” models, or as large ensembles of multilayer models; and such that ${\cal E}^G$ exactly encodes the desired EA. To clarify this point, we will include a section in the Supplementary Material where some of these relevant examples will be explicitly modeled in our framework (e.g. a “shallow” GNN, under the right action of $G = S_n$ on $\cal Z$).
>
> As for Transformers, it is well known that they can be seen as a particular type of GNN; and so, in particular, they can also be modeled using our framework (again, as "shallow" Transformers or also large ensembles of Transformers).
>
> **Q3: Regarding Assumption 1.**
>
> We believe that only a couple of our assumptions are truly fundamental for achieving our results; namely, the convexity and joint invariance of $\ell$, and the joint-equivariance of $\sigma_*$. On the other hand, some of the more “technical” assumptions we believe can be relaxed. For instance, it is known (see e.g. [9, 51]) that the boundedness of $\sigma_*$ can be replaced by assumptions on the data distribution $\pi$ (e.g. bounded “finite-moments”, or compact support). Also, passing through the Wasserstein sub-Gradient Flow (as in [13]), and considering weak solutions, we believe that some of the differentiability constraints on $\ell$ and $\sigma_*$ could be potentially lifted.
>
> **Regarding the Limitations**
>
> Generally speaking, we plan on tackling many of the detected limitations of our results in upcoming works. Namely, we will be sure to continue experimenting with our results (particularly, Theorems 5 and 6) on more realistic and practical datasets (i.e. beyond the simple student-teacher setting). We also comment on the applicability of our work in our General Response.

---

> > ### Comment · Reviewer_aSNH · 2024-08-09
> > **Thank you**
> >
> > I thank the authors for their thoughtful response and thorough work. I am satisfied with the changes promised in the global response and I will keep my positive evaluation.

---

> > > ### Author Response · Authors · 2024-08-09
> > >
> > > We thank the reviewer for their comment, and are very happy for their positive evaluation. Thanks again!

---

### Official Review · Reviewer_4cbm · 2024-06-28

**Soundness:** 3
**Presentation:** 3
**Contribution:** 3
**Rating:** 7
**Confidence:** 3

**Summary:**

The paper investigates the learning dynamics of neural networks with several symmetry-leveraging techniques from a mean-field perspective. The main result indicates that optimizations with data augmentation and feature averaging (and the corresponding Wasserstein gradient flows) are equivalent in the mean-field limit under mild assumptions, while equivariant architecture may suffer from limited expressivity. Moreover, for equivariant data, freely-trained models also have the mean-field flow remaining in the set of strongly invariant distributions if initialized there, which is supported by the experiments. Finally, the authors propose a heuristic to discover equivariant architectures by gradually expanding the equivariant subspace.

**Strengths:**

* The math in the paper is rigorous and detailed. Complete proofs and abundant examples and remarks are provided. As more of a practitioner myself, I can understand the main ideas of the paper without much difficulty, though I did not check the proofs carefully.
* The experiments are compact and well-designed, clearly supporting the theoretical results.

**Weaknesses:**

* This is not essentially a weakness, but the theoretical results don't bring many interesting insights to me. They mostly match my existing intuitions. E.g. for equivariant data, freely-trained models tend to have the same training dynamics as symmetry-aware models. I will leave it to other reviewers to comment on the theoretical significance of these results.

* I suggest the authors provide more detailed explanations about some mathematical terminology, e.g. the pushforward of a probability measure; dt-a.e. standing for almost everywhere wrt dt. This could make the paper more accessible to the general audience.

**Questions:**

* Can you explain what is the connection between jointly equivariant and equivariant? The way I understand it is that we have some group actions on $\mathcal Z$ induced from the definition of joint equivariance, i.e. $\sigma_* (\rho_g x, M_g z) = \hat\rho_g \sigma_*(x,z)$. In other words, $M_g$ is determined by the group action on data and the model architecture. Then, if $M_g z = z$, then a NN model parametrized by $z$ is equivariant. Is this correct?
* Proposition 5 states that equivariant architectures can lead to optimal solutions provided some universality conditions. How restrictive are these conditions?
* Is the property of data equivariance (defined in L150) equivalent to the marginal distribution on $\mathcal X$ being invariant and the function $f: \mathcal X \to \mathcal Y$ being equivariant, given that the mapping from $\mathcal X$ to $\mathcal Y$ is deterministic? If so, would it be too restrictive to assume the invariance of $\pi_\mathcal X$?
* In Section 4.1, the authors show that SI-initialized but freely-updated models remain in $\mathcal E^G$ for large $N$s. This is indeed an interesting phenomenon. I wonder if this is related to the exact WI-initialized teachers. Can you provide, in addition to the arbitrary particles in Appendix D, the WI-initialized particles? Most likely, sampling from the pushed-forward empirical measure would result in a non-symmetric teacher, but there's a chance that the WI teacher might contain two pairs of symmetric parameters (e.g. $\theta_1^*, g\theta_1^*, \theta_2^*, g\theta_2^*$) and another singled out, making it mostly equivariant. After all, if the teacher is not equivariant, then the data is not equivariant, and the assumptions in Theorem 5 do not hold.
* Section 4.2 reminds me of another work on discovering symmetry [1], which initially specifies a large candidate group and discovers the subgroup symmetry by inspecting the weights of relaxed group convolution filters. Can you comment on the connection between your method and theirs?

## References
[1] Wang, Rui, et al. "Discovering Symmetry Breaking in Physical Systems with Relaxed Group Convolution." Forty-first International Conference on Machine Learning.

**Limitations:**

The authors have discussed the limitations.

---

> ### Author Rebuttal · Authors · 2024-08-06
>
> We thank the reviewer’s insightful feedback. We will address the signaled weaknesses in our final version, including in the Appendix a table with relevant notation and further reference for the main concepts. We will also further stress those aspects that, we believe, go beyond the usual intuition on symmetrically-trained NNs. These include the non-trivial, quite interesting fact that models freely-trained on equivariant data, stay for big N within the space of equivariant parameters; and the non-necessarily intuitive difference between two types of parameter distributions that naturally produce equivariant models: WI measures (which define them in general), and SI measures (which encode traditional “Equivariant Architectures” and have possibly fewer approximation guarantees).
> We next answer the questions posed:
>
> **Q1: About the connection between jointly equivariant and equivariant**
>
> In the setting of traditional NNs, the group action $M$ is determined by the action on data and on the model architecture (as displayed in Sect. A.4); and indeed a NN model parameterized by $z$ s.t. $M_g z = z$ for all $g\in G$ corresponds to an equivariant architecture (thus, an equivariant model). A slight subtlety is that a shallow NN model can combine different $z_i$, e.g. $(1/N).\sum_{i=1}^N \sigma_*(x,z_i)$, thus being an equivariant function without all the $z_i$ satisfying $M_g.z_i = z_i$ (i.e. without explicitly having an "equivariant architecture" in the traditional sense). This is the core principle behind WI distributions. Further details on the assumption of $\sigma_*$ being jointly equivariant (i.e. equivariant in both arguments) are given in our Global Response.
>
> **Q2: Regarding Proposition 5**
>
> Depending on the group considered and the architecture itself, these conditions may be generally satisfied. As mentioned in the Sect. A.4, important architectures such as CNNs and GNNs (see [35, 58, 59] for further discussion) are known to be “first-order-universal”, which implies that, taking the infinite-width limit in our setting, universality in the target class of models is guaranteed. We will elaborate more on this in Section A.4  in the final version.
>
> **Q3: On the property of data equivariance in L150**
>
> The reviewer is right. In general, $\pi$ G-equivariant implies $\pi_{\cal X}$ G-invariant [BT20]. Further, if $\pi$ is equivariant and such that $Y = f(X)$ for some function $f: \cal X \to \cal Y$, Prop.10 in Sect. A.3 implies $f$ is a G-equivariant function $\pi_{\cal X}$-a.e. Conversely $\pi_{\cal X}$ G-invariant plus $Y = f(X)$ for $f$ as before implies $\pi$ G-equivariant. As discussed in the literature (e.g. [10, 24, 34]), assuming $\pi_{\cal X}$ G-invariant might seem restrictive in some settings (e.g. in image classification, one shouldn’t assume that “images can arrive with any possible orientation” when training), but such assumption is reasonable when “exploiting symmetries” does make sense. As mentioned in remarks after Theorems 3 and 5, one could also just neglect the assumption of $\pi$ being equivariant, and introduce the inductive bias by applying data augmentation (which forces the marginal on $\cal X$ to be G-invariant). Notice also that $\pi$ “equivariant” is usually more general than the situation discussed above, including models like $Y = f(X) + \xi$ with $f$ as before, and suitable random noises $\xi$ possibly correlated to $X$.
>
> **Q4: On SI-initialized but freely-updated models remaining in ${\cal E}^G$ for large N**
>
> The exact WI-initialized teacher particles are a key part of our empirical result, as they make the teacher equivariant and thus the data distribution equivariant too. We mentioned the WI-initialized particles only implicitly in Appendix D (in terms of the symmetrization of the empirical measure), but they are exactly as stated by the reviewer: for every particle $\theta_i$, the symmetrized version has both $\theta_i$ and $g.\theta_i$, with $g$ the non-trivial element of $S_2$. Explicitly (omitting the scale parameter and the transposition for clarity), they are given by:
>
> $\theta_1 = (-1, 0, 0, 0.5)$, $g.\theta_1 = (0.5, 0, 0, -1)$
>
> $\theta_2 = (0.5, 1, 0, 1)$, $g.\theta_2 = (1, 0, 1, 0.5)$
>
> $\theta_3 = (-0.5, 0.3, 1, 0)$, $g.\theta_3 = (0, 1, 0.3, -0.5)$
>
> $\theta_4 = (0, -1, -0.5, 1)$, $g.\theta_4 = (1, -0.5, -1, 0)$
>
> $\theta_5 = (0.7, -0.7, 0.5, 0.7)$, $g.\theta_5 = (0.7, 0.5, -0.7, 0.7)$
>
> Note that, besides them being symmetric pairs, none of these particles lie within ${\cal E}^G$ (given by vectors of the form $(a, b, b, a)$ for $a,b \in \mathbb{R}$), making the empirical demonstration of Theorem 5 quite remarkable from our point of view.
>
> **Q5 on Section 4.2 and work on symmetry-discovery**
>
> We were unaware of [WHG+23] and will properly reference it. As in our work, their method starts with the most constrained architecture (in our case, the null space), respecting symmetry w.r.t. the largest possible group; and symmetries are then "broken" as models are trained on data. Their method seemingly works “one-shot”, while using a weighted combination of group convolutional filters to find “symmetry-breaks”; ours iteratively constructs an invariant linear subspace of the parameter space by adding dimensions as symmetries are broken. Our Theorem 5 guarantees that our method won’t leave ${\cal E}^G$, and part of our future work is establishing symmetry-breaking guarantees for our heuristic (comparable to their Proposition 3.1). The latter is discussed around L.1535 in Section C.3.2. in light of the MF interpretation. Finally, both methods aren’t really discovering the “underlying symmetry” of the data, but rather “optimizing architectures” compatible with such symmetries (be it the “right” subspace of equivariant parameters, or the “right” weights for each convolutional filter). Identifying the true underlying structure of symmetries is a much harder problem that is yet to be tackled in both cases.

---

> > ### Comment · Reviewer_4cbm · 2024-08-08
> >
> > Thank you for your response. My concerns have been addressed, and I'm happy to keep the current recommendation.
> >
> > I have some more comments about Q4&5. For Q4, how many particles are there in the WI-initialized teacher? Originally I thought it was 5, as you mentioned in L1585. But the WI distribution $(\nu_{\theta^*}^{N_*})^G$ has support on 10 (5 pairs of) points. Do you randomly sample 5 particles from this distribution, or do you include all 10 points? If it is the latter case, then the teacher is indeed strictly equivariant.
> >
> > For Q5, I think the difference between "discovering the symmetry of the data" and "optimizing model architectures compatible with data symmetries" is subtle. I feel these two objectives are often connected to each other. In [1], for example, one can tell from the optimized architecture what is the maximal symmetry in data. This is also done in [2] but in a quite different way, by solving the Lie derivative constraint wrt a non-equivariant network. Conversely, one can also start by discovering the data symmetry and design equivariant architectures accordingly [3]. These works may be relevant if you plan to further investigate the idea in Sec 4.2.
> >
> > [1] Discovering Symmetry Breaking in Physical Systems with Relaxed Group Convolution. ICML 2024.
> >
> > [2] LieGG: Studying Learned Lie Group Generators. NeurIPS 2022.
> >
> > [3] Generative Adversarial Symmetry Discovery. ICML 2023.

---

> > > ### Author Response · Authors · 2024-08-09
> > >
> > > Thank you very much for your response, and we are happy as well for your positive recommendation. Regarding your follow-up questions:
> > >
> > > Regarding L.1585, we agree that the current phrasing can easily lead to confusion, and we will be sure to change it in the final version. We only wanted to refer to the arbitrary teacher having 5 particles and, indeed, as you have noticed, the WI teacher has exactly 10 particles (which are the ones we provide in our original rebuttal). Namely, it is exactly $G$-equivariant (and not just an empirical sample from a $G$-equivariant measure). As we mention in our original rebuttal, the current version “implicitly” mentioned both of these facts, as from our remarks in lines 196-203 and the phrase in line 1587-1588; but we will surely make this clear and explicit in our final version.
> > >
> > > We do want to stress that the exact equivariance of the WI teacher doesn’t mean the problem is any easier. Since the WI-teacher’s particles don’t live in ${\cal E}^G$, one intuitively would believe that the freely-trained particles of a NN (initialized within ${\cal E}^G$) would prefer to “escape” ${\cal E}^G$ in order to better approximate the teacher. However, as shown by our numerical experiments, this doesn’t happen, and the particles remain (up to numerical error) within ${\cal E}^G$.
> > >
> > > Regarding Q5, we deeply thank you for the provided references and ideas. We will be sure to consider them for our upcoming works regarding the topic.

---

### Official Review · Reviewer_2QUz · 2024-07-12

**Soundness:** 3
**Presentation:** 2
**Contribution:** 3
**Rating:** 7
**Confidence:** 3

**Summary:**

This work studies the symmetric structure of the model and data structures with respect to the action of $G$ and studies the Wasserstein gradient flow (WGF) for learning overparameterized neural networks under the mean-field (MF) regime. In particular, the authors consider data augmentation (DA), feature averaging (FA), and equivariant architecture (EA) as symmetry-leveraging techniques. They show that for symmetric data, DA and FA follow the same dynamics by exploiting weakly and strongly invariant laws (WI and SI) on the parameter space. In particular, the following statements are given.

- DA and FA are equivalent in terms of optimal values and also equivalent to standard training (free training) for WI measures (Theorem 2).
- Furthermore, this equivalence is shown over the space of general measures when the data distribution is equivalent (Corollary 2).
- WGF for an invariant functional preserves weak invariance of the measure (Theorem 3 and Corollary 3). Moreover, there is an equivalence of WGF for DA, FA, and free training under suitable conditions.

**Strengths:**

The paper proposes the notion of WI and SI and proves several equivalences between DA, FA, and free training. These results indicate when these techniques (DA and FA) are meaningful.

**Weaknesses:**

- This work mainly focuses on two-layer neural networks; hence, it is unclear if the results can be applied to deep neural networks.
- The results, especially for WGF, seem to fully rely on the mean-field limit $N\to\infty$. Hence, it’s unclear whether the theory has some implications for finite-neuron settings.
- (Minor) The paper is somewhat hard to follow because of the numerous notations. I suggest including a table that summarizes these notations.

**Questions:**

- In Theorem 4, the noisy SGD (4) involves the projection step of the noise. I think such a projection is non-trivial in general. Are there any practical examples?
- Is WGF essential for the results in Section 3.3? Do similar results hold for other dynamics (e.g., underdamped version)?

**Limitations:**

Limitations are addressed in the paper.

---

> ### Author Rebuttal · Authors · 2024-08-06
>
> We thank the reviewer for their relevant feedback about the paper. We will make sure to address the detected weaknesses in our final version, specifically:
>
> **W1: Regarding our work mainly focusing on two-layer neural networks.**
>
> As mentioned in our Global Response, the full MF theory for deep NNs is an open, active research endeavor, in which sustained progress is being done. We think that a significant part of our results can be extended to settings in which MF results on deep models are (or will be) available; this has been left for future work, however, since the involved technicalities are usually quite intricate. Nevertheless, as noted in our Global Response, our current setting already allows to go far beyond two-layer NNs (and we will be sure to further stress this in the core of our work). Indeed, our "activations" $\sigma_*$ are general functions on parameter spaces that can, by themselves, encode complex deep architectures (including e.g. kernels in CNNs, heads in Graph Attention Networks, etc). Thus, the resulting "shallow models" can represent large stacks of such (jointly-trained) multilayer “units”; i.e. deep models which, although not fully general, are way more interesting than single-hidden-layer NNs. Finally, as put forward in most of the growing literature on the MFL of overparametrized NNs (see references cited in the paper), the MF behavior can already become apparent for reasonable/realistic finite values of $N$; which speaks to a possibly broad applicability of such framework. Along this line, see our comments below regarding W2.
>
> **W2: On our results fully relying on the mean-field limit as $N\to \infty$ and the theory having no clear implications for finite $N$**
>
> Although not fully conclusive, the MFL and its associated WGF is one of the most promising approaches to mathematically explain the generalizability capacity of the SGD training dynamics of overparametrized NNs. Furthermore, it is the asymptotic approximation of NNs known to better describe their feature learning capabilities [36, 37, 38, 46]. In practice, as shown both by experiments in those papers and by our numerical results, the MF viewpoint provides truly useful insights (and new results) on the training dynamics, that manifest themselves even for relatively small numbers of neurons (1000 was enough in our case).
>
> But not only does the MF theory provide new insights for finitely many (though large numbers of) neurons, that would be unaccessible from a fixed $N$ standpoint (as is the case for Theorem 5). Ongoing research in this field (based on theoretical tools such as propagation of chaos, optimal transport, WGFs and functional inequalities, see e.g. [9,18]) also has the potential to provide non-asymptotic estimates and quantitative answers to concrete questions with practical implications (e.g. How many neurons and training steps should be enough in order to attain a given level of generalization/population error? Which architecture/training method takes better advantage of given, limited resources?). We thus expect the MF theory to increasingly bring useful insights and concrete implications for finite-neuron settings, including for the issues addressed in this work. We will give further details on non-trivial implications of our theoretical results, for the finite-N setting, in the final version of our work.
>
> **W3: On the paper being somewhat hard to follow because of the numerous notations**
>
> As stated in our Global Response, we will include a table explaining our relevant notation.
>
> We now address the proposed questions:
>
> **Q1: Regarding the projected noisy SGD dynamics.**
>
> - Indeed, as we also mention in the core of our work, computing the projection onto ${\cal E}^G$ is a highly complex problem (see [23, 24]) that can’t always be efficiently solved in practice. However, our alluded results hold even taking $\beta = 0$; that is, without including any projected noise (and thus, without ever needing to compute it).
> - As mentioned in our remarks and our experimental results, any shallow NN with all parameters initialized at {0}, being trained freely with traditional SGD and without noise, will satisfy Theorems 5 and 6 in the MF limit. We believe that the previous conditions (modulo the shallowness of the NN and taking a big N) are all fairly easy to achieve in practice. In fact, this is exactly the case of our numerical experiments (where we never assume to know G and never explicitly compute such projection; see our Global Response for details).
> - The projection of the noise is only included as a tool to allow for easy “global convergence” guarantees for the dynamic within the space of SI-distributions (following the traditional MF literature on the topic [9, 28, 38], among others): a “general” noise cannot be used, as it would always remove the dynamic from ${\cal E}^G$. In short, the projection of the noise in (4) is far from being fundamental in the proofs of Theorems 5 and 6, and their further application.
>
> **Q2: Regarded dynamics beyond the WGF**
> - In general, we remained within the domain of the classical WGF dynamics following the cited (standard) MF literature on the topic; without looking much into other variants of the dynamics, but also without discarding the applicability of our results therein.
> - Indeed, we believe Theorems 3 and 4 to be “natural” results, and expect them to hold for many potentially different training dynamics (e.g. Wasserstein sub-Gradient Flows [13], underdamped dynamics [FW23], annealed dynamics [12], among many others).
> - From [24], we know that Theorem 5 holds in a finite-N setting when exact DA is applied during training. The interesting MF phenomenon is that it also holds under free training with randomly sampled data (Theorems 5 and 6). This result doesn’t seem immediate to generalize for other asymptotic dynamics (e.g. annealed or underdamped), and we thus consider it an interesting question to be tackled in future work.

---

> > ### Comment · Reviewer_2QUz · 2024-08-12
> >
> > Thank you for the detailed response. The authors have adequately addressed my concerns. Given the strength of the paper, some limitations seem negligible. I will raise my evaluation. The discussion provided in the general comments will significantly help readers understand the importance and applicability of the theory.

---

> > > ### Author Response · Authors · 2024-08-12
> > > **Thank you**
> > >
> > > We thank you for pointing out your concerns, as they have undoubtedly helped us provide a clearer exposition of our central ideas and contributions in the final version of our paper. We are also thankful for your positive comments and the evaluation adjustment.

---

### Official Review · Reviewer_j4Mu · 2024-07-14

**Soundness:** 3
**Presentation:** 2
**Contribution:** 3
**Rating:** 7
**Confidence:** 3

**Summary:**

This paper presents a mean-field analysis on a class of overparameterized neural networks that are expressed as an ensemble of $N$ units, trained with SGD under symmetries in data distribution and possible use of symmetry-leveraging techniques (data augmentation, feature averaging, and equivariant architectures). While the analysis is based on prior work on mean-field analysis on the same class of neural networks e.g. based on Wasserstein gradient flows [1], the contribution of this work is in extending to learning under symmetries. The setup is as follows:

- In considered neural networks, a unit computes $\sigma_*:(\mathbf{x},\theta_i)\mapsto\hat{\mathbf{y}}$ and the network computes $\Phi_{\theta}^N:(\mathbf{x},\theta)\mapsto \frac{1}{N}\sum_{i=1}^N\sigma_*(\mathbf{x},\theta_i)$.
- Overparameterization means the width $N\to\infty$, yielding a limiting ensemble $\Phi_\mu$ where $\mu$ is understood a probability measure on the space of parameters $\theta_i$.
- The parameters $\theta$ are optimized using noisy SGD.

The mean-field theory of the neural networks considered above [1] aims to analyze the training dynamics of the limiting ensemble $\Phi_\mu$, given that its optimization using a convex loss function becomes a convex optimization problem (unlike $\Phi_{\theta}^N$ which induces non-convex optimization) and that its global optimum may be approximated by training $\Phi_{\theta}^N$ with a large width $N$. Specifically, it is known that SGD on $\Phi_{\theta}^N$ approximates Wasserstein gradient flow on $\Phi_\mu$ in the scaling limit $N\to\infty$ under certain assumptions, which then, under certain assumptions, converges to the global optima of the convex optimization problem.

Given the background, the paper presents a mean-field analysis given that the data generating distribution is possibly under a symmetry described by a compact group $G$, and symmetry-leveraging techniques are possibly used. The authors consider data augmentation, feature averaging (i.e. symmetrization; also called the Reynolds operator), and equivariant architectures, in accordance to literature [2]. The idea for the analysis is to distinguish between two types of symmetries for the measure $\mu$ on parameters: weakly-invariant and strongly-invariant, where weakly-invariant refers to an invariant measure on the subspace containing possibly non-invariant elements, and strongly-invariant refers to a measure on invariant subspace. The authors first show in Section 3.1 that, given that the unit $\sigma_*(\cdot, \cdot)$ is jointly equivariant, $\Phi_\mu$ is equivariant if and only if $\mu$ is weakly-invariant. Then, feature averaging (and data augmentation) can be understood as being related to weakly-invariant distributions obtained through symmetrization, and equivariant architecture is related to strongly-invariant distributions.

The authors then, in Section 3.2, begin mean-field analysis on optimality under a set of assumptions including that the unit $\sigma_*(\cdot, \cdot)$ is jointly equivariant (Assumption 1), and show that:

- (Proposition 3) Optimizing with respect to an invariant loss function yields a weakly-invariant optimum if it has a unique minimizer,
- (Corollary 1) If input data distribution is invariant, optimizing under data augmentation and feature averaging are equivalent and corresponds to approximating the symmetrized version of mean of the labeling function,
- (Corollary 2) If data distribution is equivariant, optimizing under data augmentation and feature averaging provide no advantage,
- (Proposition 4 and 5) But even when data distribution is equivariant, equivariant architectures (i.e. strongly invariant measures $\mu$) may not achieve the optimum with respect to a given invariant loss function, unless universality properties that are "good enough" for the given data distribution are assumed.

Then, in Section 3.3, the authors provide mean-field analysis on training dynamics, and show that:

- (Theorem 3, Corollary 3, and Theorem 4) Under an invariant loss function and Assumption 1 (among others), a training dynamics $\mu_{0:t}$ following the Wasserstein gradient flows yields a unique weakly-invariant solution $\mu_t$. If the data distribution is equivariant, from Corollary 2 this result applies to freely-trained neural networks without symmetry leveraging techniques, and data augmentation and feature augmentation again provide no advantages.
- (Theorem 5 and 6) Under an invariant loss function and Assumption 1 (among others), a training dynamics $\mu_{0:t}$ starting at a strongly-invariant initial condition $\mu_0$ and following the Wasserstein gradient flows yields strongly-invariant solutions $\mu_t$. Importantly, this means that if the data distribution is equivariant, $\mu_t$ stays strongly-invariant throughout training even though it is not explicitly forced or constrained to be invariant. Furthermore, equivariance of data distribution can be dropped if one of the symmetry leveraging techniques are used, which all yield coinciding solutions.

In Section 4, the authors provide synthetic experiments focusing on validating Theorem 3-6, and then propose and demonstrate an empirical method for finding good equivariant architecture for the task at hand (i.e. the most expressive among invariant subspace) by training according to Theorem 5 and 6 until convergence.

[1] Lenaic Chizat and Francis Bach. On the global convergence of gradient descent for over-parameterized models using optimal transport. Advances in neural information processing systems, 31, 2018.

[2] Clare Lyle, Mark van der Wilk, Marta Kwiatkowska, Yarin Gal, and Benjamin Bloem-Reddy. On the benefits of invariance in neural networks. arXiv preprint arXiv:2005.00178, 2020.

**Strengths:**

- S1. The paper targets an important and challenging problem of understanding training dynamics of overparameterized neural networks under symmetries and their leveraging. The technical approach that applies mean-field analysis of generalized shallow neural networks [1] to learning under symmetries is original as far as I am aware.
- S2. The paper considers three types of symmetry-leveraging techniques: data augmentation, feature averaging, and equivariant architectures. These three are sufficiently general to represent currently used techniques [2]. The descriptions as well as strengths and weaknesses of each technique discussed in Section 2.3 are correct.
- S3. The authors' approach of imposing distributional symmetries on the measure $\mu$ to describe equivariant architectures as strongly-invariant (and feature averaging as weakly-invariant given that the unit $\sigma_*(\cdot, \cdot)$ is jointly equivariant) is interesting and could be potentially useful in future work.
- S4. The results presented in Section 3 and 4 are interesting, in particular Theorems 5 and 6 which shows that overparameterization and equivariant units, combined with equivariant data distribution or symmetry leveraging, leads to the property that strongly-invariant parameter initialization stays strongly-invariant throughout the training even without explicit constraints to do so. (But also see W1.)

[1] Lenaic Chizat and Francis Bach. On the global convergence of gradient descent for over-parameterized models using optimal transport. Advances in neural information processing systems, 31, 2018.

[2] Clare Lyle, Mark van der Wilk, Marta Kwiatkowska, Yarin Gal, and Benjamin Bloem-Reddy. On the benefits of invariance in neural networks. arXiv preprint arXiv:2005.00178, 2020.

**Weaknesses:**

- W1. I have concerns regarding Assumption 1, which is a key assumption underlying Section 3 and assumes that the unit $\sigma_*:(\mathbf{x},\theta_i)\mapsto\hat{\mathbf{y}}$ is jointly equivariant with respect to the input $\mathbf{x}$, the parameters $\theta_i$, and the output $\hat{\mathbf{y}}$. This assumption states that we have already (partially) baked in the symmetry constraint to our model class. This may be natural for analyzing equivariant architectures, but does not coincide with the practical uses of data augmentation and feature averaging (e.g. please see [2]), as they are generally applied to an arbitrary, non-equivariant function, while in the setup of this work they are applied on already equivariant units. This limits the usefulness of the results regarding data augmentation and feature averaging. Perhaps the setup of this work could be relevant to symmetry breaking in already equivariant architectures [3], but I am not entirely sure.
- W2. In Line 92-93, the authors claim that "Note that requiring an infinite and i.i.d. sample from $\pi$ is key when letting later $N\to\infty$. This is not truly a restriction, since one can always sample from the empirical distribution of finite data points". This seems problematic since (1) if we can infinitely sample from $\pi$, then we have access to it, which violates basic assumptions in machine learning (Line 82), and (2) if we infinitely sample from an empirical distribution supported on finite set of data points, the resulting distribution would not be $\pi$, and consequently the SGD scheme in Equation (1) would not be optimizing the objective in Equation (2). (Please correct me if I am wrong.)
- W3. The writing of the paper could be in general improved. For example, the paper uses an excessive number of abbreviations (MF, MFL, NN, SL, DA, FA, EA, WI, SI, WGF, RMD, V), which is hurting readability. It would be also helpful to provide an example of a measure and its (non-equivalent) invariant symmetrization and projection in Definition 5; these are important concepts that are used in all later sections, but I was not able to directly grasp them from reading the text. For example, considering a uniform distribution on $[0, 1]\subset\mathbb{R}$, and a multiplicative group {1, -1} acting on $\mathbb{R}$ by multiplication, may be sufficiently informative.
- W4. The architecture discovery algorithm proposed in Section 4.2 was a bit questionable to me, since the architecture of the ensemble is already specified (at least up to a significant portion) by the jointly equivariant unit(s) $\sigma_*(\cdot,\cdot)$, and the algorithm searches for their parameters within a strongly-invariant regime. I am not sure why this algorithm can be understood as doing architecture discovery, instead of doing the usual parameter optimization.
- W5. Currently, it is not immediate how the study of the ensemble considered in the paper would benefit the development of practical neural networks that leverage symmetry. Showing how some of the currently used symmetry-leveraging deep neural networks can be viewed as such ensembles could be informative in this directions.

[2] Clare Lyle, Mark van der Wilk, Marta Kwiatkowska, Yarin Gal, and Benjamin Bloem-Reddy. On the benefits of invariance in neural networks. arXiv preprint arXiv:2005.00178, 2020.

[3] Rui Wang, Elyssa Hofgard, Han Gao, Robin Walters, and Tess E. Smidt, Discovering symmetry breaking in physical systems with
relaxed group convolution, arXiv preprint arXiv:2310.02299, 2023.

**Questions:**

Please see the Weaknesses section.

**Limitations:**

The authors have partially addressed limitations of their work, for example in Section 5. Since the discussions are currently scattered across the main text, I encourage the authors to gather them in a dedicated limitations section.

---

> ### Author Rebuttal · Authors · 2024-08-06
>
> We thank the reviewer for their careful reading and relevant feedback on the paper.
> We will incorporate an explicit Limitations section in the Appendix of the revised version, as well as a simple yet clarifying example of the different notions of symmetric distributions (e.g. if ${\cal Z}=\mathbb{R}$ with the multiplicative action of $G=$ { -1, 1 }, setting $\mu = \delta_x$ for $x \neq 0$, we get $\mu^G = \frac{1}{2} (\delta_x + \delta_{-x})$, and $\mu^{{\cal E}^G} = \delta_0$). The beginning of our Global Response mentions other clarifying sections we plan to add.
>
> We next address the perceived weaknesses:
>
> **W1**
>
> Your concerns regarding the “a priori knowledge” of symmetries that’s encoded in the fact that $\sigma_*$ is assumed jointly-equivariant, are generally justified, but miss a key point in the setting of the paper. We refer the reviewer to our Global Response, where we provide thorough revision of the assumption of $\sigma_*$ being jointly-equivariant (also partially addressed in Appendix A.4; we will be sure to make it more explicit in the core of our work). In short, $\sigma_*$ can describe many configurations beyond single-hidden-layer NNs, and the property of it being jointly-equivariant doesn’t necessarily imply that “everything inside it” needs to be equivariant as well. For instance, if we take $\sigma_*$ to be the “traditional” single-hidden-layer NN and we consider the group action on $\cal Z$ to be trivial on the hidden layer; then all our results apply with no constraints on the architecture whatsoever (see our Global Response and the proof of Proposition 4). In particular, DA, FA and “vanilla” training with distributionally equivariant data are still tightly related in terms of their optimization and WGFs (regardless of the properties of the inner activation). More interestingly, when $M$ is taken to be a more intricate action (and the unit $\sigma_*$ respects it), the EAs are also more attractive, and our results on DA and FA work just as well.
>
> **W2**
>
> We agree with you on the fact that performing SGD with an infinite i.i.d. sequence sampled from $\pi$ is not equivalent to sampling infinitely many times with replacement from a (finite) given empirical measure $\pi_k$, “previously” obtained by sampling $k$ points from $\pi$. What we wanted to stress is that, from the theoretical point of view, our treatment of the first and second cases (often respectively called “population SGD” and “empirical SGD” in the literature) is indistinct and can be applied to both situations. We will rewrite said statement, to make our intent clearer. We also agree that having access to an infinite i.i.d. sequence of a given law can be considered a way of “accessing” the whole distribution but, of course, we do not expect this to be the actual application setting. Our results, as well as the whole MF approach to NNs, should be understood as an approximation that is valid for a large enough i.i.d. sample.
>
>
> **W4**
>
> We believe that our discovery heuristic can be nicely justified through our numerical experiments. As mentioned in our Global Response, our experiments are run with a traditional single-hidden-layer NN without any specific regard for the architecture (by default, code frameworks often apply activations pointwise). In particular:
> - We assume only that the symmetry group acts via permutations, and little to no care is put into truly encoding anything into $\sigma_*$ (see our Global Response for further discussion). In particular, we never assume to “know” the group $G \leq S_n$ encoding data symmetries.
> - We simply initialize our architecture with all parameters being 0, and let it train unconstrained for a set number of epochs. Theorem 5 guarantees that such training will never leave the subspace ${\cal E}^G \leq \cal Z$, which is possibly of much lower dimension than $\cal Z$ (e.g. imagine convolutional layers as a subspace of all possible linear layers between images).
> - Our heuristic iteratively discovers ${\cal E}^G$, by capturing the directions on which the trained parameters “break” the symmetry (under the assumption that this indeed happens). Despite not explicitly “discovering” new ways of building NNs, it does provide the “most expressive weight-sharing scheme” that respects data symmetries. It also yields a “basis” for such space, allowing to later build NNs with $dim({\cal E}^G) << dim(\cal Z)$ parameters.
> - Indeed our heuristic is only doing parameter optimization, but our theory (cf. Theorem 5) guarantees that it will never leave a strict, possibly much smaller subspace of $\cal Z$.
>
> In short, the “discovery” part is that we start knowing little to nothing about our data symmetries (e.g. only that they are given by a finite group acting via permutations), and we end up with a way of building strongly-equivariant NN (i.e. more “parameter-efficient” than what we started with) in the smallest possible parameter subspace signaled by the data itself.
>
> **W5**
>
> Architectures such as DeepSets, GNNs and CNNs (which are massively used in practice) were key pieces of inspiration for the development of our work. Namely, we know that our “unit” $\sigma_*$ allows for modeling such equivariant architectures both as wide single-hidden-layer networks, and as large ensembles of multilayer NNs (both in such a way that $\cal Z$ is a large linear space and ${\cal E}^G$ represents exactly the desired EA). We will be sure to include a section in the Supplementary Material where some of these practical examples shall be described in detail. Knowing this, and despite the theoretical nature of our work, we believe that our results in Section 3 could provide key guarantees and ideas for users of practical equivariant architectures.

---

> > ### Comment · Reviewer_j4Mu · 2024-08-08
> >
> > Thank you for the comprehensive rebuttal. I am trying to technically understand the response on the joint-equivariance of single-hidden-layer NN when the group acts trivially on the hidden layer (regarding W1).
> >
> > Following Line 65-67, let $\mathcal{X}=\mathbb{R}^d$, $\mathcal{X}=\mathbb{R}^c$, and $\mathcal{Z}=\mathbb{R}^{c\times b}\times \mathbb{R}^{d\times b}\times \mathbb{R}^b$, $z = (W, A, B)\in \mathcal{Z}$, $\sigma:\mathbb{R}^b\to\mathbb{R}^b$, and $\sigma_*(x,z)\coloneqq W\sigma(A^\top x+B)$. From Line 170-171, the joint-equivariance is written as $\sigma\_* (\rho\_g \cdot x, M\_g \cdot z) = \hat{\rho}\_g \sigma\_* (x,z)$ for all $g,x,z\in G\times \mathcal{X}\times \mathcal{Z}$.
> >
> > Now let $M\_g$ be a trivial action. Then joint-equivariance is required as $\sigma\_* (\rho\_g \cdot x, z) = \hat{\rho}\_g \sigma\_* (x,z)$, or equivalently $W\sigma(A^\top \rho\_g \cdot x+B) = \hat{\rho}\_gW\sigma(A^\top x+B)$, for all $g,x,z\in G\times \mathcal{X}\times \mathcal{Z}$. As far as I am aware, this is not true in general; let $A=0$, $\sigma=\mathrm{id}$, $W=I$, and let $B\in\mathbb{R}^b$ have distinct entries, then we have $\sigma\_* (\rho\_g \cdot x, z) = B$ which is not equal to  $\hat{\rho}\_g \sigma\_* (x,z) = \hat{\rho}\_gB$ for faithful actions $\hat{\rho}_g$. Therefore I do not see how joint-equivariance holds for single-layer NNs if the group acts trivially on the hidden layer. Am I missing something?

---

> > > ### Author Response · Authors · 2024-08-09
> > >
> > > We thank you very much for your response, and we will take the opportunity to provide a clarifying example for the question at hand.
> > >
> > > As noted in  Lines 1012-1016, under the same example you are considering, the “natural” action of $G$ on the parameter space ${\cal Z} = \mathbb{R}^{c \times b} \times \mathbb{R}^{d \times b}\times \mathbb{R}^b$ corresponds to taking, for $g \in G$ and $(z = W, A, B) \in \cal Z$: $$M_g.z = M_g.(W, A, B) := (\hat{\rho}_g.W.\eta_g^T, \rho_g.A.\eta_g^T, \eta_g.B). $$
> > >
> > > This is what we refer to as the “intertwining action” on the layer parameters, as it recalls the traditional definition of an intertwinning linear map from representation theory (also, see the discussion on lines 165-180). Notice that, indeed, this involves the action of the group on the corresponding input/output spaces of each layer (e.g. from $\mathbb{R}^d$ to $\mathbb{R}^b$ and from $\mathbb{R}^b$ to $\mathbb{R}^c$).
> > >
> > > In our original rebuttal, notice that we referred to “a trivial action on the intermediate layer”, and not plainly “a trivial action”. Indeed, we referred to taking $\eta$ to be trivial, rather than $M$ itself; the latter, as you have correctly noticed, doesn’t work in general.
> > >
> > > Thanks to the orthogonality (and linearity) of our group action, a similar development to yours yields, for any $\sigma: \mathbb{R}^b \to \mathbb{R}^b$, any $g \in G$ and for any $z = (W, A, B) \in  \cal Z$:
> > >
> > > $$\sigma_*(\rho_g.x, M_g.z) = \sigma_*(\rho_g.x, (\hat{\rho}_g.W.\eta_g^T, \rho_g.A.\eta_g^T, \eta_g.B)) =(\hat{\rho}_g.W.\eta_g^T). \sigma((\rho_g.A.\eta_g^T)^T.(\rho_g.x) + \eta_g.B)$$
> > > $$ = \hat{\rho}_g.(W.\eta_g^T. \sigma(\eta_g. A^T.(\rho_g^T.\rho_g).x + \eta_g.B)) = \hat{\rho}_g.(W.\eta_g^T. \sigma(\eta_g. (A^T.x + B)))$$
> > >
> > > Namely, it is enough that $\sigma: \mathbb{R}^b \to \mathbb{R}^b$ is $G$-equivariant (w.r.t. the action of $\eta$) in order to get $\sigma_*$ to be jointly equivariant. Indeed, this would yield:
> > >
> > > $$\sigma_*(\rho_g.x, M_g.z) = \hat{\rho}_g.(W.\eta_g^T.\eta_g. \sigma(A^T.x + B)) = \hat{\rho}_g.(W.\sigma(A^T.x + B)), $$
> > >
> > > as required. This justifies our claims about the pointwise activations and/or the use of a Norm-ReLU activation to ensure the joint-equivariance of $\sigma_*$ for different levels of complexity of $G$ (and its action on the intermediate layer).
> > >
> > > More specifically, if this action on the intermediate layer $\eta$ is taken to be trivial, any function $\sigma: \mathbb{R}^b \to \mathbb{R}^b$ is $G$-equivariant w.r.t. $\eta$  (indeed, $\sigma(\eta_g.x) = \sigma(x) = \eta_g.\sigma(x)$); i.e. no matter the inner activation function $\sigma$, under this action $M$, $\sigma_*$ is jointly-equivariant.
> > >
> > > In the end, a trivial $\eta$ assumes “no action of the group on the latent space”; and so we only act on the parameters on the dimensions defined by the data: e.g. $\rho$, that acts on $\cal X$, will act only on $A$ on the left; and $\hat{\rho}$, that acts on $\cal Y$, will act only on $W$ on the left as well. This readily generalizes to the case where $\sigma_*$ represents a whole multilayer unit, for which we take the action on parameters to be trivial on the hidden layers, but not on the input and output layers (where it acts according to the data). Indeed, no matter the inner activations, such $\sigma_*$ would be jointly equivariant with respect to such group action.
> > >
> > > As we also mention in our original rebuttal, a “trade-off” follows from considering such a “boring” group action $M$: the space ${\cal E}^G$ might not represent very “interesting” equivariant architectures. This doesn't seem too problematic since it wouldn’t be the use case if $\sigma$ didn’t respect any sort of equivariance to begin with. It does, however, allow to state that DA and FA are tightly related, even for unconstrained architectures. This also makes sense when we consider that DA and FA only interact with NN models through the input and output layers.
> > >
> > > We hope to have clarified your question with this example, and we’re open to answer further questions if required.

---

> > > > ### Comment · Reviewer_j4Mu · 2024-08-10
> > > >
> > > > Thank you for the thorough explanation. Together with the original response, my major concerns have been addressed. I will adjust my score accordingly.

---

> > > > > ### Author Response · Authors · 2024-08-12
> > > > > **Thank you**
> > > > >
> > > > > We thank the reviewer very much for their positive comment and for adjustment of their score.

---

### Author Rebuttal · Authors · 2024-08-06

We thank all reviewers for their insights. We here present information to help transversally clarify some of their questions. References from the manuscript are mentioned as presented therein; new references follow the format at the end of this response.

First, for better understandability of the paper, in the Appendix we will add: a table explaining relevant notation/abbreviations; a summary for a wide audience on fundamental notions used in the paper; a summary of Limitations clarifying its applicability; and an explanation on how some well-known equivariant architectures (CNNs, GNNs, DeepSets) enter our setting.

We next give general context on how close our setting is to practical applications of NNs:
- Despite the MF limit of NNs being a theoretical tool, it is the asymptotic regime that most closely describes the actual behavior of large NNs during training (as compared e.g. to the Neural Tangent Kernel), and we believe truly useful insight can be obtained from it: our experiments show that the predicted MF behavior emerges in practice already for finite, not too large $N$ (1000 was generally enough), in  reasonable practical settings of shallow NNs (standard pointwise sigmoid activation and objax’ default SGD training were used).
- A fully unified, satisfactory MF theory for deep NNs is still an open, actively tackled question (for advancements on it see e.g. [SS21],[NP23]). We believe some of our key results (e.g. Theorem 3) can be extended to such settings, undoubtedly in a future research line. Notice though that $\sigma_*$ can by itself encode a complex deep architecture (see below), and the resulting shallow model can represent way more interesting structures than single-hidden-layer NNs (e.g. ensembles of such multilayer “units” trained in parallel).

We also clarify further the “jointly-equivariant” assumption on $\sigma_*$:
- We defined our shallow models through an abstract, general $\sigma_*$, termed “activation” in the MF literature, but more complex in general than the “usual” activations. Since no constraints (beyond being orthogonal) are put on the action of $G$ on $\cal Z$, this “joint-equivariance” plays a key role in connecting the G-action on the data with the one on parameters. It doesn’t, however, imply that all “inner components” of $\sigma_*$ are equivariant.
- The joint-equivariance of $\sigma_*$ can encode a wide range of situations and  allows us to find interesting results beyond particular choices of architecture. It also encodes possible trade-offs between “freedom” in the architecture and “intricacy” of the group action $M$.
- In “traditional” single-hidden-layer NNs (cf. Appendix A.4) $\sigma_*$ is defined in terms of a “usual” activation function $\sigma:\mathbb{R}^b \to \mathbb{R}^b$ on the hidden layer and $M$ in terms of G-actions on the data (via $\rho$ and $\hat{\rho}$) and on the hidden layer (via $\eta$). These choices determine how constraining the “jointly-equivariant $\sigma_*$” assumption is. For instance: If $\eta$ is trivial (i.e. $\eta \equiv id_{\mathbb{R}^b}$), any single-hidden-layer NN is jointly-equivariant, regardless of $\sigma:\mathbb{R}^b \to \mathbb{R}^b$ (see e.g. the proof of Prop. 4). Similarly, any multilayer architecture under a G-action acting trivially on the hidden layers (but not on input/output), also results in a jointly-equivariant $\sigma_*$. Thus, our results apply to general architectures if the G-action on $\cal Z$ is well chosen. The catch is that, under such G-actions, the equivariant architectures encoded in ${\cal E}^G$ might end up being uninteresting (and so, Theorem 5 loses part of its punch); however, results relating DA, FA and vanilla training still apply.
- If G acts via permutation of input coordinates (as commonly for finite groups), $\sigma_*$ will be jointly-equivariant for any activation function $\sigma:\mathbb{R}^b \to \mathbb{R}^b$ applied pointwise (a common practice). The same holds in many other interesting cases involving finite groups (e.g. $Z_n \times Z_n$ acting on square images, $S_n$ acting on graphs/sets, etc). This discussion is also tackled in [24] regarding a similar assumption. For a more complex (possibly infinite) group acting orthogonally on the data and parameters (with non-trivial $\eta$), for $\sigma_*$ to be jointly-equivariant we have to restrain the chosen $\sigma:\mathbb{R}^b \to \mathbb{R}^b$. For instance, an O(b)-equivariant $\sigma$ (e.g. a Norm-ReLU) would grant our results to apply; but this particular choice could potentially harm the model’s expressiveness and applicability.

Last, we thank the reviewers for pointing out relevant references. We shall properly cite [WHG+23] in Section 4.2. We have also become aware of the work [HC23] describing a result comparable to Corollary 3, but in the particular case of ReLU activations and under a single symmetry transformation. We think our work branches far beyond both mentioned papers, and proper references for them will be included.

**References**
- [BT20] Benjamin Bloem-Reddy and Yee Whye Teh. 2020. Probabilistic symmetries and invariant neural networks. J.Mach.Learn.Res. 21, 1, Article 90 (January 2020), 61 pages.
- [FW23] Qiang Fu and Ashia Wilson. Mean-field Underdamped Langevin Dynamics and its
Space-Time Discretization. In: arXiv preprint arXiv:2312.16360 (2023).
- [HC23] Karl Hajjar and Lénaïc Chizat. On the symmetries in the dynamics of wide two-layer neural networks. Electron.Res.Arch., 31(4):2175–2212, 2023.
- [NP23] Phan-Minh Nguyen and Huy Tuan Pham, A rigorous framework for the mean field limit of multilayer neural networks. Math.Stat.Learn. 6 (2023), no. 3/4, pp. 201–357
- [SS21] Justin Sirignano, Konstantinos Spiliopoulos (2021) Mean Field Analysis of Deep Neural Networks. Maths.Oper.Res.47(1):120-152.
- [WHG+23] Rui Wang, Elyssa Hofgard, Han Gao, Robin Walters, and Tess E. Smidt, Discovering symmetry breaking in physical systems with relaxed group convolution, arXiv preprint arXiv:2310.02299, 2023.

---

### Decision · Program_Chairs · 2024-09-25

**Decision:**

Accept (spotlight)

**Comment:**

A fine work investigating the Mean Field limit of overparametrized shallow models under symmetric data and/or symmetry-leveraging techniques. The reviewers have unequivocally voted for acceptance.